# Chronically implantable $\mu$LED arrays for optogenetic cortical surface stimulation in mice

Ryan Greer [1,3] ✉, Antonin Verdier [2,3], Emma Butt [1], Yunzhou Cheng[1], Ella Callas[2], Niall McAlinden [1], Alicia Aniorte[2], Eya Mabrouk Kakaouia[2], Magdalena Pereyra [2], Martin D. Dawson [1], Brice Bathellier [2,4] & Keith Mathieson [1,4] ✉

Cortical implants are a proven clinical neurotechnology with the potential to transform our understanding of cognitive processes. These processes rely on complex neuronal networks that are difficult to selectively probe or stimulate. Optogenetics offers cell-type specificity, but achieving the density and coverage required for chronic, high-resolution modulation remains a challenge. We fabricated 100-element $\mu$LED arrays (200 $\mu$m pixel pitch, $2 \times 2$ mm$^2$ footprint) coupled into chronically implantable systems for optogenetic stimulation of the mouse cortex. The $\mu$LEDs remain stable for over 300 hours continuous operation time in vivo, allowing for months-long chronic experiments. Simultaneous electrophysiology recordings confirmed robust neuronal responses at low $\mu$LED drive currents (< 5 mA), minimising thermal effects. Here we show that our device can be chronically implanted in freely-behaving mice and drive behavioural outcomes from spatiotemporal, patterned, optogenetic stimulation of auditory cortical circuits.

Cortical implant technologies have received significant attention in recent years, demonstrating their potential in clinical neuroscience[1,2]. Among their many applications, the transfer of information to brain networks holds particular promise for investigating and restoring lost function or sensation. Optogenetics enables precise modulation of neuronal activity using light, offering cell-type specificity by employing light-sensitive proteins (opsins), which are genetically introduced into specific neuronal populations. This motivates the development of optical cortical implants, tailored for optogenetic neuroscience studies in behaving animals. These devices would ideally offer broad spatial coverage at high resolution, combined with millisecond-scale temporal precision, high light intensities and chronic implantability[3].

Optical stimulation of the cortex is commonly achieved using external light sources, such as laser scanning single-photon[4]/two-photon systems[5], or digital light projectors (DLPs)[6]. These systems are capable of achieving very high spatial resolution. However, the associated optical components and electronic control systems preclude chronic implantation, hindering translation to experiments on freely-behaving animals. Fibre bundles can deliver high-density spatio-temporal optical stimulation within a cortical region[7] with the possibility of chronic implantation and use in freely-behaving animals[8], but still require complex bench-top optical systems that preclude further development towards wireless implementation. For that, implantable optoelectronic solutions are needed. Micro-LED (light-emitting diode) ($\mu$LED) probes offer implantable, spatiotemporal optogenetic control over neuronal populations, with the possibility of very high stimulation site density[9–12] and fully-wireless options[13,14]. Chronic implantation of the optical source necessitates robust encapsulation strategies, addressing challenges such as biocompatibility, long-term stability[15] and thermal effects[16]. Two-dimensional arrays of $\mu$LEDs offer fully

[1]Institute of Photonics, Dept. of Physics, University of Strathclyde, Glasgow, UK. [2]Université Paris Cité, Institut Pasteur, AP-HP, INSERM, CNRS, Fondation Pour l'Audition, Institut de l'Audition, Paris, France. [3]These authors contributed equally: Ryan Greer, Antonin Verdier. [4]These authors jointly supervised this work: Brice Bathellier and Keith Mathieson. ✉e-mail: ryan.greer@strath.ac.uk; keith.mathieson@strath.ac.uk

implantable optogenetic stimulation coverage across the surface of the cortex. Previously, our group demonstrated the Utah optrode array (UOA), a µLED array with 181 sites, delivering light through penetrating optrodes[17] to deeper-lying structures of the non-human primate (NHP) cortex. Other groups have also demonstrated that this optoelectronic approach is viable. For example, Pollmann et al.[18] developed a µLED array coupled to a single-photon avalanche diode (SPAD) array capable of spatiotemporal optogenetic stimulation and fluorescence monitoring, with relatively large µLED dimensions and spacing, optimised for imaging performance requiring uniform illumination. Chronically implantable LED arrays were also demonstrated on the surface of the NHP cortex[19–21], where stimulation site spacing and device size limited the application to NHPs. The technology has been adapted for rodent studies, demonstrating site-specific activation in the cortex[22–24] with higher density devices being produced using organic LED (OLED) technology for in vitro studies[25]. The transfer of information to the mouse cortex requires high-density µLED arrays with spacing of a few hundred microns whilst covering significant tissue areas (up to 4 mm²)[6]. Previous µLED arrays match the stimulation site density and spacing of our device[26] and have been coupled to

arrays of electrodes (ECoG) integrating electrophysiology recording[27]. While both the density and scale sufficient for high-resolution cortical surface stimulation has been achieved, chronic implantation in mice and simultaneous demonstration of robust neuronal and behavioural responses remains an open challenge for the field (Supplementary Table 1).

Here we present a chronically implantable array of 100 µLEDs that can deliver high-intensity blue light over a 2 x 2 mm² footprint, enabling spatiotemporal optogenetic stimulation across the mouse auditory cortex. The µLED array is integrated into miniaturised, opto-electronic systems for in vivo experimentation in mice. Chronic implantation of the µLED array is enabled through integration to a flexible package (though the array itself is rigid), and can remain stable for up to 300 h experimental time. Electrophysiology recordings from the mouse auditory cortex demonstrate robust activation thresholds of 4 - 5 mW mm⁻² on the cortical surface. This corresponds to a drive current of ~ 5 mA, resulting in safe temperature changes predicted by our modelling for single-pixel illumination. Spatial resolution within cortical layer 2/3 was estimated at ~ 600 µm, which agreed with optical modelling. Lastly, chronically implanted µLED arrays providing

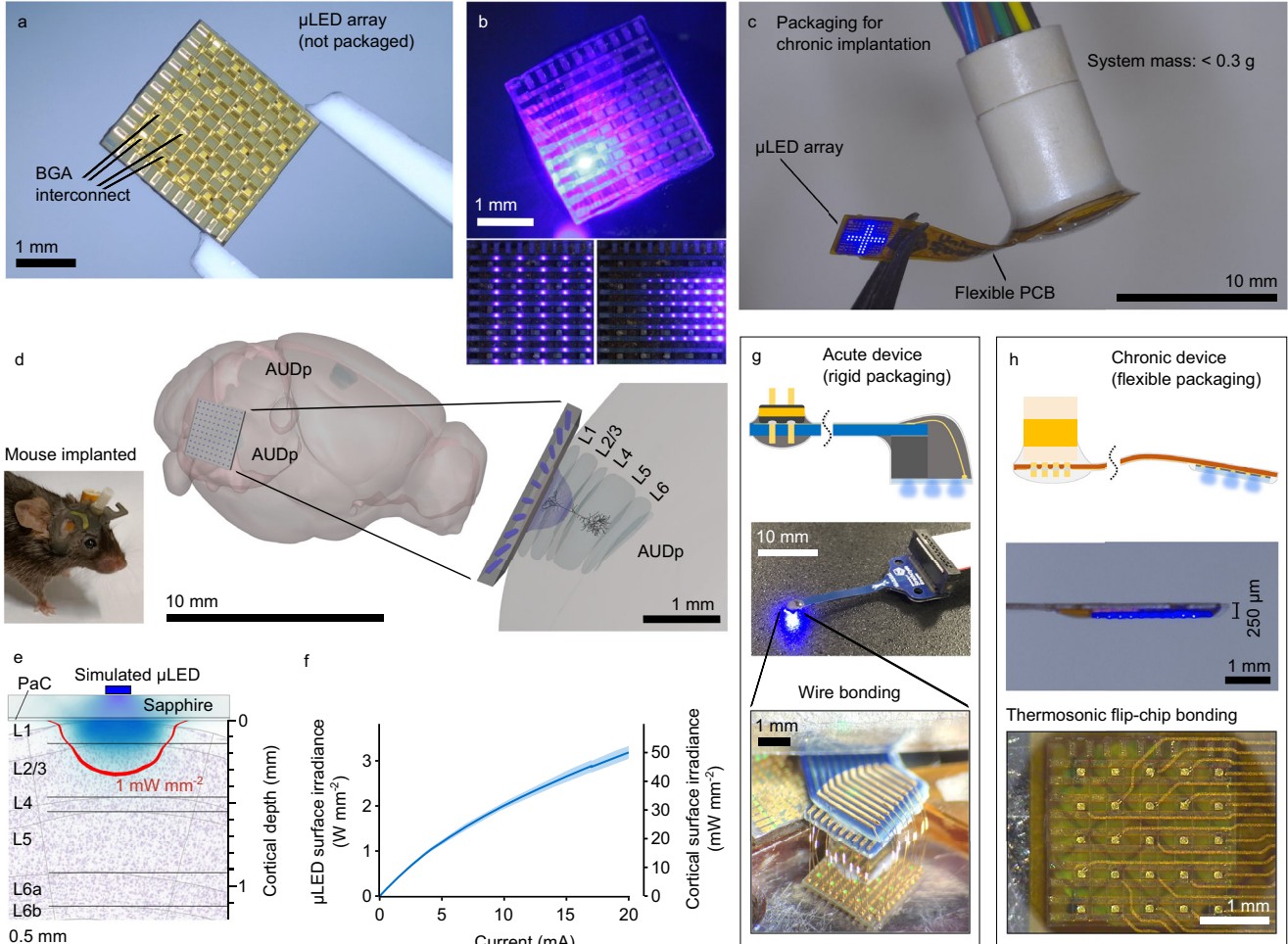

**Fig. 1 | Design and modelling of µLED array systems for optogenetic stimulation of the mouse cortical surface. a** Photograph of the fabricated side of unpackaged µLED array, highlighting ball-grade array (BGA) interconnection pads. **b** Sapphire (emitting) side of µLED array showing different illumination patterns and software-adjustable brightness. **c** Photograph of µLED array integrated to a flexible, lightweight package for chronic implantation. **d** Photograph of a mouse with a device chronically implanted over the auditory cortex (left). 3D model of µLED array over mouse auditory cortex (right); the 1 mW mm⁻² level is shown given a single µLED illuminating a cortical surface irradiance of 10 mW mm⁻²; illustration

produced using[50,51]. **e** Modelled optical profile of light propagation from a single illuminated µLED; irradiance levels are the same as (**d**). Model reflects chronic implantation with 150 µm sapphire and 15 µm parylene-C (PaC). **f** Irradiance-current (L-I) characteristic at the µLED surface and cortical surface (mean and standard deviation for n = 10 µLEDs). **g** Schematic and photograph of rigid device for acute experimentation; zoom shows gold ball-wedge wire bonds for interconnection and µLED addressing. **h** Schematic of chronically implantable device packaging and photograph of side-profile; bottom image shows the thermosonic flip-chip bonding process used for highly compact interconnection.

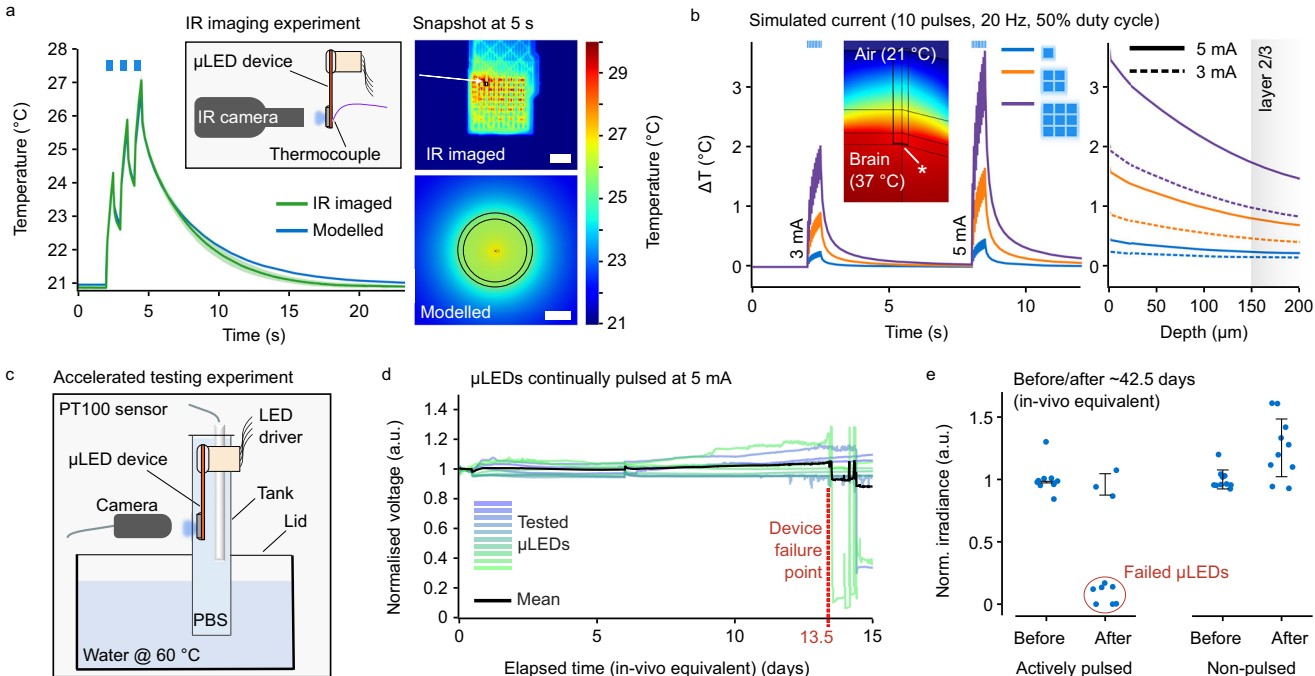

**Fig. 2 | Device evaluation demonstrates minimal tissue heating and longevity of chronic operation. a** Comparison of infrared (IR) imaging (mean and standard deviation for $n = 5$ repeated measurements) with equivalent thermal model (left); the inset shows a schematic of the IR measurement setup. IR image and screenshot of modelled surface with analysed regions highlighted (right); scale bars are 1 mm. **b** Modelled temperature increase on brain surface (indicated by asterisk) (left); modelled heat protocol is equivalent to that used in in vivo experiments, with simulated thermal power equivalent to a drive current of 3 mA and 5 mA per $\mu$LED. Peak temperature as a function of cortical depth (right). Dashed lines: 3 mA; solid lines: 5 mA. Line colour indicates the number of $\mu$LEDs simulated. **c** Schematic of accelerated ageing experiment showing the chronic device submerged in a tank of heated phosphate-buffered saline (PBS) solution with $\mu$LEDs being actively driven. **d** Normalised measured voltage during accelerated ageing for 5 mA drive current (each $\mu$LED was activated in a repeated sequence at ~1 min intervals for 10 pulses at 20 Hz, 50% duty cycle); $n = 10$ tested $\mu$LEDs for a single device encapsulated with 30 $\mu$m parylene-C and medical-grade silicone. **e** Optical measurements of actively tested and untested $\mu$LEDs before and after the accelerated ageing experiment for the same device as shown in (**d**). Error bars show the mean and standard deviation of $n = 10$ tested $\mu$LEDs, excluding the failed $\mu$LEDs.

spatially confined optogenetic stimulation drove robust learning outcomes in behavioural discrimination experiments, in both head-fixed and freely-behaving mice.

## Results

### $\mu$LED array design, fabrication and packaging

We developed a 100-site matrix-addressable array of GaN-on-sapphire $\mu$LEDs through microfabrication (Fig. 1a, b, Supplementary Fig. 1 and Supplementary Movie 1). The device features a dense array of $\mu$LEDs, arranged in a 10 × 10 grid with 200 $\mu$m spacing, enabling precise individual site or patterned illumination within a cortical region. Detailed fabrication steps are provided in the Methods; briefly, the $\mu$LEDs are 40 $\mu$m squares emitting blue light at a wavelength of approximately 450 nm, close to the peak absorption of ChR2 and a number of other common opsins. Emission is through the sapphire substrate, which was thinned to 150 $\mu$m. Substrate thinning is important because the $\mu$LEDs emit with a cosine dependence, making the distance from the source to the brain surface the dominant factor affecting spatial resolution.

The $\mu$LED arrays were packaged to a lightweight, flexible device (Fig. 1c), enabling chronic implantation in the mouse model (Fig. 1d). However, the stimulation area of the $\mu$LED array remains rigid, constrained by the GaN-on-sapphire fabrication process. Therefore, the device cannot achieve conformal contact with the brain surface as with flexible neural interfaces, however, the brain locally adapts to this constraint in the same way as it does as for cranial window implantations[28]. Histological analysis after three months chronic implantation, demonstrates plastic deformation around the implant site, with the size of the affected region comparable to the geometry of the $\mu$LED array (Supplementary Fig. 2). Immune response was then quantified by examining microglial build-up from the surface through the auditory cortex (Supplementary Fig. 2g). This was statistically significant when comparing the Iba-1+ cell response from the implanted side (228 ± 24 per mm²) to the non-implanted (contralateral) side (181 ± 18 per mm²). Excluding layer 1 from the analysis shows no significant difference between the implanted side (200 ± 34 per mm²) and contralateral side (133 ± 11 per mm²), highlighting a small localised immune response, approximately limited to layer 1 of the mouse cortex.

The determine whether light from the device passes beyond this depth, understand the resultant spatial resolution and estimate the required light intensity, we modelled the device using a ray tracing (Monte Carlo) simulation (Zemax OpticStudio 2021). Brain tissue was modelled using the scattering and absorption parameters for grey/white matter[29,30] and setting the refractive index to that of water ($n = 1.33$). This omits contributions from structures such as the vasculature. The device model was experimentally validated with in-air optical measurements (Supplementary Fig. 3), giving us confidence in the optical models to provide a good estimate of experimental conditions. Optical modelling indicates that at a 10 mW mm⁻² cortical surface irradiance, the 1 mW mm⁻² contour (approximate activation threshold for ChR2) penetrates through to the mouse cortical layer 2/3, assuming the device is implanted directly on the cortical surface (Fig. 1e). Optical outputs from the $\mu$LEDs were experimentally characterised at both the $\mu$LED surface and cortical surface, providing inputs to the modelling. For acute electrophysiology experiments, a 150 $\mu$m thin cortical window was used, which increases the emitter-cortex distance, therefore increasing the illuminated area on the

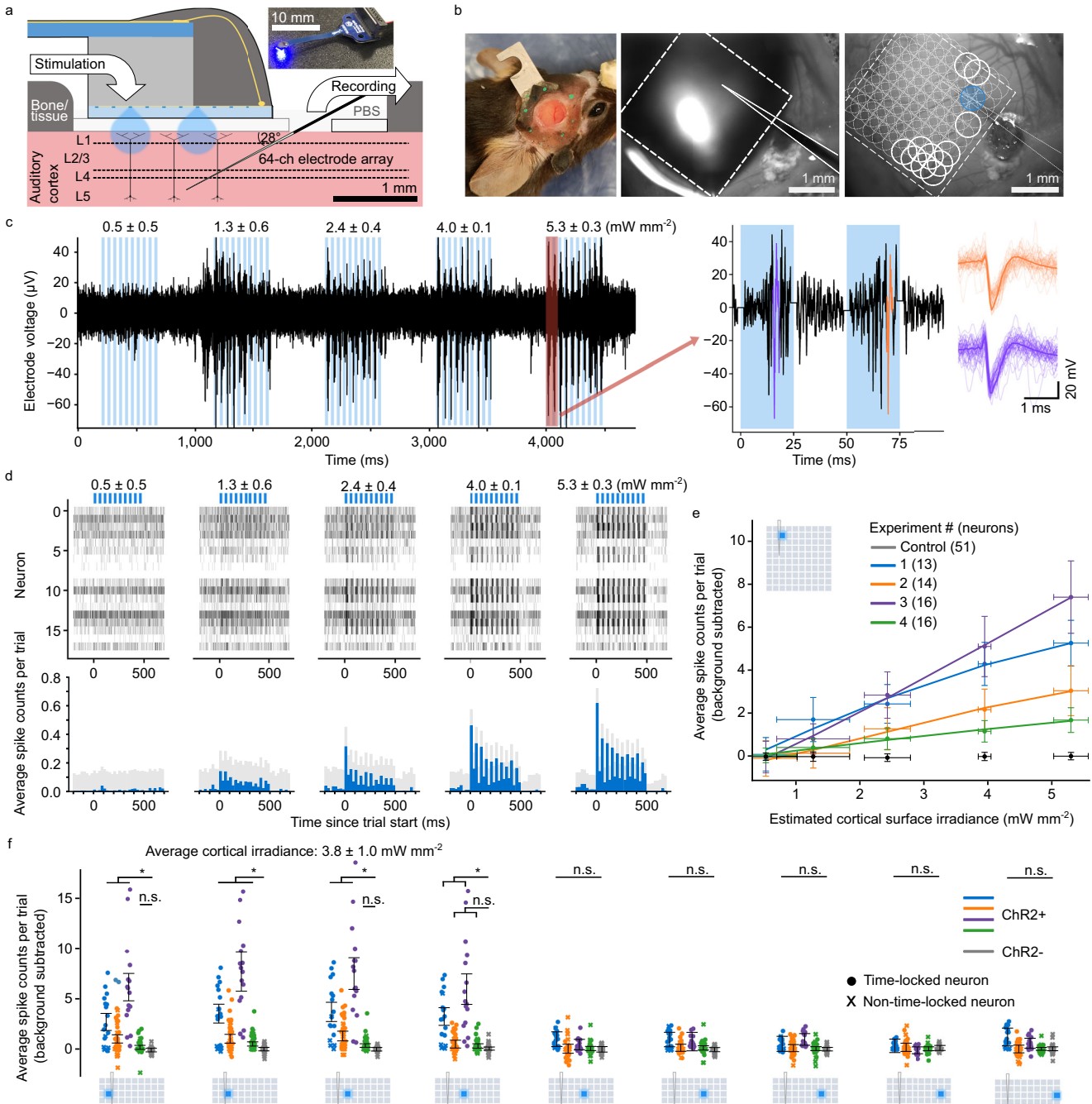

**Fig. 3 | Optogenetic stimulation by μLED array elicits reliable neuronal responses from mouse auditory cortex (AC) at low drive currents. a** Schematic of head-fixed in vivo experiment. Inset shows acute device used to place μLED array over the cortical window. **b** Mouse with cranial window implanted over right AC (left). Images of craniotomy with μLED array and recording probe (middle) and μLED layout superimposed (right). **c** Electrophysiology recording from a single electrode (filtered using parameters detailed in Methods). Each μLED is pulsed 10 times at 20 Hz, 50% duty cycle for 5 cortical irradiances (mean ± standard deviation) (μLED drive currents: 2.4, 4.0, 5.8, 8.0, 10.4 mA). Average waveforms of highlighted sorted spikes (right). **d** Raster plot of sorted spike times, accumulated across 50 trials (top). Peri-stimulus time histogram (PSTH) plot showing binned average spike counts per trial with the mean of background subtracted, averaged over all neurons. **e** Dose response curves for four mice plus a control, showing average spike counts per trial with background subtracted. Vertical error bars show pooled mean ± pooled SEM (ChR2+: $n = 13$, $n = 14$, $n = 16$, $n = 16$ time-locked neurons; ChR2-: $n = 51$ neurons); horizontal error bars show estimated cortical irradiance delivered by μLED (mean ± standard deviation). Panels (**c**–**e**) correspond to the same μLED. **f** Average spike counts per trial with background subtracted, under stimulation from different μLEDs. Error bars show pooled mean ± pooled SEM (ChR2+: $n = 18$, $n = 45$, $n = 16$, $n = 53$; ChR2-: $n = 51$ neurons); time-locked neurons are shown as circles, non-time-locked as x's. Significance levels determined using the independent two-sample, single-tailed t-test with alt. hypothesis: spike counts (ChR2+) > spike counts (ChR2-); *$p < 0.05$, n.s. means alt. hypothesis rejected at $p < 0.05$); exact $p$-values given in Supplementary Table 2. Average cortical irradiance given as pooled mean ± pooled standard deviation.

cortical surface, but reducing the irradiance. A cortical surface irradiance of 10 mW mm$^{-2}$ can be achieved with a drive current of less than 3 mA when the device is directly implanted (Fig. 1f).

In vivo electrophysiology experiments were carried out with the $\mu$LED array placed within a compact craniotomy in an acute preparation. This necessitated a miniaturised rigid package to lower the $\mu$LED array onto a cortical window and allow space for a silicon probe to be inserted (Fig. 1g and Supplementary Fig. 4a). Chronic implantation was then achieved by optimising this packaging (Fig. 1h and Supplementary Fig. 4b). Highly compact interconnection was achieved using thermosonic flip-chip bonding, reducing the thickness of the device tip to 250 $\mu$m and enabling chronic implantation, and the device to be flush with the mouse skull and surrounding tissue (Supplementary Fig. 12a). The $\mu$LED array design methodology enables the rapid development of custom packaging to facilitate a range of in vivo experimental paradigms, brain regions or different animal models. Furthermore, monolithic packaging of multiple $\mu$LED arrays could enable high-density information transfer within and between cortical regions in a single chronic implantation (Supplementary Fig. 5).

## Thermal evaluation and lifetime testing

Tissue heating is known to perturb neuronal activity, elevating or suppressing the firing activity of neurons[16,31]. This is particularly important when using implantable $\mu$LEDs for optogenetic stimulation because they are actively powered, dissipating most of this power as heat. Ideally, the temperature increase would remain within the commonly quoted limit of 1 - 2 °C[32]. Our specific device geometry and a custom material stack results in complex heat flows, necessitating an in-depth thermal evaluation[33]. Infrared (IR) imaging was used in conjunction with an equivalent 2D rotationally-symmetric mathematical model replicating the device with flexible packaging, chronically implanted with the sapphire surface in direct contact with cortical tissue. Appropriate material thermal properties were applied, with the brain approximated as water. To verify the accuracy of the modelled device, the sapphire surface of the $\mu$LED array was imaged using an IR camera in air (Supplementary Fig. 6), which agreed closely with the modelled output (Fig. 2a). These experiments, along with previous work demonstrating the accuracy of this model architecture with in vivo IR imaging[34], gave us confidence in our thermal models to provide a good approximation of experimental conditions.

The thermal model was then used to assess the temperature increase in vivo by simulating heat pulses equivalent to 3 mA and 5 mA per $\mu$LED (Fig. 2b and Supplementary Fig. 7). The peak temperature as a function of depth is shown (Fig. 2b right). For a single $\mu$LED driven at 5 mA, the modelled peak temperature increase remained well below 1 °C. For 2 × 2 and 3 × 3 patterns, the peak temperature increase can exceed 1 °C, thus the drive current must be selected carefully. As an example, driving a 3 × 3 pattern at 3 mA per $\mu$LED results in a peak temperature increase of ~2 °C on the brain surface and ~1 °C in layer 2/3. Despite being minimal, temperature increases stress the need for control experiments to rule out thermal effects influencing neuronal or behavioural responses. Thus, we carried out control experiments, both during electrophysiology and behavioural studies.

Accelerated ageing tests were conducted to ensure the devices would survive under the conditions of chronic implantation (Fig. 2c and Supplementary Fig. 8). Our encapsulation stack includes a thin-film oxide and SU-8 layer atop the $\mu$LEDs, as well as the sapphire substrate. Subsequent packaging steps include a polyimide PCB, medical-grade epoxy underfill and 15 $\mu$m parylene-C (PaC) conformal coating. For accelerated testing, an enhanced encapsulation strategy consisting of an additional 15 $\mu$m PaC coating and medical-grade silicone dip coating was evaluated. The devices were submerged in phosphate-buffered saline (PBS) solution at approximately 60 °C for two weeks. Heating of the solution allowed for an accelerated timescale to be evaluated[35]; in this case, the acceleration factor was ~4.10

$\mu$LEDs across the array were actively current-driven in a repeated sequence, and the voltage recorded. The $\mu$LEDs were also monitored optically to ensure the current was driving light output and not flowing through an alternative path.

The experiment duration was over 1000 h (42.5 days) of in vivo equivalent time. Under continuous operation, measured voltage remained stable across all $\mu$LEDs for up to 324 h (13.5 days) before the $\mu$LEDs subsequently failed (Fig. 2d). To quantify the failure mechanism, $\mu$LEDs from untested rows and columns of the array were evaluated, with all untested $\mu$LEDs remaining functional for the entire experiment duration (Fig. 2e). The encapsulation strategy is therefore stable when the device is passive, but actively driving the $\mu$LEDs shortens the lifetime of the devices, suggesting either higher electric fields or thermal effects degrade the encapsulation. Nevertheless, devices remain stable for over 300 h of continuous operation time. Since most in vivo experiments typically run for ~1–3 hours per day, we conservatively estimate that the devices should remain stable over months long chronic experimentation. Other encapsulation strategies were evaluated (Supplementary Fig. 8b), with 30 $\mu$m of PaC and silicone providing the best performance. Lastly, we evaluated devices which were chronically implanted for 29 days, actively used for behavioural experiments, and then explanted (Supplementary Fig. 8c). These devices showed a minimal degradation in performance, demonstrating the robustness and re-usability of the chronically implantable devices. Furthermore, implanted devices did not have the additional medical-grade silicone layer, which has been shown to be more effective than parylene-C alone in chronically implanted devices[15].

## Acute electrophysiology recordings reveal stimulation threshold

Electrophysiology recordings were conducted in the auditory cortex (AC) of awake, head-fixed mice under simultaneous stimulation from the $\mu$LED array, driven with our custom control system (Supplementary Fig. 9). Experiments were conducted in 4 transgenic mice expressing ChR2 in all excitatory neurons (Emx1-IRES-Cre x Ai27)[36,37] (denoted ChR2+). An additional control experiment was conducted in an Emx1-IRES-Cre transgenic mouse that did not express ChR2 (denoted ChR2-). We delivered high-intensity blue light to the AC to optogenetically stimulate neuronal populations within layer 2/3, allowing us to establish both a stimulation threshold and spatial resolution. Each mouse was prepared with a 5 mm diameter craniotomy with 150 $\mu$m transparent cortical window, optically exposing the AC. A $\mu$LED array, in the acute device configuration (Fig. 1g), was lowered onto the cortical window. A 64-channel linear electrode array was then inserted through a small hole in the cortical window (0.5 mm diameter) at an angle of 28° to the cortical surface, with recording sites extending from the cortical surface to a depth of approximately 680 $\mu$m, consistent with layer 5 (Fig. 3a, b).

Six recording sessions were conducted in total, five ChR2+ mice (one mouse was repeated) and a control mouse (ChR2-) to ensure detected neuronal responses were not thermally induced or stimulation artefacts. The linear electrode array was positioned at approximately the same location relative to the $\mu$LED array for all experiments (Supplementary Fig. 10a). Each experiment comprised a set of trials, each trial illuminating an individual $\mu$LED at a set drive current for 10 pulses (20 Hz, 50% duty cycle) with 500 ms wait between trials. This optogenetic protocol was chosen based on previous use of these parameters to drive strong neuronal activation[6,38]. The $\mu$LED location was varied in a pseudo-random sequence for each trial. Trials were repeated up to 50 times for each combination of $\mu$LED and cortical surface irradiance. For each $\mu$LED, the drive current was individually calibrated in software to account for variations in optical output due to device fabrication (shown in Supplementary Fig. 3c). In this section, the quoted standard deviations represent the variation in cortical irradiance delivered by each $\mu$LED at the drive currents used during the

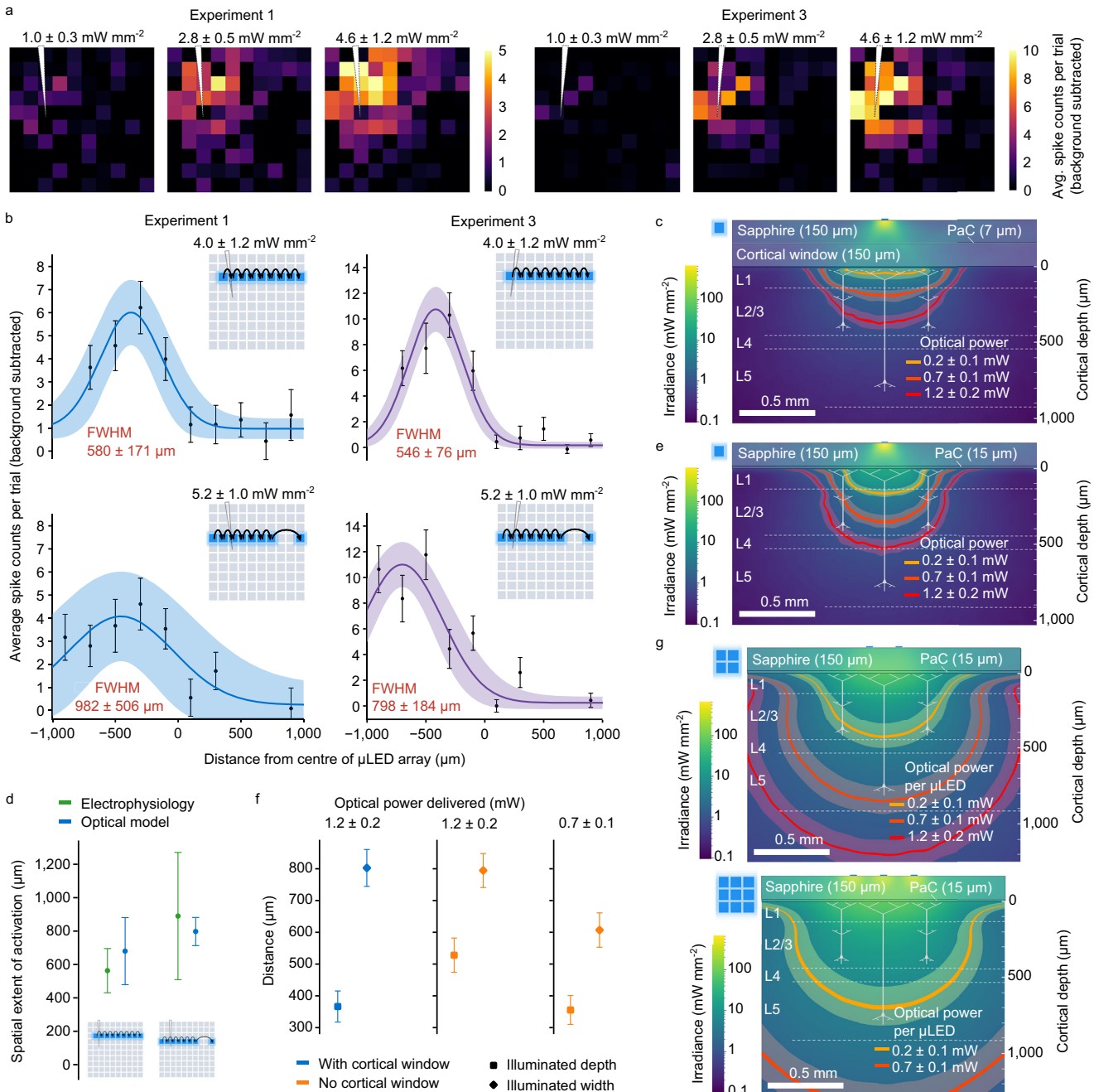

**Fig. 4 | Spatial resolution identified from neuronal responses supports optical modelling. a** Average spike counts per trial (background subtracted) for all tested μLEDs; mean of $n = 13$, $n = 16$ time-locked neurons; cortical surface irradiances given as pooled mean ± pooled standard deviation. **b** Neuronal response to μLEDs illuminated across individual rows, with Weighted Gaussian fits; error bars show pooled mean ± pooled SEM of average spike counts per trial ($n = 13$, $n = 16$ time-locked neurons). Cortical surface irradiances given as pooled mean ± pooled standard deviation for μLEDs across each row. **c** Cross-section from Monte Carlo simulation showing light propagation from μLED into brain tissue (acute device model with 7 μm parylene-C); contours show the 1 mW mm$^{-2}$ threshold for modelled cortical surface irradiances of 1.0 ± 0.3, 2.9 ± 0.5, 5.3 ± 0.7 mW mm$^{-2}$. Equivalent optical power delivered into the brain tissue was determined from

optical modelling. **d** Comparison of spatial resolutions determined from the FWHM of weighted Gaussian fits in (**b**) (pooled mean ± pooled SEM of $n = 29$ time-locked neurons) and optical model in (**c**) for the highest tested power; taken as the width of the 1 mW mm$^{-2}$ threshold contour at a depth of 150 μm. **e** Same as (**c**) but μLED array modelled directly on brain surface (chronic device model with 15 μm parylene-C); cortical surface irradiances of 4.0 ± 1.1, 11.3 ± 2.0, 20.4 ± 2.8 mW mm$^{-2}$ with corresponding optical power/drive current same as (**c**). **f** Modelled comparison of spatial resolution (diamonds) and depth of penetration (squares) of 1 mW mm$^{-2}$ contour with and without a cortical window. Error bars arise from shaded regions in (**c**) and (**e**). **g** Summation of Monte Carlo simulation results for multiple simultaneously illuminated μLEDs, delivering the same optical power/drive current per μLED as in (**c**) and (**e**).

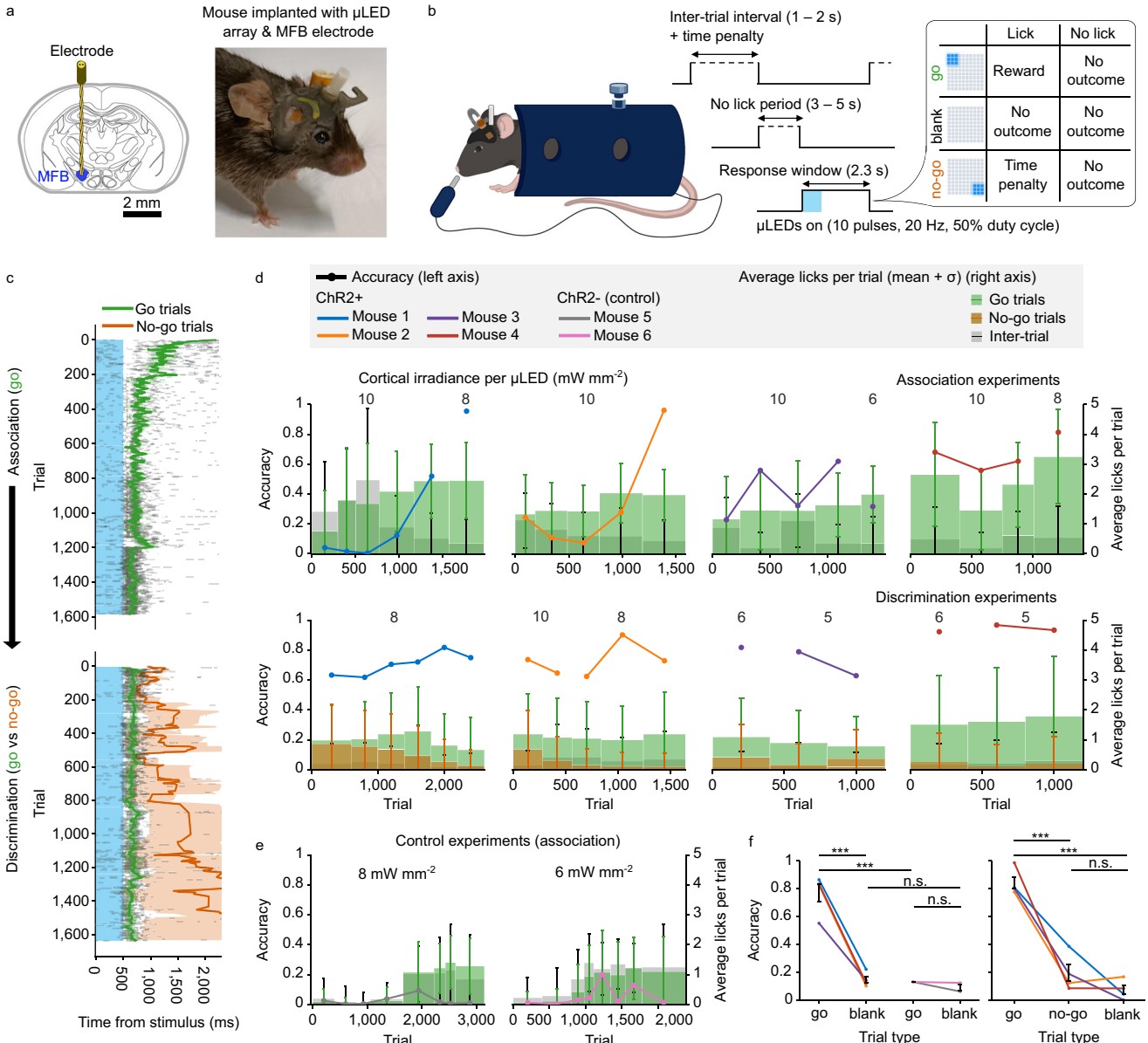

**Fig. 5 | Mice chronically implanted with μLED array reliably discriminate spatially confined illumination patterns on the auditory cortex (AC). a** Coronal schematic of the mouse brain showing the medial forebrain bundle (MFB) stimulator (left). Mouse chronically implanted with μLED array and MFB stimulator (right). **b** Schematic summary of behavioural experiment; each trial consists of a period where the mouse must not lick before the optogenetic stimulus is delivered. For go trials, the mouse should lick the detector, and for no-go trials, it should not; a small number of blank trials providing no optogenetic stimulus are included. **c** Example lick raster plots for Mouse 2 with mean and standard deviation shown for the lick times for the go and no-go trials. For all ChR2+ mice, the association experiment (go only) is followed by the discrimination experiment (go vs no-go). Control (ChR2-) mice only do the association experiment. **d** Learning curves for association (top) and discrimination (bottom) experiments. Each bar represents

the mean ± standard deviation of the lick counts per trial over a session/day, with bar colour representing trial type (right axis). The thick line shows the accuracy (% of trials correct) for each session (left axis). Cortical irradiance per μLED shown above corresponding session(s). Exact number of trials per session given in Supplementary Tables 3 and 4. **e** Same as (**d**) but for the association experiment with ChR2- mice. **f** Overall accuracy for each trial type (within the best 500 consecutive trial range); association experiment (left), discrimination experiment (right). Error bars show mean ± SEM of average accuracy (ChR2+: $n = 4$ mice, ChR2-: $n = 2$ mice). Significance levels determined using the independent two-sample, single-tailed $t$ test with alt. hypotheses: go accuracy > no-go or go accuracy > blank accuracy and/or ChR2+ accuracy > ChR2- accuracy. ***$p < 0.001$, n.s. means alt. hypothesis rejected at $p < 0.05$; exact $p$-values given in Supplementary Table 5.

experiments. The optical power of each μLED on the tested device was measured before and after the experiment, with this minor degradation in μLED performance assumed to be due to conductive solution ingress in the acute device.

The following analysis identifies a threshold cortical surface irradiance and μLED drive current needed to reliably elicit neuronal responses, allowing us to determine electronic driver specifications and examine thermal effects. An example of a recorded waveform is

shown (Fig. 3c) for 5 trials of stimulation by a single μLED, stepping through cortical surface irradiances; this μLED was chosen as it provided reasonably consistent irradiance and was located near the insertion point of the recording electrode array. Spike sorting was used to attribute the recorded electrophysiology signal to the neuronal firing activity of individual neurons (Fig. 3c right), allowing us to determine if neuronal responses were time-locked to the optogenetic stimulus and under what conditions. For the following analyses, only

single-unit activity is considered. Spike times are shown as a raster plot for all neurons (Fig. 3d top). These were summed in 25 ms bins and averaged over all neurons to yield a peri-stimulus time histogram (PSTH) (Fig. 3d bottom). The background was taken as the mean spike counts per bin in both 200 ms periods before the 10 pulses start and after the 10 pulses end. The PSTH shows the neuronal response is time-locked to the stimulus, and is particularly strong at cortical surface irradiance values exceeding 4 mW mm$^{-2}$. A dose response was plotted, showing average spike counts per trial with background subtracted (ChR2+: time-locked neurons, ChR2-: all neurons), against estimated cortical surface irradiance (Fig. 3e); we quote the cortical surface irradiance as an estimate due to the aforementioned reduction in optical performance, as shown by the horizontal error bars. Neurons were determined as time-locked if the average spike counts per trial during the on period were greater than $2\sigma$ from the average background spike counts. Here, threshold activation is defined as a 50% success rate - 1 spike per 2 pulses on average per neuron. This is equivalent to 1 spike per 50 ms time window, which remains consistent with the temporal kinetics of ChR2/H134R[39]. This spike rate was achieved at moderate cortical surface irradiances between $4.0 \pm 0.1$ mW mm$^{-2}$ and $5.3 \pm 0.3$ mW mm$^{-2}$, corresponding to a drive current of between 8.0 mA and 10.4 mA. These drive currents are slightly higher than expected due to the reduction in optical performance observed with this device. In stable devices, we demonstrate a consistent optical output of 5 mW mm$^{-2}$ at the cortical window/tissue interface for ~ 5 mA drive current (Supplementary Fig. 1d). The average spike counts per trial were plotted for different $\mu$LEDs illuminated at approximately the same threshold irradiance, across the width of the device (Fig. 3f). For ChR2+ mice, mean spike counts were higher for $\mu$LEDs located near the electrode array. The control mouse (ChR2-) showed no time-locked responses, indicating no confounding effects of temperature or experimental artefacts. Variation in the neuronal response between experiments could be due to variation in the relative positioning of the recording electrode array to the $\mu$LED stimulation site (Supplementary Fig. 10a), the flatness of the $\mu$LED array placement on the cortical window, or the overall electrode array insertion quality and surrounding tissue health. Improved consistency between experiments is possible with the removal of the cortical window and with our chronically implantable devices, where the supporting polyimide PCB is flexible, making it easier to fix into position.

## Spatial resolution

Simultaneous electrophysiology recordings under stimulation from individual $\mu$LEDs allowed examination of the spatial resolution of the device. Spatial maps were plotted showing average spike counts per trial with background subtracted for all tested $\mu$LEDs (Fig. 4a), plotted for the two experiments with the strongest responses; Experiment 1 and Experiment 3. Weighted Gaussian curves were fitted to rows 3 and 4 of the map corresponding to the highest cortical surface irradiance (Fig. 4b). The spatial resolution was inferred from the full width at half maximum (FWHM) of these fits. Due to the use of a single shank recording electrode device with a fixed position, spatial resolution estimates could only be obtained from $\mu$LEDs located in rows near the edge where the electrode array was inserted.

Optical models were developed and compared with spatial resolution estimates obtained through electrophysiology. The 1 mW mm$^{-2}$ contour (approximate threshold for ChR2) was plotted, given the cortical surface irradiances evaluated during the in vivo experiments (Fig. 4c); the shaded error is not indicative of the model, but is based on the variance in experimental cortical irradiance levels. To make a comparison of the experimental spatial resolution (Fig. 4b) and the optical model (Fig. 4c), we take a cut of the model at a depth of 150 $\mu$m (consistent with layer 2/3, which is the target neuronal layer) and use the width of the 1 mW mm$^{-2}$ threshold contour as an estimate of the width of activation. Experimental and modelled estimations of the

width of activation are in reasonable agreement (Fig. 4d). This comparison is an approximation, ignoring dendritic contributions and second-order network effects, and implies that the optical models could be used to inform device optimisation to further improve spatial resolution. These estimates assume the highest tested cortical irradiance, however, responses can be seen at lower irradiances and would result in a smaller illuminated volume and width. This comparison is also based on the cortical window between the sapphire surface and the brain used in electrophysiology experiments. This exacerbates the spread of light caused by the Lambertian emission profile from the $\mu$LEDs and can be removed for chronic implantation.

The 1 mW mm$^{-2}$ contour was plotted with the $\mu$LED array modelled directly in contact with the brain tissue (Fig. 4e). In both simulations, shown in Fig. 4c, e, the same optical power was delivered to the brain (hence requiring the same drive current); although the cortical surface irradiance without the window is higher because the illuminated spot size is smaller (Supplementary Fig. 1d). Both the depth and width of the 1 mW mm$^{-2}$ threshold contour were compared in each case (Fig. 4f). For equivalent optical power (1.2 mW), removal of the cortical window allows deeper penetration of light with the same lateral width (Fig. 4f middle column), while reducing the optical power to 0.7 mW results in a reduction in light spread with the same depth of penetration (Fig. 4f right column). Different patterns ($2 \times 2$ and $3 \times 3$) were also modelled based on the same delivered optical power per $\mu$LED (Fig. 4g). The activated volume significantly increases in both cases, demonstrating the ability of the $\mu$LED array to activate an entire cortical region if necessary, at modest drive currents.

## Behavioural experiments with chronically implanted devices

Chronic implantation of $\mu$LED arrays can facilitate repeated optogenetic stimulation of distinct ensembles of neurons from a cortical region, over days or weeks. To demonstrate this, we trained mice to respond to artificial perceptions generated by illuminating spatially confined patterns on the surface of the auditory cortex (AC). To motivate the mice to learn the task, we delivered bi-phasic electrical charge through intra-cranial stimulation of the medial forebrain bundle (MFB). This reliably activates reward centres in the brain, and the animals do not become sated as with water or food rewards[40]. Mice were first implanted with the MFB stimulator electrode, then a square-shaped craniotomy was performed to place the $\mu$LED array in direct contact with the cortical tissue after careful removal of the dura mater, with dental cement used to secure the device to the skull (Fig. 5a and Supplementary Fig. 12a). After a 5-day recovery period, implanted mice were head-fixed and placed in a contention tube, with a lick detector within reach (Fig. 5b and Supplementary Fig. 12). Between behavioural experimental sessions, mice were able to remain non-encumbered due to the lightweight device packaging, however, they were sometimes housed individually to avoid damage to the connector assembly.

Experiments were carried out over a number of sessions (1 session per day), each consisting of several hundred trials usually over a 1-2 hour learning period, dependent on mouse performance (Supplementary Tables 3 and 4). On each trial, the $\mu$LED array delivers a 10-pulse burst (20 Hz, 50% duty cycle) of illumination at between 5–10 mW mm$^{-2}$ per $\mu$LED to the surface of the AC. Again, this optogenetic protocol was selected to drive consistent neuronal activation, aiming to achieve strong salience for the animal. Although the device facilitates individual pixel operation, a $3 \times 3$ square of $\mu$LEDs was used to illuminate one corner of the diagonal depending on the trial type (go or no-go). This achieved a broad area of illumination and was based on previous experiments using a digital light projector (DLP) system to illuminate 400 $\mu$m diameter regions of the AC, to drive behavioural outcomes in ChR2-expressing mice[6,41]. The response window opens for 2.3 seconds on the onset of the illumination, during which the mouse has the opportunity to receive a reward of MFB stimulation if it licks in response to the go stimulus. If the mouse licks in response to the no-go

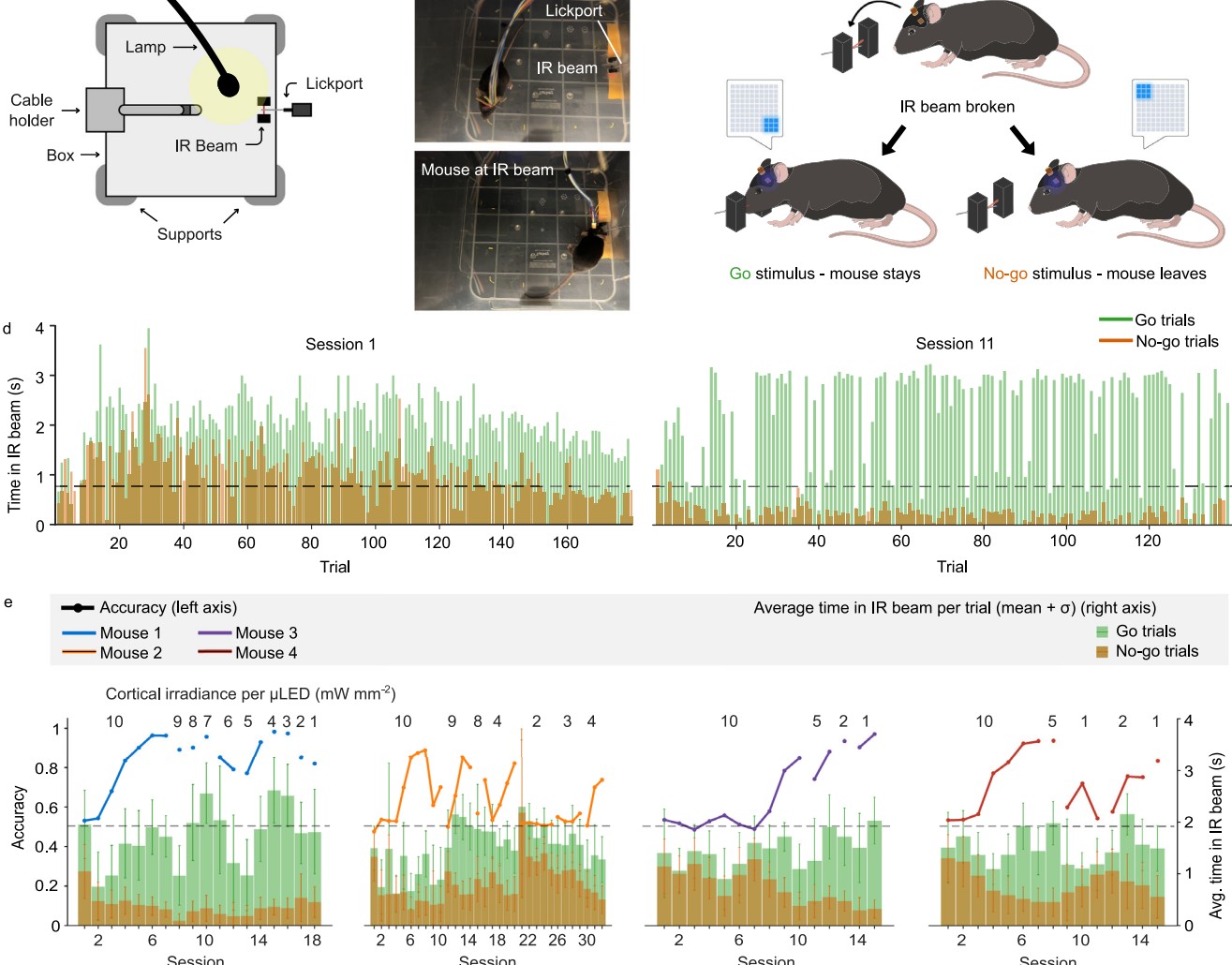

**Fig. 6 | Freely-behaving mice achieve similar discrimination performance to optogenetic stimulation with µLED arrays as head-fixed. a** Schematic diagram of the setup used for freely-behaving experiments. **b** Photographs within the chamber showing the mouse moving to the infrared (IR) beam to begin the task. **c** Schematic illustration of freely-behaving discrimination task. **d** Cumulative time spent in the IR beam for each trial in the first session (left) and in the eleventh session (right) for Mouse 1. **e** Learning curves for each mouse. Each bar represents the average cumulative time spent in the IR beam per trial over a session/day (mean ± standard deviation), with bar colour representing trial type (right axis). The thick line shows the accuracy (% of trials correct) for each session (left axis) with the chance level indicated by the dashed line. Cortical irradiance per µLED shown above corresponding session(s). The exact number of trials per session is given in Supplementary Data 1.

stimulus, a time penalty is incurred, which is added to the random inter-trial interval to delay the beginning of the next trial. All other responses result in no outcome, including those during the blank trial, where no illumination is presented. Prior to the illumination starting and response window opening, the mouse must not lick the detector for a set amount of time, to avoid continuous licking to receive the reward without learning the task; an experimental trial is illustrated in Fig. 5b. The mice are trained on two distinct tasks - the association experiment only presents go trials and the discrimination experiment presents both go and no-go trials; in both cases there are a small number of blank trials where no µLEDs are illuminated, to indicate background licking activity. Initially, the mouse is trained to lick in response to the go stimulus in the association experiment. After learning the association task, defined as maintaining consistent accuracy above 80% (typically after 4–6 sessions), the mouse moves onto the discrimination experiment, where the no-go stimulus is introduced. In the discrimination experiment, the mouse should continue licking in response to the go stimulus while ignoring the no-go stimulus. Figure 5c shows a lick raster plot for both association and

discrimination experiments, with the mean and standard deviation given for the go and no-go licks within the response window; these are shown for the same animal (Mouse 2), which appears to learn the task well.

Learning curves were plotted to show how the accuracy evolves over the duration of the experiments (Fig. 5d). The accuracy was computed for each trial from the percentage of go trials where the mouse licked (for the association experiment) or the combined percentage of go trials where the mouse licked and no-go trials where the mouse did not lick (for the discrimination experiment). A trial is only counted towards the accuracy if the number of licks within the response window is greater than $2\sigma$ from the mean of the background lick rate, determined from the average lick counts per session from the inter-trial interval, as shown by the grey bars. Since the mice appear to stop licking once they receive a reward, this background lick rate is not necessarily present over the course of the response window, and may provide an underestimate of the accuracy. Mice consistently learned the association task, with the exception of Mouse 3 demonstrating more variability in performance between sessions. Good accuracy is

maintained during the discrimination experiment, demonstrating adaptive behaviour based on the same intensity but different spatial location of the illumination pattern. Lick raster plots have been generated for each trial type and all experiments, which demonstrate elevated licking timed with the onset of the go optogenetic stimulus (Supplementary Fig. 13). Two control mice (denoted ChR2-) received the same implantation of MFB stimulator and $\mu$LED array and were trained only on the association task (Fig. 5e). In both cases, onset of licking to receive a reward was delayed until the MFB voltage was increased to encourage some licking (at ~1000 trials in each case). Mice began to lick continually, but this was not elevated above background licking during the response window, as with the ChR2-expressing mice. Mouse 5 did appear to lick timed to the illumination during the final session, which can be seen in the lick raster plot in Supplementary Fig. 13b, although this was not statistically significant when compared with the background lick rate. This delayed learning could be the result of the mouse visually perceiving the blue light. In future experiments, the transparent area above the $\mu$LED array could be covered, or a masking light used to saturate the mouses' vision. The overall accuracy for each trial type is summarised for the 500 consecutive trials with the highest accuracy for each experiment (Fig. 5f). On average, mice were able to learn both tasks, achieving an accuracy of 80% at cortical surface irradiance levels similar to the thresholds determined from electrophysiology experimentation (5–10 mW mm$^{-2}$) but over an increased illuminated area and volume as was shown by the optical modelling in Fig. 4g. Consistent behavioural responses were observed as low as 5 mW mm$^{-2}$, which can be driven at a per-$\mu$LED current of less than 3 mA (Supplementary Fig. 1d). Thermal analyses suggest this current level is safe, even for up to nine $\mu$LEDs illuminated simultaneously (Fig. 2b and Supplementary Fig. 7j).

### Behavioural discrimination experiments in freely-moving mice

Freely-behaving mice implanted chronically with $\mu$LED arrays over the auditory cortex conducted a similar behavioural discrimination task; this time using water deprivation to motivate learning. A flexible cable was mounted above a behavioural chamber to allow the mice to conduct the experiment while moving freely. The box was equipped with an infrared (IR) beam sensor to detect mouse presence, and a lickport to deliver the reward (Fig. 6a, b, Supplementary Fig. 14 and Supplementary Movie 2). For these experiments, we expressed the opsin ChRmine, which has a much larger photocurrent than ChR2 and a broad excitation spectrum, resulting in ~10x sensitivity at 450 nm[42]. Control (opsin-negative) mice were not included as the stimulation protocol used here (Fig. 6c) was identical to the head-fixed behavioural experiments - these already confirmed successful optogenetic activation and no confounding effects influencing behaviour. Per-$\mu$LED cortical irradiance was set at 10 mW mm$^{-2}$ and progressively reduced to 1 mW mm$^{-2}$ in the cases where the behavioural outcome accuracy was maintained at a strong level (typically > 80%) across sessions. Mice performed daily experimental sessions, each containing several hundred trials (Supplementary Data 1). Each trial begins when the mouse voluntarily breaks the IR beam, triggering optogenetic stimulus presentation and beginning a 1 s response window. During go trials, mice needed to remain in the IR beam for a cumulative time of at least 75% of the response window to receive a reward. For no-go trials, correct performance required remaining in the IR beam for less than 75% of the response window to avoid a time penalty. Between trials, mice had to remain outside the IR beam for a set refractory period, which encouraged movement away from the reward port.

During initial sessions, mice showed no difference in the average time spent in the IR beam between the two stimuli types Fig. 6d left), with performance remaining at chance level (~50%) (Fig. 6e). However, in subsequent sessions the accuracy (computed as the proportion of trials correct) improved, with consistent accuracy at or above 80% indicating the task was learned (Fig. 6d right). The number of sessions

required for learning varied between subjects, with Mouse 1 requiring 5 sessions to reach and maintain 80% accuracy compared to 7 sessions for Mouse 2, although performance was less consistent for the latter (Fig. 6e). Once discrimination learning was established, we observed greater consistency in the time spent in the beam during go trials and a marked decrease during the no-go trials (Fig. 6d right). After achieving stable task performance, we explored progressively reducing cortical irradiance per $\mu$LED and noted discrimination performance at irradiance levels that tended to be lower than those tested in the head-fixed experiment (5 mW mm$^{-2}$), presumably due to the use of the more sensitive ChRmine opsin. All mice were able to perform the discrimination task above chance level, for irradiances as low as 1 mW mm$^{-2}$, except for Mouse 2, which failed to discriminate when the cortical irradiance was reduced to 2 mW mm$^{-2}$ (50% chance level accuracy). However, upon increasing to 4 mW mm$^{-2}$, discrimination performance increased again (Fig. 6e). Detailed analysis of all animals is provided in Supplementary Fig. 15.

## Discussion

We presented an array of 100 matrix-addressable GaN-on-sapphire $\mu$LEDs for spatiotemporal optogenetic stimulation of the mouse cortical surface. The $\mu$LEDs are fabricated on a thinned sapphire substrate, therefore maintaining stress-matching to the $\mu$LED quantum well structures, resulting in high-intensity sources capable of driving robust optogenetic responses in the mouse cortex from a single site or spatiotemporal patterns. The trade-off is a reduction in spatial resolution due to the 150 $\mu$m standoff between source and brain surface. In the future, this could be reduced by using top-emitting devices or removing the LED structures from their substrates, either through laser lift-off processes or through selective chemical etching[43]. The latter offers fully flexible interfaces and can both deliver reduced source-tissue proximity and similar stimulation site spacing[26,27]. However, device quantum efficiency is compromised, and further in vivo validation of these approaches need to confirm the $\mu$LEDs can achieve sufficient intensity for behavioural outcomes as well as long-term, chronic stability. Our $\mu$LED arrays are able to drive above-threshold irradiances several hundreds of microns deep and with broad spatial coverage over a cortical region, although thermal implications must be considered when activating a large number of pixels simultaneously. Reliable neuronal responses and behavioural outcomes were achieved while maintaining low temperature increases at the implanted site. Furthermore, flexible, biocompatible packaging enabled chronic implantation of the $\mu$LED arrays in the mouse model. Although the $\mu$LED arrays are rigid, scalability over a larger surface could be achieved through integration of several $\mu$LED arrays tiled on a single device (Supplementary Fig. 5). This could facilitate experimentation across brain regions or in larger animal models such as non-human primates. This potential for rapid scalability and device customisation, along with robustly established stimulation parameters and in vivo stability, makes this an ideal platform for chronic studies into cortical function and the targeted transfer of information to the brain.

Activation thresholds were determined through electrophysiology recordings from the auditory cortex of awake mice, with less than 5 mW mm$^{-2}$ on the cortical surface driving robust neuronal responses. This corresponds to a drive current of ~ 5 mA (Supplementary Fig. 1d), resulting in safe temperature increases according to thermal models for single-pixel illumination (Fig. 2b). Control experiments, in non-opsin-expressing mice, were conducted throughout to ensure measured responses, either neuronal or behavioural, were not thermally-induced. Low device operating power further opens the prospect of wireless, battery-powered operation[13,14].

Spatial resolution was examined by comparing neuronal responses to optical models of light propagation in tissue, indicating a spatial spread of ~ 800 $\mu$m. Direct implantation of the $\mu$LED array on the cortical surface can improve spatial resolution by ~ 200 $\mu$m, to an

illuminated width of ~ 600 μm, in addition to reducing the required drive current (Fig. 4f). Using our optical models, we investigated if the spatial resolution could be improved through light-shaping. We modelled and experimentally validated an optical interposer, which collimates the light using small vias fabricated into an otherwise opaque material (Supplementary Fig. 16). Incorporating this interposer could reduce the illuminated width to ~ 500 μm. The trade-off is optical losses in the silicon, resulting in less light penetrating the tissue. The optical losses can be kept to a minimum with a 50 μm thick interposer and 200 μm diameter optical vias, which can deliver an equivalent optical power to the brain if the μLED output power is increased by a factor of approximately 4-5. This is possible given the low-power operation of our μLEDs; however, less power could also be used to target shallower depths and improve the spatial resolution further. The slight improvement in spatial resolution indicates that the dominant contribution arises from light scattering in tissue. To confirm this, we modelled a perfectly collimated light source (as can be implemented in optical systems, such a digital light projectors - DLPs) incident on a 200 μm diameter area of brain tissue (Supplementary Fig. 16d). The illuminated volume and resolution is similar to the interposer models, demonstrating light scattering is indeed the dominant factor limiting any non-penetrating, optical cortical surface stimulation technology.

The ability to manipulate distinct neuronal ensembles at the intracortical scale, while observing behavioural outputs, is key to exploring the circuit dynamics underlying perception. Chronically implanted mice were trained to respond to causal artificial perceptions generated by spatially confined optogenetic stimulation of a distinct sub-region of the auditory cortex, while not responding to an identical, but spatially separated, stimulus. Consistent and robust discrimination performance was observed across all ChR2-expressing animals at comparable cortical irradiance levels as determined through our electrophysiology experiments, as well as in a previous study using a digital light projector system[6]. Chronic implantation of the μLED arrays allowed repeated optogenetic stimulation of the same neuronal ensembles over many days (50–70 days following post-surgery recovery), as well as experiments in freely-behaving animals, with a relatively straightforward surgical procedure enabling implantations across many animals (Supplementary Table 6). Accelerated ageing experiments confirmed device stability for > 300 h of continuous operation time, suitable for months-long chronic experimentation if necessary (Fig. 2d). Furthermore, chronically implanted and tested devices were shown to remain operational after explantation, offering the potential of device re-usability (Supplementary Fig. 8c). Histological analyses from mice implanted for three months showed an increase in immune response on the brain surface, and plastic deformation around the implant site (Supplementary Fig. 2). This is expected given the μLED array is rigid, demonstrating similar effects to standard cranial window implantations[28]. Nevertheless, behavioural performance was maintained at a high level, highlighting the stability of the cortical network over the duration of the chronic implantation. Future studies could include reducing the illuminated area or the lateral separation of the go and no-go illuminated patterns to further explore the possibilities of optogenetic cortical implants for the targeted transfer of information to the brain.

The Emx1-IRES-Cre x Ai27 strain of mice were used across in vivo experiments. These mice express ChR2 in all excitatory neurons, most of which are pyramidal neurons and also express the opsin in their dendrites[36,37]. Freely-behaving experiments instead used AAV vectors encoding ChRmine under the control of the CaMKIIα promoter, similarly targeting pyramidal neurons, and providing strong photocurrent for a broad excitation spectrum, including the wavelength of our μLEDs[42]. We cannot say whether neuronal responses resulted from direct somatic activation, summed dendritic activation or broader network effects, or a combination of these. To gain

further insight, we conducted a multi-unit analysis (MUA) using a thresholding approach, providing a neuronal response as a function of recording depth (Supplementary Fig. 17). These results show robust responses at the tip of the electrode array in layer 5. Optical modelling suggests the light did not penetrate this deep, even at the highest tested power (Fig. 4c), indicating we may be recording responses from layer 5 pyramidal neurons being driven through activation of their dendrites in layer 2/3, or by other second-order effects. Spatial resolution appears to widen at deeper recording sites (Supplementary Fig. 17e), further suggesting integration of dendritic responses contributing to deeper neuronal responses. Soma-expressing opsin variants could be employed in future studies to gain a clearer understanding of this stimulation paradigm[44]. This study was limited to spatial patterning with fixed timing (20 Hz, 50% duty cycle), selected to drive strong neuronal activation and behavioural salience. This protocol was chosen as a proof-of-concept for behavioural experiments, but is not necessarily optimal for optogenetic control. Future studies could include optimisation of the optogenetic stimulation protocol, and more sophisticated temporal patterning of the μLED array could also allow exploration of the involvement of temporal features in the neural code of the cortex. Pairing the excitation of genetically targeted neurons with hybrid electrical stimulation could offer further insight into the specificity of the stimulation. For example, high-frequency electrical stimulation is known to recruit inhibitory neuronal networks[45]. A hybrid stimulation approach could therefore offer insights into how best to control neuronal activity, with implications in clinical devices. Transformative experiments in humans have restored lost sensation through intracortical electrical stimulation[2,46]. However, stimulated patterns do not necessarily translate to the expected perception, highlighting the need for a deeper understanding of the spatiotemporal code[38,47]. This is likely to require cell-type specificity, emphasising the opportunity for optogenetic cortical stimulators, such as the one detailed here.

## Methods
### μLED array fabrication
The μLED arrays were fabricated on a commercial InGaN on patterned sapphire wafer before being diced, individualised, and released (Supplementary Fig. 1). The sapphire substrate is thinned to 150 μm before processing; after thinning, the wafer is optically polished (Valley Design Corp.). A palladium layer (Pd) – 100 nm – is deposited using an E-beam evaporator (Edwards 306), to define the 40 μm pixel shape and also act as a spreading layer. A thin silicon dioxide (SiO2) layer patterned by a photoresist (Oxford Plasmalab 80 Plus, Electronic Micro Systems 4000, Suss Microtec mjb4) is used as an etching mask to etch the Pd layer and p-GaN layer, forming the pixel structure. Another SiO2 layer is then used to mask the n-GaN etching, which forms the pads for wire bonding or flip-chip bonding as well as the n-type mesa, with 10 rows, each with 10 pixels per array, forming the 10 x 10 array. A lift-off process using a bi-layer of photoresist deposits a layer of Ti:Au (100 nm:300 nm) along the n-type mesa, the bond pads as well as a contact on top of each p-type pixel. Two insulation layers are formed over the n-metal tracks. Firstly, 800 nm of SiO2 is deposited and vias etched open using an RIE tool (Oxford Plasmalab 80 Plus). A layer of SU-8 TF 6005 photoresist is spun and patterned then hard baked at a temperature of 200 °C for 2 h. A lift-off process is then repeated on top of these insulation layers, depositing Ti:Au (100 nm:300 nm) in columns forming p-contacts to the pixels, as well as further depositing metal on the wire bonding and flip-chip bonding pads. A final layer of SU-8 TF 6005 is spun on top of these tracks, vias opened above the bond pads and hard baked. A protective layer of photoresist (SPR4.5) is spun on top of the wafer, and individual arrays are released using a diamond saw dicing tool (Disco Hi-Tech DAD3320).

## µLED array packaging

The µLED array has 20 pads for interconnection, connecting to the 10 sets of anodes or cathodes, and allowing all 100 µLEDs to be matrix-addressed. There is an option to interconnect through wire bonding pads (on two edges of the device) or a ball-grade array (BGA), for flip-chip bonding.

For acute, head-fixed experimentation in the mouse model, the µLED arrays were packaged into a custom printed circuit board (PCB) system (Supplementary Fig. 4a). The acute system consists of two PCBs (0.2 mm thin) approximately 3 cm in length which are glued together, with the µLED array attached to a silicon spacer block (1.25 x 1.25 x 1 mm³) to allow placement in the cranial recess avoiding surrounding bone and tissue while remaining flat on the cortical window (Fig. 3a). Epoxy adhesive paste (MG Chemicals 9460TC) is used to glue the components together. Placing and gluing of components is achieved using a flip-chip bonder system (Finetech Pico ma) and programmable adhesive dispenser (Metcal DX-350). A 1.27 mm pitch 20-pin rectangular connector is soldered to both PCBs, allowing electrical connection to the 10 p and n tracks. The µLED array is electrically connected to 75 nm thick ENIG-coated (electroless nickel immersion gold) tracks on each PCB through 25 µm diameter gold ball-wedge wire bonds (Micro Point Pro i5000 Dual). The wire bonds are then potted, and the PCB shanks glued together (MG Chemicals 9510) (Fig. 1g). The system is coated with 7 µm of parylene-C (SCS PDS2010).

The µLED array was integrated to a flexible system for chronic implantation and used in behavioural experiments (Supplementary Fig. 4b). The shank is 100 µm thin and 2 mm wide, breaking out 20 connections and allowing every µLED to be individually addressed. The system was developed by bumping 75 µm diameter gold balls onto a commercial flexible polyimide PCB (PCBway) and attaching the µLED array through the BGA pads using thermosonic flip-chip bonding. A medical-grade epoxy (EPOTEK MED-301-2) is then used to underfill the µLED array for electrical and mechanical protection. A miniature circular Omnetics connector is soldered at the opposite end, which is potted using the same epoxy. The system is coated with 15 µm of parylene-C (PaC). We further explored the encapsulation procedure in accelerated tests only by adding an additional layer of PaC (15 µm) as well as a medical-grade silicone dip coating (NuSil MED-2214). Supplementary Fig. 4c shows the steps involved in applying the silicone dip coating. For our devices, a silicone primer (NuSil MED-161) was firstly applied using a small bristle brush and allowed to dry for one hour. A small piece (3 x 3 mm²) of UV-sensitive dicing tape was attached to the sapphire surface of the device. The device was fully submerged in MED-2214 which had been decanted into a beaker. While submerged, the beaker was rotated slowly several times with the device held at different angles, before carefully lifting the device out and leaving it suspended on a drying rack for 10 min at room temperature until the silicone partially cured. A second coating was applied in the same manner as before and the device suspended on a drying rack, this time above a hotplate set to 50 °C, for one hour. The dicing tape was exposed to UV light and peeled off, so the sapphire surface was clear, but the rest of the device was encapsulated; this was necessary to minimise standoff distance between the µLEDs and the brain surface. A final cure was carried out at 150 °C until the silicone had fully cured.

## Optical modelling and characterisation

The µLEDs were optically modelled using a custom ray tracing (Monte Carlo) model and simulation (Zemax OpticStudio 2021 in non-sequential mode). The model consists of a single 40 µm square GaN µLED containing a light source acting as the QW (quantum well) structure (Supplementary Fig. 3a). The µLED has a gold contact on the back, which is modelled as a mirror. The model includes a 2 mm square sapphire substrate (150 µm), parylene-C layer (7 µm for acute device with cortical window or 15 µm for chronic device directly on brain tissue), optional cortical window (150 µm) and auditory cortex. The auditory cortex was modelled as grey matter, with a 150 µm layer of white matter on the surface. The refractive index of water was used as it approximately matches the reported values for brain tissue[48], while incorporating suitable scattering coefficients to approximate the optical characteristics of the brain; white matter: scattering coefficient, $\sigma_s = 50$ mm⁻¹, absorption coefficient, $\sigma_a = 0.14$ mm⁻¹, anisotropy index, $g = 0.78$; grey matter: $\sigma_s = 11$ mm⁻¹, $\sigma_a = 0.07$ mm⁻¹, $g = 0.88$[29,30]. The ray tracing simulation simulates $2.10^7$ photons.

To characterise the µLED arrays, the laboratory set up consists of a source/measurement unit (Keysight B2911A) and multiplexer (Keithley 2000) for software-programmable addressing of individual µLEDs. The optical power of all 100 µLEDs was measured under a sweep of drive currents. Using a similar optical model but based on the geometry of our laboratory characterisation setup (Supplementary Fig. 3a right), the proportion of optical power from the µLED reaching the detector (detector scale factor, DSF) was estimated using a ray tracing simulation. The optical model in Supplementary Fig. 3a (left) was then used to estimate the proportion of power from the µLED entering the brain tissue (brain scale factor, BSF). Dividing by the area of the illuminated spot size on the cortical surface (approximate diameter 530 µm with window or 270 µm without window, Supplementary Fig. 3b) gives the cortical irradiance in mW mm⁻². This gives a calibration between measured optical power and cortical irradiance and therefore, cortical irradiance as a function of µLED drive current. The measured optical power to irradiance calibration equation is given at the cortical surface.

$$\text{CSI } (mWmm^{-2}) = \frac{\text{Measured optical power } (mW)}{\text{DSF}} * \frac{4\,\text{BSF}}{\pi d^2} \quad (1)$$

Where CSI is the cortical surface irradiance, BSF the brain scale factor, DSF the detector scale factor and $d$ the diameter of the illuminated spot on the cortical surface.

We generate regression fits for current-cortical irradiance for all 100 µLEDs in a device. The drive current can therefore be individually inferred for each µLED from the desired cortical surface irradiance during experimentation by software, allowing normalisation across the array (Supplementary Fig. 3c).

## Optical model experimental validation

To validate the optical modelling experimentally, a pinhole (100 µm in diameter, 40 µm thick) is positioned in front of a photodetector (Supplementary Fig. 3d). The pinhole was positioned as close to the detector as possible to ensure that all light that passes through the pinhole is collected by the detector. The pinhole-photodetector is then translated through the µLED emission profile, measuring the light intensity at each point. The measurement is repeated at various µLED-pinhole distances. The light measurement at each point allows for the beam profile reconstruction with a resolution of approximately the pinhole size. This method is modified from ref. 49.

## Optical interposer validation and modelling

We bonded a thin (80 µm) silicon interposer with 80 µm diameter holes onto the sapphire surface of a packaged device to investigate improving spatial resolution through collimation (Supplementary Fig. 16). Integration of the interposer and hole alignment to pixels was achieved using flip-chip bonding. Emission profiles of devices with and without the interposer were imaged with an SLR camera in 10 µM fluorescein solution (Supplementary Fig. 16a left). To compare the lateral light spread, raw green pixel data was plotted (Supplementary Fig. 16a right, blue lines - error bars computed from different shutter integration times), demonstrating a significant reduction in light spread with the interposer. Both images were compared with Monte Carlo models of equivalent geometry, modelling fluorescence by taking a cut through detectors at equivalent distance from the device (Zemax OpticStudio 2021) (Supplementary Fig. 16a right, orange lines).

The experimentally validated optical interposer was modelled in brain tissue at equivalent delivered optical power to Fig. 4 (Supplementary Fig. 16c).

## Thermal characterisation

The $\mu$LED array was imaged using an infrared (IR) camera (FLIR SC7000-Series) focussed on the sapphire surface using an integration time of 1.5 ms and refresh rate of 100 Hz (Fig. 2a and Supplementary Fig. 6a). Since the camera returns a digital value proportional to the number of detected IR photons, a calibration is necessary to convert this value to actual temperature in degrees celcius. The calibration uses an E-type thermocouple fixed to the back surface of the device. The sapphire surface was brought into focus first by focussing on the mesa structures in the device, and then moving the stage back through the known thickness of the sapphire (150 $\mu$m) (Supplementary Fig. 6b).

An area of sapphire, located close to the $\mu$LED being tested was used for calibration (Supplementary Fig. 6b). Using a heat gun, the device and thermocouple were simultaneously and uniformly heated to over 50 °C and allowed to cool down to room temperature, while both IR camera and thermocouple measurements were made at an interval of 100 ms. This process was repeated 5 times, and a second-order polynomial was fitted to the thermocouple measurements against camera counts (Supplementary Fig. 6c).

For the thermal measurement, the $\mu$LEDs were pulsed under several stimulation protocols. For verification of the model and to more easily assess the time constant of the thermal drop-off, a simple protocol was used; single $\mu$LED pulsed at 5 mA (1 Hz, 50% duty cycle) (Fig. 2a). The IR image in Fig. 2a shows a frame corresponding to the peak temperature from the IR camera calibrated using the thermocouple measurements. The calibration was carried out using a measurement from the sapphire area of the image with the remaining image scaled accordingly, however, this assumes linear emissivity between the materials and so is only indicative.

Thermal modelling used the COMSOL Multiphysics 5.6, heat transfer package. Two thermal models were designed; firstly, to simulate the IR imaging experiment (i.e., the device suspended in air) (Supplementary Fig. 7a) to verify the accurateness of the model, and secondly to simulate a chronic in vivo experiment (Supplementary Fig. 7b). Both are 2D rotationally-symmetric models and implement the device as a layer of gallium nitride (GaN) atop sapphire with parylene-C encapsulation. The device model also includes the poly-imide PCB with a layer of copper on each side, gold bump bonds between the GaN and PCB, and an epoxy underfill. As the model is rotationally symmetric, care has been taken in designing the geometry such that the resulting 3D model has the correct volumes of each material, particularly the $\mu$LED itself, which acts as the heat source in the model. The chronic implantation was simulated by placing the device over a volume of brain tissue with encapsulating cement atop, in a modelled environment. Brain tissue was approximated using the thermal properties of water.

To input thermal power to the model, firstly the electrical and optical power were measured as a function of drive current for a set of 10 representative $\mu$LEDs on one of our devices (Supplementary Fig. 7c). The optical power generated by the $\mu$LED is calculated from the measured power using optical modelling (see Optical modelling Methods section). By subtracting the optical power from electrical power, thermal power was determined (Supplementary Fig. 7d). A second-order polynomial is fitted so thermal power can be calculated for any drive current, and this is then input to the model as a train of pulses (3 mA/5 mA, 20 Hz, 50% duty cycle, Supplementary Fig. 7e).

Since the model is rotationally-symmetric, the $\mu$LED heat source cannot be implemented as a cuboid (the true geometry of the $\mu$LED). Instead, a cylinder of equivalent volume is used and IR imaging used to verify the accurateness of this assumption. To implement multiple $\mu$LEDs at the same time, a concentric ring is used with a radius similar to the distance between the rows of $\mu$LEDs. The volume of this ring is equivalent to the volume of 3 $\mu$LEDs or 8 $\mu$LEDs, with the centre $\mu$LED having a volume equivalent to a single $\mu$LED. A similar comparison was made between IR imaging with squares of $\mu$LEDs pulsed and the model with rings, and both cases were found to agree well (Supplementary Fig. 7g, h).

## Accelerated ageing experiments

The accelerated testing setup uses a heated water bath to ensure a steady temperature and safety for running the experiment over a long timescale. A custom-machined aluminium lid is placed over the water bath with a hole to allow a polystyrene tank to protrude through (Fig. 2c and Supplementary Fig. 8a). This setup allows the device to be submerged in PBS within the tank while preventing water evaporating and escaping the chamber.

Several $\mu$LEDs across the array were pulsed constantly (approximately once each per minute) with a fixed current (5 mA, 10 pulses at 20 Hz, 50% duty cycle) and the voltage was measured to evaluate changes in the resistance of the device (Fig. 2d). A microscope camera images the device for the duration of the experiment. A PT100 sensor is also placed in the solution for accurate temperature measurements over the course of the experiment. 10 different $\mu$LEDs were tested in a repeated cycle for 337 h (approximately 14 days), which corresponds to an accelerated time of 1025 h (approximately 42.5 days) based on the Arrhenius relation for the accelerated testing of polymers[35].

## Animals and ethics

Every experiment involving animals has been carried out following a protocol approved by an ethical commission, in accordance with the French Ethical Committee (Direction Générale de la Recherche et de l'Innovation) and European legislation (2010/63/EU). Procedures were approved by the French Ministry of Education in Research after consultation with the ethical committee #59 (authorisation number APAFIS#28757-2020122111048873).

Optogenetic experiments were conducted on adult transgenic mice (Emx1-IRES-Cre x Ai27D) expressing ChR2 in all excitatory neurons of the pallium[37]. Control experiments were performed on adult transgenic mice (Emx1-Cre) that did not express any opsin.

For freely-behaving experiments only, mice were instead transfected with AAV vectors encoding ChRmine under the control of the CaMKII$\alpha$ promoter, which were injected to drive expression in pyramidal neurons of the auditory cortex (pAAV-CamKII$\alpha$-ChRmine-mScarlet-Kv2.1-WPRE (AAV8). Addgene number: 130991). ChRmine's activation spectrum is red-shifted, but it is broad with less than 50% reduction of elicited current at the wavelength of the $\mu$LED arrays ($\lambda = 450$ nm)[42]. Since ChRmine has much higher photocurrents than ChR2, blue light illumination was sufficient to elicit robust enough neuronal activation for behaviour.

Mice were group-housed in an animal facility (08:00-20:00 light) at a temperature of $22 \pm 2$ °C and humidity between 60–65%, in cages with access to enrichment (running wheel, cotton bedding and wooden logs).

## Surgery: cranial window implantation

For chronic unilateral access to the auditory cortex (AC), a cranial window was introduced, and a metal post for head fixation was implanted on the opposite side. The procedure was performed on 8–12 week-old mice. Buprenorphine (Vertergesic, 0.05–0.1 mg/kg) was administered 30 min before anaesthesia with isoflurane (3%), maintained at 1–1.5% and the animal placed on a thermal blanket during the procedure. The eyes were covered using Ocry gel (TVM Lab), and Xylocaine 20mg/ml (Aspen Pharma) was injected locally at the incision site. The right masseter was partially resected, and a large craniotomy (5 mm diameter) was made above the AC, guided by skull sutures. A 5 mm circular coverslip (150 $\mu$m thick) was sealed over the craniotomy

using cyanolite glue and dental cement (Ortho-Jet, Lang). For the electrophysiology experiment, a grounding electrode (PlasticsOne MS303T/2-B) was implanted contralaterally.

Post-surgery, mice received a subcutaneous injection of glucose (G30) and Metacam (1 mg/kg). The mice were then housed for one week with metacam administered by drinking water or dietgel (ClearH20) without any manipulation. All animals were housed together before and after surgery without any degradation to the implanted grounding electrode.

## Acute in vivo electrophysiology experiments

Electrophysiology recordings were conducted in mice implanted with a glass cortical window above the AC for 3-4 weeks. One hour prior to in vivo experiments, mice were anaesthetised with isoflurane (3% at induction), placed on a thermal blanket and maintained at 1–1.5%. Excess dental cement was trimmed to allow better access to the craniotomy. A 0.5 mm hole was carefully drilled by hand with diamond-coated drill tips in the glass window. Mice were then awakened, and Kwik-Cast silicone (World Precision Instrument) was applied on the cranial window to prevent drying.

Six recording sessions were conducted in five mice: four transgenic mice expressing ChR2 (ChR2 +) and one not expressing ChR2 (ChR2-) as a control. A 64-channel linear recording electrode array (NeuroNexus A1 × 64 Poly2-6 mm-23S-160) was inserted 1.4 mm at an angle of 28° with respect to the brain surface through the previously drilled hole. The electrode array extended to a vertical depth of ~ 680 μm corresponding to the electrode located furthest from the shank tip being positioned on the cortical surface. This allowed estimation of the cortical layer being recorded from and the lateral distance of the recording electrodes from stimulation sites (Fig. 3a, b, Supplementary Figs. 10a, 17). The μLED array was placed on top of the cortical window above the recording electrode array. Voltage traces were recorded at 20 kHz using an RHD2000 USB interface board (Intan Technologies).

The stimulation protocol cycled through each μLED in a pseudo-dorandom sequence, pulsing 10 times (20 Hz, 50% duty cycle) for five cortical irradiance levels: $1.0 \pm 0.3$, $1.2 \pm 0.6$, $2.8 \pm 0.5$, $3.2 \pm 0.8$, and $4.6 \pm 1.2$ mW mm$^{-2}$ every 500 ms. Each μLED-irradiance combination was repeated over 50, 20, or 10 trials, prioritising μLEDs near the electrode array, which was necessary due to head-fixed experimental time constraints. Eight μLEDs were tested at all five irradiances, while others were tested at three: $1.0 \pm 0.3$, $2.8 \pm 0.5$, and $4.6 \pm 1.2$ mW mm$^{-2}$. Cortical irradiances quoted as pooled mean ± pooled standard deviation. Non-functional μLEDs or those with insufficient power were excluded. μLED drive currents corresponded to $2.4 \pm 0.2$, $4.0 \pm 0.3$, $5.8 \pm 0.5$, $7.7 \pm 0.6$ and $10.0 \pm 0.8$ mA. Currents quoted as mean ± standard deviation. Each mouse underwent a 5030-trial experiment lasting over approximately 1 h.

## Electronic control system

A custom electronic control system was developed to power the μLED array and address all 100 μLEDs via software protocols (LabVIEW 2021, National Instruments) (Supplementary Fig. 9). The system includes an Arduino UNO R3 microcontroller for communication with the LabVIEW-equipped PC and programmable 10-channel current DACs using two DC2903A development boards (Analog Devices). A custom shield PCB connects to the Arduino UNO, allowing SPI communication with the current supplies. The PCB also houses a 10-channel solid-state switch array (Texas Instruments TM7211) controlled by the microcontroller. In addition, the microcontroller supplies a trigger signal to the Intan recording system to align recorded data with the μLED stimulation onset and offset.

## Spike sorting and analysis of electrophysiology data

Raw electrophysiology data was captured with Intan RHX and filtered using a bandpass filter with cutoffs at 300 Hz and 5.5 kHz followed by a common average reference subtraction across recording channels (Supplementary Fig. 10b). Pre-processed electrophysiology data was then automatically spike sorted using the Spike Interface package and Kilosort 2.5 (Python 3.10). Manual curation was carried out with Phy to remove incorrectly identified clusters. Stimulation artefacts were also manually identified and excluded from analysis.

A custom Python pipeline was used to aggregate spike counts based on trigger signals recorded by the Intan recording controller and to perform all subsequent data analysis. Spike counts were binned into 5 ms intervals, with each trial consisting of 100 bins (500 ms) of stimulus, 40 bins (200 ms) before stimulus onset, and 40 bins (200 ms) after stimulus offset, totalling 180 bins (900 ms). For simplicity, further binning into 25 ms bins was carried out to coincide with the stimulation protocol (20 Hz, 50% duty cycle).

The pre-stimulus and post-stimulus bins were used to calculate mean background spike counts, which were subtracted from the primary data recorded during the 500 ms stimulus. This allowed determination of which neurons were time-locked to the stimulus and the magnitude of their responses. Data visualisation was carried out using the Matplotlib and Plotly packages in Python 3.

The dose responses (Fig. 3e and Supplementary Fig. 11a) show pooled mean, with error bars as pooled SEM of average spike counts per trial (background subtracted); ChR2+ (optogenetic): $n = 13, 14, 16, 16$ time-locked neurons; ChR2- (control): $n = 51$ neurons. Sigmoid curves were fitted to each set of points to aid visualisation.

Spatial resolution plots (Fig. 4b) show error bars as pooled SEM of standard deviations of average spike counts per trial (background subtracted); $n = 13, 16$ time-locked neurons. Gaussian curves were fitted to each row using data points, x, weighted by the error bars.

$$y = y_0 + A \cdot \exp\left( -\frac{(x - \mu)^2}{2\sigma^2} \right) \qquad (2)$$

Initial values for $(y_0, \mu, A, \sigma)$ were $(0, -500, 50, 200)$ with bounds as $([0, 2], [-900, 0], [0, 200], [100, 1000])$. The x-axis is in μm. The standard error of the fitted parameters defined the shaded region. Reduced $\chi^2$ values were 0.56 and 1.40 for rows 3 and 4 of Experiment 1, and 1.68 and 4.43 for rows 3 and 4 of Experiment 3.

One experiment was omitted (Supplementary Fig. 11): a repeat on the same animal as Experiment 1, carried out a day after a recording probe broke and was manually removed. This data was considered contaminated and excluded from Figure 3.

## Surgery: chronic implantation of flexible packaged μLED array

The procedure was performed on 8–12 week-old mice. Buprenorphine (Vertergesic, 0,05-0,1 mg/kg) was administered 30 minutes before anaesthesia with isoflurane (3%), maintained at 1 –1.5% and the animal placed on a thermal blanket during the procedure. The eyes were covered using Ocry gel (TVM Lab), and Xylocaine 20 mg/ml (Aspen Pharma) was injected locally at the incision site. A square craniotomy of 2.4 x 2.4 mm$^2$ was performed above the auditory cortex using skull bone sutures as a landmark. The μLED array was placed directly on the cortical surface after removal of the dura mater, and the edges were glued to the skull using SuperGlue (Loctite). The flexible PCB packaging was fixed with dental cement (Ortho-Jet, Lang) and an MFB electrode was implanted.

Post-surgery, mice received subcutaneous glucose (G30) and Metacam (1 mg/kg). They were housed for one week with Metacam provided via drinking water or DietGel (ClearH2O) without manipulation. All animals were co-housed before and after surgery without damage to the implanted grounding electrode. However, occasional temporary isolation occurred when co-housed mice chewed on the connectors of implanted devices.

## MFB stimulation

All mice used in the head-fixed behavioural experiments were implanted with an intracranial electrode stereotaxically targeting the medial forebrain bundle (MFB) to deliver a stable reward[40].

The MFB electrode was implanted using stereotaxic coordinates (AP -1.4, ML +1.2, DV +4.8) using a NeuroStar stereotaxic device (Stereodrive 018.117). The electrode and headplate were secured to the skull using dental cement (Ortho-Jet, Lang).

MFB stimulation was provided with a pulse train generator (PulsePal V2, Sanworks) delivering 2 ms biphasic pulses for 100 ms at 50 Hz.

## Behavioural experimental procedure

The experimental setup is shown in Supplementary Fig. 12. The $\mu$LED array is controlled by the Arduino-based electronic control system, while the MFB stimulator is connected to a pulse train generator. A PC running LabVIEW 2023 (for $\mu$LEDs) and MATLAB 2024 routines manages the overall experiment. A National Instruments DAQ detects licks, applies MFB stimulation as a reward, and triggers the LabVIEW routine to apply the appropriate $\mu$LED array illumination pattern.

Mice performed behavioural tasks five days per week (Monday to Friday). Throughout the behavioural training period, food and water were provided ad libitum. **Habituation**: Mice were habituated to head fixation over two days by remaining in the contention tube without reward for 20 min on the first day and 40 minutes on the second day. On the third day, mice received a systematic reward for licking the spout, with a maximum reward rate of 1 reward per second for 500 trials to establish the association between licking and reward. To encourage licking in non-water-deprived mice, the spout was positioned very close to their mouths. During this initial phase, before calibration, MFB stimulation was set at 3 V and reduced by the experimenter if it caused a motor reaction.

**Go training (association)**: Once animals were spontaneously licking, the association experiment was conducted. Go trials were presented with 90% probability, while the remaining trials were blank trials (no illumination and no reward/time-out period). A trial consisted of a random inter-trial interval between 1 and 2 seconds, a 'no lick' period between 3 and 5 seconds (these timings were randomised to avoid prediction of the illumination based on timing), and a fixed response window of 2300 ms. A single lick was registered as a response. Licks during the response window on a go trial was scored as a 'hit' and triggered an immediate reward via MFB stimulation. No lick was scored as a 'miss'. Successful learning of the association task comprised the mouse reaching an accuracy of 80% over an experimental session. Mouse 3 struggled to reach 80% but was moved onto the discrimination task anyway.

**Go/no-go training (discrimination)**: Licks for go trials were rewarded the same as the association experiment. During presentation of the no-go pattern, the absence of a lick was scored as a 'correct rejection', and the next trial immediately followed. Any licking during the response window of the no-go trials was scored as a 'false alarm'; no reward was issued, and a random time-out penalty (between 5 and 7 seconds) was incurred. Each session contained 47.5% probability of a go trial, 47.5% probability of a no-go trial and 5% probability of a blank trial. Due to initially high performance, discrimination tasks were terminated after a set number of trials, dependent on time available for experimentation.

All parameters for the behavioural experiments are given in Supplementary Tables 3 and 4.

## Behavioural data analysis

The voltage on the lick detector was recorded using a National Instruments DAQ. The timing of individual licks was determined from the activation and deactivation of the sensor (a single lick was registered when the mouse's tongue made and then broke contact with the sensor). Licks were recorded throughout the experiment, including during the inter-trial period.

## Freely-behaving experimental setup and procedure

Freely-behaving experiments utilised devices identical to those used for head-fixed behaviour with 15 μm parylene-C encapsulation, with the exception of Mouse 2 which was implanted with a device with enhanced encapsulation (30 μm parylene-C and silicone). Freely-behaving experiments were carried out in a transparent polypropylene box (27 x 25 x 22) cm³ (Fig. 6a, b and Supplementary Fig 14b). The box was elevated on 3D-printed supports fixed to a metal base, facilitating component installation and removal between experimental sessions while ensuring stability during testing. The experimental apparatus featured a dedicated reward delivery zone along one edge of the box. Two rectangular holes in the floor accommodated an infrared fork sensor for monitoring time spent in the "reward zone", while adjacent small holes in the sidewall held the lickport (syringe) connected to a solenoid valve for precise reward delivery. The box lid included an opening for the cable connecting the implanted $\mu$LED array to the control electronics, with a 3D-printed cable holder. A lamp provided consistent ambient illumination throughout experiments, preventing complete darkness and ensuring that light emitted from the $\mu$LEDs would not influence experimental outcomes or animal behaviour.

Association sessions were conducted before the main task to familiarise the animals with the experimental environment, including the presence of a reward zone, optogenetic stimulation delivered via the $\mu$LED array and wearing the connection cable. During this phase, the cortical surface irradiance was set to 10 mW mm$^{-2}$ per $\mu$LED in a $3 \times 3$ pattern (20 Hz, 50% duty cycle, 10 pulses). In the beginning, the rewards were automatically delivered when the mouse broke the IR beam. Then, the required cumulative beam break duration before reward delivery was progressively increased up to 75% of the response window (1 s). Training continued until animals achieved 80% performance accuracy.

Mice advanced to the discrimination experiment employing spatially opposed go and no-go stimuli. These consisted of $3 \times 3$ $\mu$LEDs patterns (20 Hz, 50% duty cycle, 10 pulses) positioned at opposite corners of the $10 \times 10$ matrix (Fig. 6c). During go trials, IR beam interruption triggered go stimulus presentation and initiated the 1 s response window. Successful performance ("hit") required maintaining IR beam break for a cumulative time of ≥75% of the response window, resulting in immediate reward delivery (4 μL water). Shorter interruptions were recorded as "misses" with no reward. During no-go trials, correct performance ("correct rejection") required exiting the beam zone before reaching the 75% threshold, with no reward delivered. Cumulative beam break times exceeding the threshold were classified as "false alarms", triggering a randomised 3-4 second time-out penalty. All trials were followed by a randomised 1-2 second refractory period during which the beam must remain unbroken before the next trial begins (Supplementary Fig. 14a). Discrimination experiments were terminated after a set number of trials, dependent on time available for experimentation.

## Histology

**Tissue preparation**: Tissue samples were obtained and analysed from mice chronically implanted with the $\mu$LED array between 83–101 days; post-surgery recovery: 30 days, behavioural training: 50–70 days. At the end of the experiments, mice were deeply anaesthetised with an overdose of ketamine/xylazine administered intraperitoneally (100 mg/kg and 10 mg/kg i.p. respectively) and perfused transcardially with 1X phosphate-buffered saline (PBS1X) followed by 4% paraformaldehyde solution. Brains were extracted from the skull and kept immersed overnight in 4% paraformaldehyde solution at 4 °C. 40 μm-thick coronal sections were obtained using a vibratome (Leica VT1200S).

**Immunofluorescence**: Slices underwent blocking using a 10% normal goat serum (NGS) (Abcam AB7481) – 0.5% Triton X (Merck, 648462) in PBS1X solution for 1 h at room temperature. Slices were then incubated with primary antibody rabbit anti-Iba1 (1/500) (Anti-Iba1 antibody, monoclonal, Clone #: EPR16588, Catalogue #: ab178846 https://www.abcam.com/en-us/products/primary-antibodies/iba1-antibody-epr16588-ab178846) in a 3% NGS-PBS1X solution overnight at 4 °C. 3% NGS in PBS1X was used as a negative control. The following day, slices were washed with PBS1X for 5 min 3 times prior to an incubation with secondary antibody goat anti-rabbit AlexaFluor 488 (1/500) (Abcam AB150089) for 2h at room temperature. DAPI (Invitrogen Hoechst 33342 H3570) (1/5000) was applied as a nuclear stain to reveal the general anatomy of the preparation. Slices were mounted using FluorSave reagent (Merck, 345789). Fluorescence images were acquired using a Nikon A1HD25 confocal microscope and analysed using ImageJ.

### Reporting summary

Further information on research design is available in the Nature Portfolio Reporting Summary linked to this article.

## Data availability

The raw data used to generate the figures in the article and supplementary material may be found at https://doi.org/10.15129/29d0a492-44fa-4294-a1e2-ee8de64fd958.

Due to large dataset sizes, requests for additional data, including the in vivo electrophysiology or behavioural data, may be directed to the corresponding author.

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

## Acknowledgements

The HearLight project has received funding from the European Union's Horizon 2020 research and innovation programme under grant agreement No 964568, and we thank the wider consortium members. This work was also supported by the Royal Academy of Engineering Chair in Emerging Technologies (KM), Fondation pour l'Audition, RD-2023-1 (BB), FPA IDA02 (BB), Agence Nationale pour la Recherche, France 2030 programme, ANR-23-IAHU-0003 (BB). The authors would also like to acknowledge Shuzo Sakata for advice on the histology, and the following staff within the Institute of Photonics; cleanroom support: Ronnie Roger, Beniot Guilhabert, Ian Watson; administration: Sharon Kelly; machining work: Lewis Hannah.

## Author contributions

R.G. designed and implemented the device packaging and carried out laboratory characterisation under the guidance and supervision of K.M. A.V., E.C., A.A. and E.M.K. carried out the surgeries, in vivo experiments and data collection/analysis under the guidance and supervision of B.B. RG prepared the figures. R.G. and A.V. analysed the in vivo data and co-authored the writing of the manuscript under the supervision of K.M. and B.B. E.B. and Y.C. designed and fabricated the μLED arrays, supervised by K.M. and based on an approach developed by M.D.D. N.M. assisted with laboratory experiments and device packaging. R.G. and N.M. developed and implemented the optical modelling. B.B. and K.M. conceptualised the devices and experimental design, which was adapted over the course of the experiments through input from R.G. and A.V. Histology and histological analysis were conducted by M.P. K.M. and B.B. are joint senior authors. All of the authors reviewed, edited and contributed to the manuscript.

## Competing interests

The authors declare no competing interests.
