## [Peer Review File · Nature Communications]

Chronically implantable μ LED arrays for optogenetic cortical surface stimulation in mice

Corresponding Author: Mr Ryan Greer

Version 0:

Reviewer comments:

Reviewer #1

(Remarks to the Author)

This study by Greer, Mathieson and co-workers presents the development and in-vivo use of a 100-element micro-LED (μ LED) array for optogenetic stimulation of the mouse auditory cortex. The authors fabricate a 2×2 mm² array of GaN-on-sapphire μ LEDs with 200 μ m pitch, integrated into a chronically implantable, flexible system. Tests indicate that the device can enable stable, spatially confined stimulation over weeks, with manageable thermal impact. Electrophysiological recordings in transgenic mice expressing ChR2 confirm neuronal activation; spatial resolution is estimated to be 600 to 800 μ m, consistent with optical modelling. Chronic behavioural experiments demonstrate that mice can learn to discriminate spatial patterns of cortical activation using these arrays.

The goal of achieving high-resolution, minimally invasive optogenetic stimulation in freely behaving animals has motivated the development of many implantable μ LED systems. While this work is fundamentally different from earlier demonstrations, it convincingly demonstrates a technically robust and scalable platform that advances that goal. I believe that this report will be of interest to the broad optogenetics and neuroimplants communities. However, the manuscript needs to be revised to address a few shortcomings and to make the discussion more accessible to a broad audience.

As the authors explain, resolution is an important aspect of any optical stimulation device. The Lambertian emission profile of the μ LEDs and the separation between μ LED and target tissue lead to a much lower spatial resolution than one might have aimed for. The authors should comment in more detail on future strategies to improve this situation. There is a brief discussion on a potential "optical interposer". What about reducing the separation between light-source and tissue surface? At present there is a 150 μ m thick sapphire substrate, adding very significant distance between the μ LED and the target region in Layer 2/3. (Depending on how one counts, essentially doubling the intrinsic distance given by Layer 1.) Are there device designs or LED variants where this substrate could be thinned further or avoided entirely?

Related to the prior point, the simulation and schematics always appear to show the bare device, without encapsulation. In the methods, we learn that the parylene coating is 15-30 μ m in thickness, and there is a further silicone coating of unspecified thickness. The authors need to specify the exact geometry and should show this in the schematics as well.

In Figure 3e and other data, the variation of neuronal response to the μ LED stimulation appears very large (the average spike counts between experiment #3 and #4 appear to differ by almost a factor of four). Is this to be expected? Would the variation be similar for external stimulation, e.g. with a DMD or is this the result of heterogeneity from the device?

Likewise, the variation in average spike count for different stimulation regions across the device is dramatic, with some regions showing nearly no response. The authors should elaborate more on the origin of this effect. It remains unclear if this is related to the location of the recording device or intrinsic to either the illumination device or the animal model.

Related to this point, given the broad audience at Nature Communications, the methods used to pinpoint the location of active neurons from the limited spatial data provided by the ephys shank, which only intersects Layers 2/3 in a narrow region, should be explained, at least briefly.

Finally, how would the approach described here be scaled in the future, both in terms of device size (for work with larger animal models, a 2×2 mm² device will not be useful) and in terms of achievable resolution (where in addition to the

challenge of getting the light sources closer to the target region or finding other means of minimizing light spreading, see above, the number of μ LEDs will need to be scaled dramatically). For instance, keeping the same pitch on a 10x10mm² device would require a 25-fold increase in number of addressable LEDs and substantially larger InGaN on sapphire samples. Would this be practical at some point?

Reviewer #2

(Remarks to the Author)

In this manuscript, Greer et al. describe a microfabricated μ LED array designed for chronic implantation to interface with superficial cortical regions in mice. The authors detail the fabrication process, characterize optical transmission and heat generation during device operation, and validate functionality by demonstrating neuronal activation and behavioral discrimination experiments in the auditory cortex. While the study addresses an interesting technological area, I find significant shortcomings that substantially limit its novelty, impact, and overall contribution to the field. Consequently, I cannot support publication of this manuscript in its current form. The major concerns are detailed below:

Major concerns:

1. While the technology itself appears robust, the authors fail to adequately compare and contextualize their approach relative to recently published technologies, including the group's previous work. For example, the Utah optrode array reported by McAlinden et al. (2024) is likely more useful for cortical neuromodulation, as it separates the LED source from neural tissue and offers beam control capabilities. Pollmann et al. (2024) reported a system (SCOPE) using a similar cortical arraying idea for all-optical bidirectional neural interfacing, providing both recording and modulation with greater controllability. Shin et al. (2024) demonstrated a wireless operation solution. Taal et al. (2024) achieved an array with 1024 OLEDs, providing single-neuron resolution. These works, although cited in the manuscript, are not adequately contrasted or acknowledged.
2. The presented μ LED array is proposed as a tool for behavioral neuroscience; however, existing methods such as holographic optogenetics (see Adesnik & Abdeladim, 2021), multi-fiber approaches (e.g., Sych et al., 2019), and single-photon scanning optogenetics (e.g., Pinto et al., 2019) are not discussed. These systems, could enable experiments with much higher spatiotemporal resolution in the same neuronal ensembles, over extended periods, and with far less concern about off-target heating effects (at least in head-fixed setups). Additionally, the complexity of clean-room fabrication processes severely limits accessibility and ease of adoption by typical neuroscience laboratories, especially without clear practical advantages demonstrated by the current study.
3. The authors claim that the potential advantage of this system lies in its use with freely behaving animals (albeit at the cost of spatial resolution and deep-layer access). However, this supposed advantage is unsubstantiated in the current work, as it is not demonstrated experimentally in the paper.
4. The Lambertian radiation pattern of micro-LEDs is generally not precise. The spatial separation between the neuronal ensembles and the LEDs—introduced by the packaging process—further worsens this issue. Thus, the spatial resolution is intrinsically limited even if the LED array is made with a high density of emitters. The authors propose a beam-shaping method in supplementary information, but this approach is not experimentally validated.
5. In terms of optical efficiency, the authors demonstrate that about 90% of the electrical power is converted to heat. Although the modeling results suggest the resulting temperature rise is physiologically safe, this claim must be validated experimentally in vivo. How was this modeled in air, and what conditions were assumed in the brain tissue model? Is a 3–5 mA drive current sufficient to recruit neuronal activation, given that the authors used 10 mA in their in vivo electrophysiology experiments (which would correspond to roughly a one- to two-fold higher temperature increase)? Additionally, the absorption of blue light by tissue also contributes to heat generation, and it is unclear why this was not taken into account.
6. The flexibility claims of the device are difficult to reconcile with the use of a 150- μ m sapphire substrate. The authors must clarify this discrepancy or provide additional experimental support.
7. Blue light penetrates poorly in living brain tissue. Therefore, the authors must validate the light transmission profile predicted by the Monte Carlo simulation with experimental measurements in actual brain tissue.
8. Given the relatively large size of the head plate shown in Figure 1c, it is unclear whether this surface-interfacing arrangement will outperform the authors' previous optrode array approach. That earlier approach avoids heat generation at the neural interface and offers beam-regulating (beam-shaping) capabilities. The advantages of the new surface array over the established optrode array are not convincingly demonstrated.
9. It is unclear why the authors used a 20 Hz stimulation with a 50% duty cycle for their electrophysiology experiments, as this protocol is likely to induce a sustained cation influx through ChR2 and could result in multiple spikes per neuron per 25 ms. This protocol also generates substantial heat. In Figure 3c, the authors identify single-unit activity among multiple spikes, but it is not clear what these spikes are time-locked to – the onset of the stimulation or the offset of the stimulation. The authors also repeatedly mention using a “low” drive current; however, in their experiments they used 8.0 mA and 10.4 mA, which are relatively high currents and contradict the claim of a low-current drive. Additionally, it is unclear how the electrophysiological data were filtered and preprocessed, since one would expect evoked local field potentials given this optogenetic protocol, yet none are shown in their traces.

10. The experiment in Figure 3f demonstrating spatial resolution is a valuable addition. However, it is not justified to compare the neuronal responses directly to the Monte Carlo simulation results to derive an “activation threshold.” This approach ignores important morphological considerations of pyramidal neurons: activation of ChR2 in neuronal processes (such as apical dendrites) can contribute to action potential generation at cell bodies, especially since the viral construct used here does not have subcellular specificity. Consequently, the large apparent depth of neuronal activation shown in Figure 4g is likely due to the activation of ChR2 in apical dendrites in the superficial layers. Thus, it is very misleading that the authors’ figure suggests ~1 mW of 450 nm light can reach approximately 500 μm deep into layer 4, as this depiction overlooks the contribution of dendritic activation in superficial layers.

11. The behavioral experiments presented here do not clearly showcase the capabilities of the technology. They still require a head-fixed setup, a restraint tube, and a medial forebrain bundle (MFB) stimulator to motivate the animals during the tasks. Again, the authors used a 20 Hz stimulation with a 50% duty cycle for these behavioral experiments, which compromises the inherently high temporal resolution advantage of optogenetics. In summary, the behavioral results do not demonstrate a clear benefit of the micro-LED array approach compared to existing optogenetic methodologies for similar experiments.

Reviewer #3

(Remarks to the Author)

General comment

Greer et al. present a μLED -based optogenetic stimulation system designed for chronic cortical surface implantation in mice. This study is notable for its comprehensive in vivo validation, integrating acute electrophysiology to assess neuronal activation and chronic implantation to evaluate long-term functionality and behavioral effects. The dual experimental approach strengthens the study’s relevance in neural interface research. While the μLED array is designed for high spatial resolution, the reported optical spread raises questions about the precision of neuronal targeting. Additionally, electrophysiological recordings were limited to a single probe insertion, making it unclear whether the findings generalize across the full μLED array. Lastly, despite being intended for chronic use, the study does not include direct biocompatibility testing such as histological analysis of immune response or tissue damage. Addressing these aspects would further strengthen the validation of the system. Therefore, the following points should be considered before publication.

Major comment

1. As highlighted in the abstract and title, one of the main claimed advantages of this paper is the device's suitability for chronic implantation due to its flexibility. However, the device presented in the manuscript utilizes a sapphire wafer with a stated thickness of 150 μm for the implant device. Given the rigidity of sapphire, there is concern whether sufficient flexibility has truly been achieved to justify this claim.

2. Additionally, given the expected high Young’s modulus of the described sapphire-based device, it likely exhibits limited flexibility, resulting in poor conformal contact with the brain's surface. Given this rigidity, the presence of wrinkles or folding—necessary for flexibility—seems unlikely. Therefore, it raises concerns regarding the precision and accuracy when claiming targeted stimulation in specific brain regions. Clarification and supporting data on the device's ability to accurately target stimulation despite potential mechanical mismatch would be beneficial.

3. Similarly, due to the rigidity of the device, issues related to micro-motion after implantation are likely to arise. It is expected that prolonged implantation of this rigid device may lead to damage in the brain tissue. Although the authors have confirmed device functionality post-explantation, the manuscript lacks information regarding potential damage to the brain. Therefore, histological data demonstrating the degree of tissue damage following device implantation should be provided. Additionally, despite describing an implantable device, the authors have not included data regarding immune responses, which should also be addressed.

4. Further data related to temperature management appear necessary. Duty cycle and frequency can significantly vary depending on the experimental conditions employed. Additional data specifically demonstrating temperature variations under different experimental conditions, such as varying duty cycles and frequencies, would enhance

the robustness of the results and the claims made in this study.

5. In the current experimental setup, the authors investigated temperature changes with only 10 pulses of LED activation. However, if prolonged, continuous operation occurs beyond these conditions, it is expected that temperature would continually increase. Therefore, it remains unclear whether the device can safely operate for extended periods, such as 100 seconds, without encountering significant thermal issues. Additional data evaluating the device under prolonged LED activation conditions should be provided.

6. In the presented lifetime data, the sequential activation of ten LEDs, each with a 1-

minute interval between activations, does not adequately reflect typical in vivo scenarios where multiple LEDs might operate simultaneously or with shorter intervals.

Consequently, the reported lifetime results may not accurately represent practical usage conditions. Additionally, simultaneous use of multiple LEDs could significantly raise local temperatures, potentially decreasing device lifetime. A detailed discussion addressing this scenario and its implications on device performance and longevity is warranted.

7. Following the above points, the manuscript does not adequately address the implications of the inability to simultaneously operate all LEDs. If simultaneous activation is restricted due to thermal or other limitations, this undermines the claim of effectively covering a large area. Thus, further discussion and evidence are required to substantiate how the device achieves broad-area coverage when limitations on simultaneous LED operation are considered.

8. The study presents compelling evidence that μ LED-based optogenetic stimulation can modulate behavior through artificial perception. However, the exact duration of the behavioral training period is not clearly stated in the manuscript. Given that Figure 5(d) shows a learning curve spanning multiple sessions, it would be helpful to specify how many days or weeks the behavioral training lasted. Please clarify the total duration of the behavioral training in the Methods section, including the number of sessions per day and whether any criteria were used to determine when training was complete. This information will help readers better understand the timescale required for optogenetic learning and its potential for long-term applications.

9. The manuscript claims that the μ LED array provides high spatial resolution, with a 200 μ m pitch enabling precise neuronal targeting. However, in Figure 4, the reported light spread (600–800 μ m) suggests that individual μ LEDs may activate overlapping neuronal populations. This raises the question of whether the effective resolution is truly determined by the μ LED pitch, or if light scattering imposes limitations on selectivity.

Additionally, while the study provides an estimate of spatial resolution based on electrophysiological data and optical modeling, a direct comparison with previous optogenetic stimulation technologies (e.g., fiber-optic stimulation, lower-density μ LED arrays) is not clearly presented. Given that one of the key claims of this work is improved spatial precision, it would be beneficial to include a quantitative comparison of spatial resolution with prior studies using different optogenetic stimulation methods.

10. The authors highlight the high spatial resolution enabled by the 10 \times 10 μ LED array.

However, if neuronal responses were recorded from only a single insertion site using a single 64-channel electrode probe, this significantly limits the validation of the claimed spatial resolution. The potential advantage of the μ LED array—precisely targeting different neuronal populations—cannot be fully demonstrated if only a small region is sampled. Clarifying whether multiple insertion sites were used across experiments or discussing how the recorded neuronal responses generalize across the full μ LED array would provide stronger evidence supporting the advantages of the high-density μ LED system.

Minor comment

1. The study demonstrates that μ LED stimulation can modulate neuronal activity and behavior. However, it remains unclear whether the frequency or temporal structure of the μ LED pulses influences neuronal responses differently. Further discussion on how different stimulation frequencies affect neuronal activation and behavioral performance would provide deeper insight into the functional implications of the stimulation paradigm.

2. The manuscript outlines the positioning of the μ LED array on the cortical surface and the insertion of a 64-channel electrode probe at a 28° angle. However, it does not provide sufficient detail on how the electrode was aligned relative to the μ LED array to ensure accurate mapping of neuronal responses to specific stimulation sites. Given the study's emphasis on spatial resolution, further clarification on the alignment strategy would strengthen the interpretation of the recorded neuronal activity.

3. The authors describe the use of Parylene-C, Polyimide, and Silicone for chronic implantation, as these materials are commonly used in neural interfaces. However, the manuscript does not provide any direct biocompatibility testing or histological analysis to confirm the absence of immune responses or tissue damage over time. Given that the μ LED array is designed for long-term implantation, including supporting data on tissue response or referencing previous biocompatibility studies using similar packaging materials would strengthen the validation of its chronic use.

4. The multi-unit analysis in Supplementary Figure 10 effectively demonstrates how μ LED stimulation modulates neuronal activity across different cortical depths. This analysis provides valuable insight into the spatial extent of optogenetic activation, which many readers may find crucial for understanding depth-dependent electrophysiological responses. Given its importance, incorporating this data into a main figure would

improve accessibility and further strengthen the discussion on the spatial resolution of μ LED-driven neuronal activation.

Version 1:

Reviewer comments:

Reviewer #1

(Remarks to the Author)

The authors have provided a carefully revised manuscript which discusses and as best as could be expected addresses the issues I brought up in my previous report. I would only ask that, for clarity, the following part is amended as follows to avoid any ambiguity about whether these additional steps have already been implemented (line 377):

“The trade-off is a reduction in spatial resolution due to the 150 μ m standoff between source and brain surface. IN THE FUTURE, this COULD be reduced by using ...” (or of course similar wording that clarifies this point)

Apart from this minor change (that will not require me to look at the manuscript again), I recommend publication of the study as is.

Reviewer #2

(Remarks to the Author)

In the revised manuscript, the authors carried out new experiments demonstrating the use of their device in freely behaving behavioral paradigms, and they expanded the discussion to better position their work relative to other existing technologies. From these perspectives, the revised document is appropriate in terms of significance. The authors have now addressed what they originally claimed as a major technical advantage (freely-behaving experiments) and have more clearly contextualized their contributions within the broader literature.

Furthermore, the authors included experimental validations of their prior optical and thermal simulations. While these validations remain indirect — for example, relying on IR imaging in air or water rather than in vivo, or optical measurements without considering tissue scattering and absorption — they are reasonable given the technical challenges of in vivo validation and the supporting role these data play for the core technological contributions.

Overall, I am positive about this work. Many of the previous major points have been addressed, and the technical advancement has strong potential for broad impact in the field. However, I have several remaining concerns regarding specific experiments and internal consistency in the revised manuscript. These issues should be clarified to improve scientific rigor and highlight important caveats.

Remaining Concerns

1. The addition of freely behaving behavioral experiments is exciting. However, it is unclear whether opsin-negative control mice were included. If these experiments were conducted, the results should be shown; if not, please justify their omission (noting that some other experiments already included controls).
2. Most behavioral tasks used 3x3 LED activation patterns rather than single-pixel stimulation. Is this because individual LEDs within a 3x3 block cannot be independently addressed simultaneously due to PN junction wiring? If so, this limitation undercuts the repeated claim of spatial precision. Please clarify and justify this when presenting the behavioral experiments.
3. Many in vivo experiments were conducted at ~ 10 mW/mm² at the cortical surface, which would yield >1 °C heating based on the authors' own simulations. The rebuttal argues that because cortical windows are absent in chronic implants, the same irradiance can be achieved at <3 mA per LED (Supplementary Fig. 1d), implying this is safe. This reasoning is flawed: removal of the window decreases the source–tissue distance and can increase heating at the interface, even if the driving current is lower. This caveat should be explicitly acknowledged.
4. Throughout the manuscript, the authors emphasize the accuracy of their optical and thermal simulations. Given the simplifying assumptions (water model, omission of absorption, dendritic contributions, etc.), it would be more accurate to present these as approximations that may over- or under-estimate in vivo outcomes. Please moderate claims of “accuracy” and frame results as approximate guides with caveats.
5. The histological analysis raises questions. When layer 1 is included, the implanted side shows only a modest increase in Iba-1+ cells (228 ± 24 vs. 181 ± 18 per mm²), albeit statistically significant. Excluding layer 1 increases the numerical difference but eliminates statistical significance. The use of a non-parametric Wilcoxon test may not capture this nuance appropriately.
The overall Iba-1 signal appears surprisingly low given that the dura was removed and the device directly contacted cortex. Was tissue collected long after implantation, potentially missing acute responses? Please indicate the interval between surgery and tissue harvest.
Adding GFAP staining (astrocyte marker) would provide an important complementary measure, particularly given the low microglial response reported.

6. Supplementary Fig. 10b shows that filtering introduces potential artifacts, and the apparent spiking activity may in part reflect residual LFP responses after filtering. The single-unit spike sorting (Fig. 3c, right) is therefore not fully convincing. It would strengthen confidence if the authors showed examples of clearly identified spikes in raw, unfiltered traces at higher temporal resolution (e.g., 10 ms windows), even if superimposed on LFPs. While the LFP responses themselves suggest successful opsin activation, the claims of single-unit resolution need clearer support.

7. The choice of 20 Hz at 50% duty cycle is not ideal: it imposes a higher thermal load and diminishes the temporal precision that is a key advantage of optogenetics. Please explicitly note that this protocol was chosen as a proof-of-concept for behavioral salience, but is not optimal for optogenetic control.

8. It remains difficult to track which encapsulation scheme (material, thickness, coatings) was used in each experiment. Please provide a supplementary table summarizing encapsulation conditions across all experimental contexts for clarity.

Reviewer #3

(Remarks to the Author)

The concerns regarding the behavioral experiments, LED operational parameters, and temperature increase appear to be well addressed. However, my primary and most significant concern, which was initially raised regarding the device's suitability for chronic applications, remains. The rigidity of a device is largely determined by its thickness and the Young's modulus of its constituent materials. In our view, the device's thickness and materials are not suitable for it to be considered a chronic implant. Furthermore, it is difficult to obtain precise information about the chronic in-vivo experiments conducted to resolve this issue, and we have reservations about whether the immune response data has been interpreted correctly. Further discussion on these points is required.

Regarding Comment #1

Our concern regarding the device's lack of 'flexibility' from its rigid sapphire substrate was misunderstood. The response incorrectly addresses surgical 'positioning' instead of this fundamental material limitation. The use of a rigid sapphire substrate makes any discussion of flexibility meaningless. Because the justification provided does not resolve this issue, the response is insufficient. Please provide a revised answer focused on the material's rigidity.

Regarding Comment #2

We question the authors' assertion that the 2x2 mm² cortical area is 'flat.' This claim is directly refuted by the authors' own data in Supplementary Figure 2(a), which clearly shows tissue deformation at the implant site caused by the rigid device. A visible difference in compression exists between the implanted and non-implanted hemispheres, which would not occur if the surface were truly flat. Conformal contact with soft tissue is governed by bending stiffness, not the topography of the target area. Given the high stiffness imparted by the sapphire and the device's thickness, the claim that it can achieve conformal contact on the soft cortical surface is not well-supported. This comment has not been satisfactorily addressed.

Regarding Comment #3

The response lacks information on the total implantation period for the histology study, a critical parameter for the authors' claims. To validate the device as a 'chronic implant,' the immune response must be assessed following a genuinely long-term implantation. This essential information has not been provided, which weakens the claims of chronic biocompatibility. The duration needs to be explicitly stated.

Reviewer #1 (Remarks to the Author):

This study by Greer, Mathieson and co-workers presents the development and in-vivo use of a 100-element micro-LED (pLED) array for optogenetic stimulation of the mouse auditory cortex. The authors fabricate a 2×2 mm² array of GaN-on-sapphire pLEDs with 200 pm pitch, integrated into a chronically implantable, flexible system. Tests indicate that the device can enable stable, spatially confined stimulation over weeks, with manageable thermal impact. Electrophysiological recordings in transgenic mice expressing ChR2 confirm neuronal activation; spatial resolution is estimated to be 600 to 800 pm, consistent with optical modelling. Chronic behavioural experiments demonstrate that mice can learn to discriminate spatial patterns of cortical activation using these arrays.

The goal of achieving high-resolution, minimally invasive optogenetic stimulation in freely behaving animals has motivated the development of many implantable pLED systems. While this work is fundamentally different from earlier demonstrations, it convincingly demonstrates a technically robust and scalable platform that advances that goal. I believe that this report will be of interest to the broad optogenetics and neuroimplants communities. However, the manuscript needs to be revised to address a few shortcomings and to make the discussion more accessible to a broad audience.

Comment 1

As the authors explain, resolution is an important aspect of any optical stimulation device. The Lambertian emission profile of the pLEDs and the separation between pLED and target tissue lead to a much lower spatial resolution than one might have aimed for. The authors should comment in more detail on future strategies to improve this situation. There is a brief discussion on a potential “optical interposer”. What about reducing the separation between light-source and tissue surface? At present there is a 150pm thick sapphire substrate, adding very significant distance between the pLED and the target region in Layer 2/3. (Depending on how one counts, essentially doubling the intrinsic distance given by Layer 1.) Are there device designs or LED variants where this substrate could be thinned further or avoided entirely?

Thank you for highlighting these aspects on the spatial resolution. We have expanded the discussion to include approaches that improve the spatial resolution through the removal of the substrate material (reducing the source-target distance). The paragraph has also been restructured to better contextualise our approach with recent literature.

“The μ LEDs are fabricated on a thinned sapphire substrate, therefore maintaining stress-matching to the μ LED quantum well structures, resulting in high-intensity sources capable of driving robust optogenetic responses in mouse cortex from a single site or spatiotemporal patterns. The trade-off is a reduction in spatial resolution due to the 150 μ m standoff between source and brain surface. This can be reduced by using top-emitting devices or removing the LED structures from their substrates, either through laser lift-off processes or through selective chemical etching (Klein et al. 2019). The latter offers fully flexible interfaces and can both deliver reduced source-tissue proximity and similar stimulation site spacing (Sekiguchi et al. 2022, Gu et al. 2024). However, device quantum efficiency is compromised and further in-vivo validation of these approaches need to confirm the μ LEDs can achieve sufficient intensity for behavioural outcomes as well as long-term, chronic stability.”

Comment 2

Related to the prior point, the simulation and schematics always appear to show the bare device, without encapsulation. In the methods, we learn that the parylene coating is 15-30pm in thickness, and there is a further silicone coating of unspecified thickness. The authors need to specify the exact geometry and should show this in the schematics as well.

Thank you for highlighting this. We did vary the encapsulation depending on the device configuration and experiment, which was not fully explained in the manuscript. For the acute device used for

electrophysiology experiments, we used a 7 rim parylene-C (PaC) coating. For the chronic devices which were implanted directly on the brain surface without a cortical window, we increased the PaC thickness to 15 rim. We then explored how to further improve the encapsulation of the chronic devices by using two 15 rim PaC coatings (totalling 30 rim) and a silicone dip coating, which were evaluated in accelerated testing only. With the exception of Mouse 2 in the freely-behaving experiments, which had a chronic implant with the improved encapsulation (30 rim PaC + silicone), all in-vivo behavioural experiments used chronic devices with the standard 15 rim PaC and no silicone. As a result, the optical models were designed to reflect the two main encapsulant designs: 1) a 15 rim PaC layer in front of the sapphire for chronic experiments (e.g. Fig. 4e, Fig. 4g); 2) a 7 rim PaC used in the acute device for the electrophysiology experiment, which included a cortical window (Fig. 4c). The silicone encapsulation process (now properly detailed in the methods) was designed to avoid silicone encapsulation on the sapphire (emitting) surface and so plays no role in the light transmission to the brain surface and is not included in the optical models. We address this in the manuscript as follows:

Schematics of device cross-section have been updated throughout the report (Fig. 1e, Fig. 4c, Fig. 4e, Fig. 4g, Supplementary Fig. 3a, Supplementary Fig. 16c) to include the 7 rim or 15 rim PaC used in optical models.

Fig. 1e caption has been updated to include more detail on model.

“Model reflects chronic implantation with 150 iim sapphire and 15 iim parylene-C (PaC).”

Fig. 4c and Fig. 4d captions updated to clarify different PaC thicknesses.

“(acute device model with 7 iim parylene-C)” and “(chronic device model with 15 iim parylene-C)” respectively.

Supplementary Fig. 3a caption also updated.

“7 μm parylene-C (PaC) modelled for acute device (with cortical window) and 15 μm PaC modelled for chronic device (implanted directly on brain tissue).”

Text in Results relating to device lifetime testing has been updated to improve clarity.

“Subsequent packaging steps include a polyimide PCB, medical-grade epoxy underfill and 15 iim parylene-C (PaC) conformal coating. For accelerated testing, an enhanced encapsulation strategy consisting of an additional 15 iim PaC coating and medical-grade silicone dip coating was evaluated.”

“Other encapsulation strategies were evaluated (Supplementary Fig. 8b), with 30 iim of PaC and silicone providing the best performance.”

Methods (riLED array packaging) text has also been updated to clarify the different encapsulations as well as to detail the silicone dip coating procedure fully along with an additional panel (Supplementary Fig. 4c).

“The system is coated with 15 iim of parylene-C (PaC). We further explored the encapsulation procedure in accelerated tests only by adding an additional layer of PaC (15 iim) as well as a medical-grade silicone dip coating (NuSil MED-2214). Supplementary Fig. 4c shows the steps involved in applying the silicone dip coating. For our devices, a silicone primer (NuSil MED-161) was firstly applied using a small bristle brush and allowed to dry for one hour. A small piece (3 x 3 mm²) of UV-sensitive dicing tape was attached to the sapphire surface of the device. The device was fully submerged in MED-2214 which had been decanted into a beaker. While submerged, the beaker was rotated slowly several times with the device held at different angles, before carefully lifting the device out and leaving it suspended on a drying rack for 10 minutes at room temperature until the silicone partially cured. A second coating was applied in the same manner as before and the device suspended on a drying rack, this time above a hotplate set to 50 °C, for one hour. The dicing tape was exposed to UV light and peeled off, so the sapphire surface was clear but the rest

of the device was encapsulated; this was necessary to minimise standoff distance between the μ LEDs and the brain surface. A final cure was carried out at 150 °C until the silicone had fully cured.”

Supplementary Figure 2 (c) Steps for medical-grade silicone dip coating encapsulation. Applying UV-sensitive dicing tape (i). Dip 1 + beaker rotation (ii). Dry 1 (10 mins, room temp) (iii). Dip 2 + beaker rotation (iv). Dry 2 (1 hour, 50 °C) (v). UV exposure and peel back dicing tape (vi). Final dry (150 °C, until rubber) (vii).

Methods (Optical modelling and characterisation) text has been updated to clarify parylene-C thicknesses.

“The model includes a 2 mm square sapphire substrate (150 μ m), parylene-C layer (7 μ m for acute device with cortical window or 15 μ m for chronic device directly on brain tissue), optional cortical window (150 μ m) and auditory cortex.”

Comment 3

In Figure 3e and other data, the variation of neuronal response to the μ LED stimulation appears very large (the average spike counts between experiment #3 and #4 appear to differ by almost a factor of four). Is this to be expected? Would the variation be similar for external stimulation, e.g. with a DMD or is this the result of heterogeneity from the device?

The variability of neuronal responses in Fig. 3e is indeed large, but this is expected because extracellular electrophysiology recordings are intrinsically variable and because the distance between the electrode insertion site and the μ LED array varied from experiment to experiment by roughly ± 200 μ m (Supplementary Fig. 10a), leading to variance in the μ LED to electrodes distances. The sources of variability in population firing rates in this figure are therefore: the health of the tissue which depends on multiple factors of the preparation (quality of the craniotomy, fresh insertion or after a few days) and the variation in position of the electrode array with respect to the μ LED array. We observed similar variability in recordings made with DMDs in the Bathellier lab. The same device was utilised throughout all electrophysiology experiments and experienced some degradation in optical performance, however we quantified this by monitoring light output versus drive current before and after the experiments and factored this into the error bars shown on, for example, Fig. 3e. Also, the device was current-driven during experiments, which means even if there are fluctuations in electrical performance the voltage is modulated to ensure consistent optical power is delivered to the brain. The exact horizontal alignment of the array with the cranial window varied slightly also. We updated the results to explain this variability:

“Variation in the neuronal response between experiments could be due to variation in the relative positioning of the recording electrode array to the μ LED stimulation site (Supplementary Fig. 10a), the flatness of the μ LED array placement on the cortical window, or the overall electrode array insertion quality and surrounding tissue health. Improved consistency between experiments is possible with removal of the cortical window and with our chronically implantable devices, where the supporting polyimide PCB is flexible, making it easier to fix into position.”

Comment 4

Likewise, the variation in average spike count for different stimulation regions across the device is dramatic, with some regions showing nearly no response. The authors should elaborate more on the origin of this effect. It remains unclear if this is related to the location of the recording device or intrinsic to either the illumination device or the animal model.

In our electrophysiology experiments, we use a single shank linear recording electrode array to measure neuronal responses under stimulation from individual μ LEDs across the device. Despite driving responses across the region underneath the μ LED array, we were only able to detect responses from a subset of neurons and ensembles clustered nearby the recording electrodes (see Supplementary Fig. 10a), giving rise to the non-homogeneous single-unit spiking response shown in Fig. 4a, which is averaged from the activity of all detected neurons when each μ LED is illuminated individually.

These experiments allowed us to record individual neuron response being driven as a function of space, thereby establishing a spatial resolution, or width of activation by an individual μ LED. For this study a single electrode shank configuration was sufficient. Thank you for highlighting this, we have clarified this point within the Results section:

“The spatial resolution was inferred from the full width at half maximum (FWHM) of these fits. Due to the use of a single shank recording electrode device with fixed position, spatial resolution estimates could only be obtained from μ LEDs located in rows near the edge where the electrode array was inserted.”

Comment 5

Related to this point, given the broad audience at Nature Communications, the methods used to pinpoint the location of active neurons from the limited spatial data provided by the ephys shank, which only intersects Layers 2/3 in a narrow region, should be explained, at least briefly.

Throughout electrophysiology experiments, we used transgenic mice (Emx-IRES-Cre x Ai27d) which express ChR2 in all excitatory neurons, most of which are pyramidal neurons. There are a high density of pyramidal neurons and dendrites in layer 2/3 of the auditory cortex where we targeted our optogenetic stimulation and neuronal recordings. The auditory cortex was identified using anatomical landmarks for surgical implantation of the cortical window, which a hole was inserted in and the recording electrode array shank inserted for electrophysiology. We inserted the shank at an angle of 28° to the surface of the cortex, with the probe extending to a vertical depth of approximately $680 \mu\text{m}$. Using micrographs, the lateral position of the recording electrodes with respect to the stimulation sites in each experiment was also approximated (Fig. 3a, Fig. 3b, Supplementary Fig. 10a). Therefore, analyses surrounding thresholds and spatial resolution focussed on μ LEDs located nearby the recording electrodes. We have updated the methods (Acute in-vivo electrophysiology experiments) to provide more detail on this.

“The electrode array extended to a vertical depth of $\sim 680 \mu\text{m}$ corresponding to the electrode located furthest from the shank tip being positioned on the cortical surface. This allowed estimation of the cortical layer being recorded from and the lateral distance of the recording electrodes from stimulation sites (Fig. 3a, Fig. 3b, Supplementary Fig. 10a, Supplementary Fig. 17).”

We then carried out spike sorting to attribute the recorded electrical activity to individual neurons, and determine whether responses were time-locked to the optogenetic stimulus through a statistical threshold. The spiking activity of these neurons (average spikes counted per trial) allowed quantification of neuronal activity as a function of μ LED drive current and, by assuming the neuron as a single point, the spatial extent of activation by a single μ LED. We have further elaborated on spike sorting within the results.

“Spike sorting was used to attribute the recorded electrophysiology signal to the neuronal firing activity of individual neurons (Fig. 3c right) allowing us to determine if neuronal responses were time-locked to the optogenetic stimulus and under what conditions. For the following analyses, only single unit activity is considered.”

Comment 6

Finally, how would the approach described here be scaled in the future, both in terms of device size (for work with larger animal models, a 2x2mm² device will not be useful) and in terms of achievable resolution (where in addition to the challenge of getting the light sources closer to the target region or finding other means of minimizing light spreading, see above, the number of μ LEDs will need to be scaled dramatically). For instance, keeping the same pitch on a 10x10mm² device would require a 25-fold increase in number of addressable LEDs and substantially larger InGaN on sapphire samples. Would this be practical at some point?

To demonstrate how our μ LED array-based optogenetic stimulation approach could be scaled, we integrated four μ LED arrays onto a flexible PCB arranged in a line, separated by 1 mm gaps (Supplementary Fig. 5).

Supplementary Figure 5: Flexible, chronically implantable device integrating 4 μ LED arrays arranged in a line, separated by approx. 1 mm gaps.

We fabricate the μ LED arrays at scale (220 arrays per 2" wafer) (Supplementary Fig. 1b) and our flip-chip bonding process allows several μ LED arrays to be packaged to a single flexible PCB, where the geometry could be custom-designed for a particular neuroscience experiment or animal model. This is a clear advantage of our packaging approach which allows a number of 10x10 μ LED array chips to be tiled such as the one demonstrated (Supplementary Fig. 5). While each 2x2 mm² chip is rigid, the PCB is flexible and could allow conformability over a larger curved/contoured surface area. In the 10x10 mm² example, we would integrate 25 μ LED arrays to a flexible PCB and interconnection scalability could be addressed by multiplexing circuitry on the PCB. We have expanded the discussion to re-iterate this advantage and refer to the new device shown.

“Furthermore, flexible, biocompatible packaging enabled chronic implantation in the mouse model and **scalability through potential integration of several μ LED arrays tiled on a single device (Supplementary Fig. 5). This could facilitate experimentation across brain regions or in larger animal models such as non-human primates. This potential for rapid scalability and device customisation**, along with robustly established stimulation parameters and in-vivo stability makes this an ideal platform for chronic studies into cortical function and the targeted transfer of information to the brain.”

Regarding minimising light spreading, we have discussed different approaches with the most promising being the optical interposer, which we had modelled (and since experimentally validated – see response to Reviewer 2, comment 4). These interposers are simple to fabricate at scale as they rely on standard silicon processing techniques. They can then be individually integrated to each μ LED array across a tiled device.

Reviewer #2 (Remarks to the Author):

In this manuscript, Greer et al. describe a microfabricated μ LED array designed for chronic implantation to interface with superficial cortical regions in mice. The authors detail the fabrication process, characterize optical transmission and heat generation during device operation, and validate functionality by demonstrating neuronal activation and behavioral discrimination experiments in the auditory cortex. While the study addresses an interesting technological area, I find significant shortcomings that substantially limit its novelty, impact, and overall contribution to the field. Consequently, I cannot support publication of this manuscript in its current form. The major concerns are detailed below:

Major concerns:

Comment 1

While the technology itself appears robust, the authors fail to adequately compare and contextualize their approach relative to recently published technologies, including the group's previous work. For example, the Utah optrode array reported by McAlinden et al. (2024) is likely more useful for cortical neuromodulation, as it separates the LED source from neural tissue and offers beam control capabilities. Pollmann et al. (2024) reported a system (SCOPE) using a similar cortical arraying idea for all-optical bidirectional neural interfacing, providing both recording and modulation with greater controllability. Shin et al. (2024) demonstrated a wireless operation solution. Taal et al. (2024) achieved an array with 1024 OLEDs, providing single-neuron resolution. These works, although cited in the manuscript, are not adequately contrasted or acknowledged.

Thank you for highlighting this point and these works, which represent significant contributions to the field of implantable optogenetic cortical stimulators.

Previously our group has published the Utah optrode array (UOA), a similar stimulation device based on a μ LED array with an array of penetrating glass optrodes allowing light delivery to deeper cortical regions (Clark et al. 2024; McAlinden et al. 2024). The UOA is targeted for the non-human primate model and is not suitable for the mouse model as it is too invasive due to the glass optrode damage. Reducing the dimension of the glass optrodes is extremely challenging in terms of the fabrication and mechanical properties and significantly affect light-coupling. Here, we approached this problem using a minimally-invasive surface stimulation approach, with no tissue penetration, and compact packaging suitable for chronic implantation in the mouse model. Furthermore, reduced pixel spacing compared with the UOA allows for denser stimulation coverage within single cortical regions of the mouse model. For example, the mouse auditory cortex is on the order of approximately 4 square millimetres in surface area, matching our device size.

The SCOPE system (Pollman et al. 2024) is an excellent study investigating the use of a μ LED array and single-photon avalanche diode (SPAD) array co-packaged onto a flexible CMOS probe for use as an implantable microscope/optogenetic stimulation platform. Their system can optically stimulate the cortical surface in a similarly minimally-invasive way. However, stimulation site spacing in the SCOPE device is significantly larger (>1 mm) than our spacing (200 μ m) and while they do perform an all-optical bidirectional interfacing experiment in the mouse model, this is done in anaesthetised, head-fixed mice as the device is too large to be implanted chronically. Furthermore, due to the relatively large spacing of stimulation sites compared with the mouse brain, they only activated 6 μ LEDs across two cortical regions. Our μ LED array can provide significantly denser stimulation coverage within a single cortical region, crucial for studying information transfer to the brain, which is becoming increasingly important as cortical sensory prostheses are developed.

The work carried out in Shin et al. 2024 is also an interesting approach where their device offers fully-implantable and wireless optogenetic control of the rodent cortex. The use of skull mounted LEDs for transcranial optogenetic stimulation utilising highly sensitive opsins allows for minimally-invasive operation. The trade-off is a reduction in spatial resolution such that dense stimulation coverage

within a cortical region would not be possible, likewise limiting this device in its ability to transfer sensory or other information to the brain.

Lastly the work carried out in Taal et al. 2024, utilises an organic LED (OLED) penetrating probe with an impressive stimulation site density and pixel-count, able to selectively and spatially target individual neurons. They also provide detailed analyses of electrophysiology data with different opsins under optogenetic stimulation with two wavelengths. Due to the integration of OLEDs on a penetrating shank, the device does not offer the minimally-invasive spatial surface stimulation capabilities of our μ LED arrays. Furthermore, OLEDs are significantly less efficient and have not been shown to achieve robust neuronal and behavioural responses as cortical surface stimulators, where a higher brightness is required due to the increased distance between the optical source and target neurons. The device presented in Taal et al. 2024 was tested in anaesthetised, head-fixed mice and is not chronically-implantable, and the study presents limited quantification of the device lifetime in-vivo, an important consideration for OLEDs which are inherently less stable than their inorganic counterparts. Consequently, the system has not been chronically implanted or demonstrated in behaving animals.

Thank you for highlighting these papers, we have enhanced our literature review within the introduction, focussing on the key points differentiating our device from the current literature.

“Two-dimensional arrays of IILEDs offer fully implantable optogenetic stimulation coverage across the surface of the cortex. Previously our group demonstrated the Utah optrode array (UOA), a IILED array with 181 sites, delivering light through penetrating optrodes (Clark et al. 2024) to deeper-lying structures of the non-human primate (NHP) cortex. Other groups have also demonstrated that this optoelectronic approach is viable. For example, Pollmann et al. (Pollmann et al. 2024) developed a IILED array coupled to a single-photon avalanche diode (SPAD) array capable of spatiotemporal optogenetic stimulation and fluorescence monitoring, with relatively large IILED dimensions and spacing, optimised for imaging performance requiring uniform illumination. Chronically implantable LED arrays were also demonstrated on the surface of the NHP cortex (Rajalingham et al. 2021; Komatsu et al. 2017; Ohta et al. 2021), where stimulation site spacing and device size limited the application to NHPs. The technology has been adapted for rodent studies, demonstrating site-specific activation in the cortex (Ji et al. 2018; Kwon et al. 2013; Lee et al. 2018) with higher density devices being produced using organic LED (OLED) technology for in-vitro studies (Steude et al. 2016). The transfer of information to the mouse cortex requires high-density IILED arrays with spacing of a few hundred microns whilst covering significant tissue areas (up to 4 mm²) (Ceballo et al. 2019). Previous IILED arrays match the stimulation site density and spacing of our device (Sekiguchi et al. 2022) and have been coupled to arrays of electrodes (ECoG) integrating electrophysiology recording (Gu et al. 2024). While both the density and scale sufficient for high-resolution cortical surface stimulation has been achieved, chronic implantation in mice and simultaneous demonstration of robust neuronal and behavioural responses remains an open challenge for the field (Supplementary Table 1).”

We have included Supplementary Table 1, which compares our work to similar chronically-implantable μ LED/LED cortical stimulator approaches. This comparison is limited to studies demonstrating chronic implantability in-vivo.

Reference	Pixel size	No. pixels	Pixel pitch	Stimulation area	Power density dynamic range on brain surface (mW mm ⁻²)	Animal model
Pollmann et al. 2024	0.2 x 0.1 mm ²	48	0.6 – 1.25 mm	6.4 x 7.8 mm ²	70	NHP
Rajalingham et al. 2021	0.5 x 0.5 mm ²	24	1 mm	5 x 5 mm ²	56	NHP
Shin et al. 2024	1.6 x 1.6 mm ²	4	>1 mm	>4 mm ²	-	Mouse (transcranial)
Komatsu et al. 2017	0.33 x 0.33 mm ²	8	4 mm	8 x 16 mm ²	-	NHP
Ohta et al. 2021	0.35 x 0.28 mm ²	48	1 mm	9 x 6.5 mm ²	>10	NHP
Lee et al. 2018	0.05 x 0.05 mm ²	12	0.5 mm	2 x 1.5 mm ²	>10	Rat
This work	0.04 x 0.04 mm ²	100	0.2 mm	2 x 2 mm ²	>50	Mouse

Supplementary table 1: Comparison of chronically implantable μ LED/LED array cortical surface stimulators

Comment 2

The presented μ LED array is proposed as a tool for behavioral neuroscience; however, existing methods such as holographic optogenetics (see Adesnik & Abdeladim, 2021), multi-fiber approaches (e.g., Sych et al., 2019), and single-photon scanning optogenetics (e.g., Pinto et al., 2019) are not discussed. These systems, could enable experiments with much higher spatiotemporal resolution in the same neuronal ensembles, over extended periods, and with far less concern about off-target heating effects (at least in head-fixed setups). Additionally, the complexity of clean-room fabrication processes severely limits accessibility and ease of adoption by typical neuroscience laboratories, especially without clear practical advantages demonstrated by the current study.

Existing techniques for spatiotemporal optogenetic stimulation of the cortex commonly utilise external light sources, such as laser scanning including single-photon (Pinto et al. 2019) and two-photon (Adesnik et al. 2021) optogenetics, and digital light projector (DLP) approaches (Ceballo et al. 2019). These have facilitated advanced neuroscience studies into cortical function, but rely on bulky optical setups, unsuitable for chronic implantation and difficult to scale across multiple animals. Multifibre approaches allow can allow for good spatiotemporal resolution (Tian et al. 2024) and the potential for chronic implantability (Sych et al. 2019). With μ LED devices, stimulation site density can scale significantly as well as offering the potential for tether-free behaviour. With the advantage of implantability comes the trade-off of significantly reduced optical source-tissue proximity, highlighting concerns regarding tissue heating and device longevity, both of which we address in the manuscript. In head-fixed setups, the advantage of the aforementioned techniques such as DLP stimulation probably outweigh that of the μ LED arrays, however there is a clear advantage moving to chronically implantable devices where freely-behaving experiments become readily possible as we demonstrate here. Our μ LED arrays are chronically implantable and offer “plug-and-play” capability, significantly reducing the work overhead on the experimenter and maximising experiment time.

The introduction has been updated to include a broader review into existing techniques and highlight the benefits and trade-offs of using the implantable μ LED devices.

“Optical stimulation of the cortex is commonly achieved using external light sources, such as laser scanning single-photon (Pinto et al. 2019) / two-photon systems (Adesnik et al. 2021), or digital light projectors (DLPs) (Ceballo et al. 2019). These systems are capable of achieving very high spatial resolution. However, the associated optical components and electronic control systems preclude chronic implantation, hindering translation to experiments on freely-behaving animals. Fibre bundles can deliver high-density spatiotemporal optical stimulation within a cortical region (Tian et al. 2024) with the possibility of chronic implantation and use in freely-behaving animals

(Sych et al. 2019), but still require complex bench-top optical systems that preclude further development towards wireless implementation. For that, implantable optoelectronic solutions are needed. Micro-LED (light-emitting diode) (μ LED) probes offer implantable, spatiotemporal optogenetic control over neuronal populations, with the possibility of very high stimulation site density (Taal et al. 2023; Voroslakos et al. 2022; Wu et al. 2015; McAlinden et al. 2015) and fully-wireless options (Park et al. 2015; Shin et al. 2024). Chronic implantation of the optical source necessitates robust encapsulation strategies, addressing challenges such as biocompatibility, long-term stability (Caldwell et al. 2020) and thermal effects (Kim et al. 2022).”

The fabrication and integration process of these devices is complex, requiring significant experience and access to a semiconductor cleanroom facility. However, we fabricate the μ LED array chips at scale (220 arrays per 2” wafer, 4-5 wafers per run) (Supplementary Fig. 1b) meaning fabrication runs need only be carried out very occasionally and devices can then be packaged and distributed to neuroscience laboratories. The packaging process connects μ LED arrays to flexible PCBs and is highly scalable and easily customisable (see response to Reviewer 1 comment 6, where we rapidly developed a new device integrating 4 μ LED arrays). It uses integration equipment (gold bump bonding and thermosonic flip-chip bonding) that is commonly available, both at Universities and as a commercial service. The optimised fabrication and packaging process, simple surgical steps (Supplementary Fig. 12a), as well as robustly established stimulation parameters, and simple electronic driver circuitry (Supplementary Fig. 9) should make these relatively inexpensive when compared with holographic, multi-photon techniques and allow them to be readily adoptable by neuroscience laboratories.

We thank the reviewer for their comments highlighting where the introduction and literature review could be improved and hope we have addressed the concerns.

Comment 3

The authors claim that the potential advantage of this system lies in its use with freely behaving animals (albeit at the cost of spatial resolution and deep-layer access). However, this supposed advantage is unsubstantiated in the current work, as it is not demonstrated experimentally in the paper.

We have since carried out a series of similar behavioural discrimination experiments in freely-behaving mice using the chronically implantable μ LED array systems, which have been added to the manuscript as new short subsections (“**Behavioural discrimination experiments in freely-moving mice**” in Results and “**Freely-behaving experimental setup and procedure**” in Method), figures (**Fig. 6, Supplementary Fig. 14, Supplementary Fig. 15**), table (**Supplementary Table 4**) and video (**Supplementary Movie 2** which replaces the original). The experimental paradigm is similar to that already reported, where the mouse should discriminate between a go and no-go stimulus which are presented as 3x3 illumination patterns on distinct regions of the auditory cortex. In this experiment, the mouse is not head-fixed and able to move freely in a custom-designed chamber. The μ LED array is driven through a flexible cable which is mounted on a holder so there is minimum tension. Similar to before there are a series of trials, randomly assigned as go or no-go. The mouse decides when to start a trial by inserting its snout into a fork to interrupt an IR beam. For go trials the mouse should remain in the fork and receive a water reward, and for no-go trials, the mouse should leave the fork to avoid a penalty. We demonstrated four freely-behaving mice were able to learn this task at similar performance to the head-fixed behavioural study, thus demonstrating experimentally the discussed advantage of our approach. A full detailed description of the experimental methods and results are given in the new manuscript as dedicated subsections and figures not shown in this document.

The introduction has been updated as follows: “Lastly, chronically implanted μ LED arrays provided spatially confined optogenetic stimulation within the auditory cortex, and we observed robust learning in behavioural discrimination experiments, **in both head-fixed and freely-behaving mice.**”

The Discussion has also been updated as follows: “Chronic implantation of the μ LED arrays allowed repeated optogenetic stimulation of the same neuronal ensembles over several days, **as well as experiments in freely-behaving animals**, with a relatively straightforward surgical procedure.” **“Freely-behaving experiments instead used AAV vectors encoding ChRmine under the control of the CaMKII α promoter, similarly targeting pyramidal neurons, and providing strong photocurrent for a broad excitation spectrum including the wavelength of our iLEDs (Marshel, 2019).”**

The “Animals” subsection of Method has been updated as follows: **“For freely-behaving experiments only, mice were instead transfected with AAV vectors encoding ChRmine under the control of the CaMKII α promoter which were injected to drive expression in pyramidal neurons of the auditory cortex (pAAV-CamKII α -ChRmine-mScarlet-Kv2.1-WPRE (AAV8). Addgene number: 130991). ChRmine’s activation spectrum is red-shifted but it is broad with less than 50% reduction of elicited current at the wavelength of the iLED arrays ($\lambda = 450$ nm) (Marshel, 2019). Since ChRmine has much higher photocurrents than ChR2, blue light illumination was sufficient to elicit robust enough neuronal activation for behaviour.”**

Comment 4

The Lambertian radiation pattern of micro-LEDs is generally not precise. The spatial separation between the neuronal ensembles and the LEDs—introduced by the packaging process—further worsens this issue. Thus, the spatial resolution is intrinsically limited even if the LED array is made with a high density of emitters. The authors propose a beam-shaping method in supplementary information, but this approach is not experimentally validated.

Initially we proposed an idea to use an opaque interposer positioned in front of the device sapphire surface, with small holes aligned to the pixels, and used Monte Carlo modelling to demonstrate a reduction in optical spread, therefore improving spatial resolution. We have since validated this approach experimentally using an 80 μ m thick silicon interposer with 80 μ m diameter holes which we aligned to the μ LEDs using flip-chip bonding. We imaged the emission profile of both a device with, and without, the interposer submerged in fluorescein (Supplementary Fig. 16), and examined the reduction in the width of the illuminated region at a depth of 150 μ m (Supplementary Fig. 16a). We have also updated the modelling so it is more consistent with our previous analyses (Supplementary Fig. 16c). We have expanded the discussion further to also discuss the trade-offs with this approach.

“Using our optical models, we investigated if the spatial resolution could be improved through light-shaping. We modelled **and experimentally validated an optical interposer, which collimates the light using small vias fabricated into an otherwise opaque material (Supplementary Fig. 16). **Incorporating this interposer could reduce the illuminated width to approximately 500 μ m. The trade-off is optical losses in the silicon resulting in less light penetrating the tissue. The optical losses can be kept to a minimum with a 50 μ m thick interposer and 200 μ m diameter optical vias, which can deliver an equivalent optical power to the brain if the iLED output power is increased by a factor of approximately 4-5. This is possible given the low-power operation of our iLEDs; however, less power could also be used to target shallower depths and improve the spatial resolution further. The slight improvement in spatial resolution indicates that the dominant contribution arises from light scattering in tissue. To confirm this, we modelled a perfectly collimated light source (as can be implemented in optical systems, such as digital light projectors - DLPs) incident on a 200 μ m diameter area of brain tissue (Supplementary Fig. 16d). The illuminated volume and resolution is similar to the interposer models, demonstrating light scattering is indeed the dominant factor limiting any non-penetrating, optical cortical surface stimulation technology.”****

We have also added the experimental validation of the optical interposer in the methods:

“We bonded a thin (80 μ m) silicon interposer with 80 μ m diameter holes onto the sapphire surface of a packaged device to investigate improving spatial resolution through collimation

(Supplementary Fig. 16). Integration of the interposer and hole alignment to pixels was achieved using flip-chip bonding. Emission profiles of devices with and without the interposer were imaged with an SLR camera in 10 I.LM fluorescein solution (Supplementary Fig. 16a left). To compare the lateral light spread, raw green pixel data was plotted (Supplementary Fig. 16a right, blue lines - error bars computed from different shutter integration times) demonstrating a significant reduction in light spread with the interposer. Both images were compared with Monte Carlo models of equivalent geometry, modelling fluorescence by taking a cut through detectors at equivalent distance from the device (Zemax OpticStudio 2021) (Supplementary Fig. 16a right, orange lines). The experimentally validated optical interposer was modelled in brain tissue at equivalent delivered optical power to Fig. 4 (Supplementary Fig. 16c).”

A dedicated supplementary figure has been added:

Supplementary Figure 16: Optical interposer (a) Emission profile of I.LLED with and without collimating interposer taken with SLR camera in fluorescein (left). Emission profile at distance of 150 I.Lm from sapphire surface compared with cuts at equivalent distance from an optical model of fluorescence, with equivalent geometry. **(b)** Schematic of optical model used to evaluate light propagation in brain tissue with collimating interposer. **(c)** Modelled optical profile for different optical interposer hole diameters; interposer thickness is 50 I.Lm and cortical window is not present. Contours show 1 mW mm⁻² threshold corresponding to equivalent optical power delivered to brain tissue as Fig. 4. I.LLED output power is scaled to deliver equivalent power to brain across models.

Comment 5

In terms of optical efficiency, the authors demonstrate that about 90% of the electrical power is converted to heat. Although the modeling results suggest the resulting temperature rise is physiologically safe, this claim must be validated experimentally in vivo. How was this modeled in air, and what conditions were assumed in the brain tissue model? Is a 3–5 mA drive current sufficient to recruit neuronal activation, given that the authors used 10 mA in their in vivo electrophysiology experiments (which would correspond to roughly a one-to two-fold higher temperature increase)? Additionally, the absorption of blue light by tissue also contributes to heat generation, and it is unclear why this was not taken into account.

Thermal modelling of a heat source equivalent to the i.iLED array operating at currents of 3 mA and 5 mA, with brain tissue modelled as water, was used to confirm safe thermal effects in-vivo. A direct thermal measurement could not be made in-vivo, as the uLED array was covered with dental cement to fix it in place. However, the thermal model of the i.iLED array with chronic device packaging was validated using infrared imaging in air (Fig. 2a, Supplementary Fig. 6, Supplementary Fig. 7a). This in turn gave us confidence that the model of the device itself was robust and could be deployed with modelled brain tissue to estimate in-vivo thermal effects (Supplementary Fig. 7b). In-vivo thermal measurements were not made here. However, we recently published work on the Utah optrode array (UOA), where a similar thermal model was compared with thermal measurements during device operation in-vivo in a non-human primate (NHP) (see McAlinden et al. 2024). This validated the UOA thermal model, which we extended and applied here. We have updated text in Results to clarify how the thermal model was validated and to reference the recent UOA publication.

“To firstly verify the accuracy of the modelled device, the sapphire surface of the μ LED array was imaged using an IR camera in air (Supplementary Fig. 6), which agreed closely with the modelled output (Fig. 2a). These experiments, along with previous work demonstrating the accuracy of this model architecture with in-vivo IR imaging (McAlinden et al. 2024), gave us confidence in our thermal models to accurately represent experimental conditions.”

We used 3 mA and 5 mA in thermal models (for 2x2 and 3x3 this was per i.iLED) as this corresponds to the maximum current range needed to elicit both robust neuronal or behavioural responses. In our electrophysiology experiments, we used a device which experienced some degradation in optical performance therefore requiring a drive current of between 8 mA and 10.4 mA to achieve threshold cortical irradiance ($\sim 5 \text{ mW mm}^{-2}$) instead of $\sim 5 \text{ mA}$ which we demonstrate in stable devices (Supplementary Fig. 1d). We have since addressed and clarified this (see response to comment 9). Thank you for highlighting this. Thermal models are also based on direct implantation of the device on the cortical surface, whereas for our electrophysiology experiment there is a cortical window, which would act as a heat spreader. Furthermore, we conducted a control experiment to ensure the elevated temperature produced by the degraded device did not elicit unwanted neuronal responses.

Blue light absorption in tissue can contribute to significant temperature increases if the power density is very high (several milliwatts or tens of milliwatts over a small surface area); this is typically only observed in optogenetic/imaging systems using high power laser light delivered through optical fibres typically 100-200 μm in diameter, resulting in very high irradiance localised at the fibre tip (Owen et al. 2019). For our i.iLED arrays, peak cortical irradiance is significantly lower (for example when the device is in direct contact with tissue, we do not exceed 0.7 mW per i.iLED over an illuminated area with 270 μm diameter). As a result, we assume the radiant efficiency of the i.iLED is the dominant factor influencing thermal effects on the brain.

Comment 6

The flexibility claims of the device are difficult to reconcile with the use of a 150- μm sapphire substrate. The authors must clarify this discrepancy or provide additional experimental support.

Thank you for highlighting this, the i.iLED array (the stimulation area of the device containing the 100 i.iLEDs located over approximately a $2 \times 2 \text{ mm}^2$ region) is indeed rigid. This is then packaged onto a 100 μm thin flexible PCB which provides flexibility in the positioning of the device during surgical implantation. This rigidity is due to the GaN-on-sapphire fabrication process used to produce the i.iLED array chips (Supplementary Fig. 1). The sapphire substrate could be removed through laser lift-off techniques (see response to Reviewer 1, comment 1 in the context of reducing source-tissue proximity to improve resolution). However, we elected to keep the sapphire as it protects the i.iLEDs (important for the chronic implantation) and is lattice-matched to the GaN material, helping to maximise the quantum efficiency of the devices. We chose chips of approximately $2 \times 2 \text{ mm}^2$ to provide stimulation

coverage over the entire auditory cortex, or other cortical regions of the mouse model. At this millimetre scale, the curvature of the brain is compensated by applying slight pressure on the μ LED array when glued on the skull. This procedure is also widely used in cranial window surgeries for two-photon imaging. It forces the brain to plastically conform to the surface array, such that brain to μ LED distance is minimised for all μ LEDs across the device. We have since carried out histology analysis and this plastic adaptation of the brain is visible on histological slices (Supplementary Fig. 2). As shown for cranial window surgeries (Holtmaat et al, 2009), the craniotomy and placement of a glass surface on the brain leads to an immune response persisting more than 10 days after surgery. We observe a similar low-level response with our chronically implanted μ LED arrays (Supplementary Fig. 2g). For larger animal models, conformability could be achieved through tiling of several μ LED arrays to a single flexible PCB which we demonstrate (see response to Reviewer 1, comment 6).

The Results section has now been updated to clarify this.

“The μ LED arrays were packaged to a lightweight, flexible device (Fig. 1c) enabling chronic implantation in the mouse model (Fig. 1d). However, **the stimulation area of the μ LED array remains rigid, constrained by the GaN-on-sapphire fabrication process; the brain locally adapts to this constraint in the same way as it does as for cranial window implantations (Holtmaat et al. 2009).**”

“Improved consistency between experiments is possible with removal of the cortical window and with our chronically implantable devices, **where the supporting polyimide PCB is flexible, making it easier to fix into position.**”

Comment 7

Blue light penetrates poorly in living brain tissue. Therefore, the authors must validate the light transmission profile predicted by the Monte Carlo simulation with experimental measurements in actual brain tissue.

Blue light does not penetrate as well through tissue as red or infrared light, due to the wavelength dependence of light scattering in the tissue. Nevertheless, several studies, including ours, confirm blue light projected onto the cortical surface with sufficient irradiance can penetrate to deeper cortical layers, resulting in robust neuronal responses in ChR2-expressing mice (Ceballos et al. 2019; Abbasi et al. 2023; Kwon et al. 2013; Gu et al. 2024). Opsin sensitivity is a crucial factor here, with transcranial optogenetic inhibition being possible with blue light with stGtACR2 (Shin et al. 2024). Our optical models utilise robust scattering and absorption coefficients from highly-cited literature (Yaroslavsky et al. 2002), backed up by experiments in living mouse brain tissue (Aravanis et al. 2007). Furthermore, we use industry standard Monte Carlo software (Zemax OpticStudio 2021) for all our optical modelling.

Nevertheless, consideration of specific device geometry used in models should be validated experimentally and we have since carried out an experiment measuring light intensity in air from our μ LED emission, to compare with an equivalent optical model. This method was modified from (Yona et al. 2016) where they measured light from an optical fibre through mouse brain slices. Here our intention was to compare our model with experimental data in air, then use robust scattering and absorption coefficients from the literature to translate the model for in vivo predictions. Supplementary Fig. 3d shows the very close agreement between our model and the measurements. It should also be noted that we observed good agreement between the imaged optical profile in fluorescein and our equivalent model (see response to comment 4). We have added detail to Method for this new measurement and added an additional figure (Supplementary Fig. 3d).

“To validate the optical modelling experimentally, a pinhole (100 μ m in diameter, 40 μ m thick) is positioned in front of a photodetector (Supplementary Fig. 3d). The pinhole was positioned as close to the detector as possible to ensure that all light that passes through the pinhole is collected by the detector. The pinhole-photodetector is then translated through the μ LED emission profile measuring the light intensity at each point. The measurement is repeated at various μ LED-

pinhole distances. The light measurement at each point allows for the beam profile reconstruction with a resolution of approximately the pinhole size. This method is modified from (Yona et al. 2016).”

Supplementary Figure 3 (d) Emission profile of single μ LED through air, measured using a PMT detector with pinhole maneuvered through emission profile (100 μ m measurement spacing) (left). Emission profile at different depths compared with cuts at equivalent distance from an optical model of device in air (right).

Comment 8

Given the relatively large size of the head plate shown in Figure 1c, it is unclear whether this surface-interfacing arrangement will outperform the authors' previous optrode array approach. That earlier approach avoids heat generation at the neural interface and offers beam-regulating (beam-shaping) capabilities. The advantages of the new surface array over the established optrode array are not convincingly demonstrated.

Thank you for highlighting this. All our mice are implanted with a headplate, including for the new freely-behaving experiments, to facilitate surgery and make it easier to plug and unplug the device cable. In principle, the headplate could be removed entirely for freely-behaving experiments, however it is lightweight and small enough that it does not impede animal behaviour (headplate weighs <1 gram and device <0.3 grams; headplate extends approximately 5 mm). If the headplate were removed, standard neuroscience techniques could be employed to pacify the animal for insertion of the device cable, rather than head clamping. As discussed in response to comment 1, we have since improved our literature review to clarify how this new μ LED array differs, both from our previous work with the UOA which is designed for NHPs and is too invasive for the mouse model (Clark et al. 2024), and to other similar minimally-invasive optical cortical surface interfaces.

Comment 9

It is unclear why the authors used a 20 Hz stimulation with a 50% duty cycle for their electrophysiology experiments, as this protocol is likely to induce a sustained cation influx through ChR2 and could result in multiple spikes per neuron per 25 ms. This protocol also generates substantial heat. In Figure 3c, the authors identify single-unit activity among multiple spikes, but it is not clear what these spikes are time-locked to – the onset of the stimulation or the offset of the stimulation. The authors also repeatedly mention using a “low” drive current; however, in their experiments they used 8.0 mA and 10.4 mA, which are relatively high currents and contradict the claim of a low-current drive. Additionally, it is unclear how the electrophysiological data were filtered and preprocessed, since one would expect evoked local field potentials given this optogenetic protocol, yet none are shown in their traces.

The choice of 20 Hz stimulation with 50% duty cycle used for electrophysiology and behaviour was chosen based on previous use of these parameters in the Bathellier lab (Ceballos et al. 2019; Bagur et al. 2025), based on observations that pulsed stimulation drives overall more action potentials than continuous illumination (Glickman et al, 2023) and would therefore be more salient for the animal. Relatively fast pulses (20 Hz) target fast-spiking auditory cortex neurons with a 50% duty cycle maintaining a strong level of neuronal firing across the duration of activation (500 ms), aiming to obtain overall strong spiking responses easily perceived by mice. With this protocol, multiple spikes

are likely induced per 25 ms pulse which are not necessarily precisely timed. Shorter pulses would lead to more precise timing and could be used to explore the role of spike timings in the neural code. However, this would require accurate calibration experiments to construct different spike patterns and verify them, and use of faster opsins than ChR2, which is outwith the scope of this study. Indeed, reducing the duty cycle would further mitigate thermal effects (see response to Reviewer 3, comment 4) however, we found that at the present pulse rate and duty cycle, thermal effects were acceptable within the range of μ LED drive currents needed to achieve consistent neuronal activation (Fig. 2b). The μ LED array coupled to our electronic driver system allows temporal patterning of stimulation across a broad range of pulse frequencies and duty cycles, with microsecond precision. This could allow us to explore the role of temporal features in the neural coding in addition to the spatial features made accessible with the μ LED array. The discussion has been updated to propose this direction as further work.

“This study was limited to spatial patterning with fixed timing (20 Hz, 50% duty cycle); more sophisticated temporal patterning of the μ LED array could also allow exploration of the involvement of temporal features in the neural code of the cortex.”

Thank you for highlighting the drive current values which seem relatively high. In the electrophysiology experiments, the device we used experienced degradation in the optical performance, meaning a higher than normal drive current (up to 10.4 mA) was needed to drive ~ 5 mW mm⁻² and achieve consistent neuronal responses. With stable devices ~ 5 mA is needed to achieve ~ 5 mW mm⁻². This degradation had already been discussed however the text in Results has been updated to make this clearer.

“This spike rate was achieved at moderate cortical surface irradiances between 4.0 ± 0.1 mW mm⁻² and 5.3 ± 0.3 mW mm⁻², corresponding to a drive current of between 8.0 mA and 10.4 mA. These drive currents are slightly higher than expected due to the reduction in optical performance observed with this device. In stable devices, we demonstrate a consistent optical output of 5 mW mm⁻² at the cortical window/tissue interface for 5 mA drive current (Supplementary Fig. 1d).”

Furthermore, these thresholds assume the use of a cortical window, which we only used in acute electrophysiology experiments. Direct implantation results in the same amount of optical power delivered to the cortical surface but with smaller illuminated area, increasing the cortical irradiance (Supplementary Fig. 1d). We investigated through optical modelling and note the same depth of penetration of light with less delivered power for when the device is directly implanted (Fig. 4f), meaning even lower drive currents could achieve similar neuronal responses.

The electrophysiology data was analysed in the context of single-unit spiking activity to allow us to infer a time-locked neuronal response to the stimulus from which threshold irradiance levels and spatial resolution could be determined. The data was pre-processed for spike sorting (and multi-unit analysis, Supplementary Fig. 17) using a bandpass filter (300 Hz to 5.5 kHz). High pass filtering will suppress low frequency signals such as LFP or electrode drift which can impair spike detection. To eliminate shared activity and common-mode noise from the signal, we subtracted the average signal across all recording channels from each channel before single unit detection. This allows for removal of mouse motion and photoelectric artefact that could have occurred during the recording. We thank the reviewer for the remark and have included raw/unfiltered example of recorded signals in Supplementary Fig. 10b. Moreover, for clarity, the methods section has been updated as follows:

“Raw electrophysiology data was captured with Intan RHX and filtered using a bandpass filter with cutoffs at 300 Hz and 5.5 kHz followed by a common average reference subtraction across recording channels (Supplementary Fig S10b). Pre-processed electrophysiology data was then automatically spike sorted using the Spike Interface package and Kilosort 2.5. Manual curation was carried out with Phy to remove incorrectly identified clusters. Stimulation artefacts were also manually identified and excluded from analysis.”

Consequently, we did not analyse local field potential activity, however, we have added an additional non-pre-processed waveform (equivalent to Fig. 3c before filtering) as Supplementary Fig. 10b.

Supplementary Figure 10 (b) **Electrophysiology waveforms** corresponding to same μ LED and trial as Fig. 3c. **Top: raw trace; middle: raw trace after filtering with bandpass FIR filter (cutoffs 300 Hz and 5.5 kHz); bottom: raw trace after common median reference subtracted (no filtering).**

Comment 10

The experiment in Figure 3f demonstrating spatial resolution is a valuable addition. However, it is not justified to compare the neuronal responses directly to the Monte Carlo simulation results to derive an “activation threshold.” This approach ignores important morphological considerations of pyramidal neurons: activation of ChR2 in neuronal processes (such as apical dendrites) can contribute to action potential generation at cell bodies, especially since the viral construct used here does not have subcellular specificity. Consequently, the large apparent depth of neuronal activation shown in Figure 4g is likely due to the activation of ChR2 in apical dendrites in the superficial layers. Thus, it is very misleading that the authors’ figure suggests ~ 1 mW of 450 nm light can reach approximately 500 μ m deep into layer 4, as this depiction overlooks the contribution of dendritic activation in superficial layers.

In Figure 3, we use neuronal responses to derive an activation threshold by selecting a μ LED nearby the recording electrode array, and quantifying neuronal responses time-locked to the optogenetic stimulus as a function of cortical surface irradiance (Fig. 3d, Fig. 3e). We then extend this analysis by analysing across a row of μ LEDs organised orthogonally to the insertion trajectory of the recording probe, allowing us to demonstrate spatial resolution (Fig. 3F). In Figure 4, we then introduce our Monte Carlo optical models as a means to separately establish spatial resolution, to compare with that obtained from electrophysiology recordings. The optical model shown for example in Fig. 4c was developed independently and not inferred from electrophysiology recordings, other than the optical power delivered in the simulation being equivalent to that used in the experiment, to ensure a valid comparison.

With the electrophysiology response, we determine a spatial extent of activation by illuminating different sites across the array and measuring neuronal response. This is used to experimentally validate the optical modelling, and we use a naïve approach where we assume activation occurs only in tissue volume where the irradiance is above 1 mW mm^{-2} (shown by the contours). It is correct that

the mouse model we use (Emx-IRES-CRE x Ai27d) expresses the opsin in the dendrites and that measured neuronal activations could be due to dendritic or secondary network effects. Later we discuss through multi-unit analysis (Supplementary Fig. 17) that we see neuronal responses on electrodes within layer 5 which is beyond the deepest extent of the 1 mW mm⁻² level shown in Fig. 4c, suggesting dendritic or second-order activations. We have improved the text in Results to better clarify the role of optical modelling here, thank you for highlighting this.

“Experimental and modelled estimations of the width of activation are in reasonable agreement (Fig. 4d). This comparison is an approximation, **ignoring dendritic contributions and second-order network effects**, and implies that the optical models could be used to inform device optimisations to further improve spatial resolution.”

Comment 11

The behavioral experiments presented here do not clearly showcase the capabilities of the technology. They still require a head-fixed setup, a restraint tube, and a medial forebrain bundle (MFB) stimulator to motivate the animals during the tasks. Again, the authors used a 20 Hz stimulation with a 50% duty cycle for these behavioral experiments, which compromises the inherently high temporal resolution advantage of optogenetics. In summary, the behavioral results do not demonstrate a clear benefit of the micro-LED array approach compared to existing optogenetic methodologies for similar experiments.

As discussed in response to comment 5, we have since carried out a series behavioural experiments under similar conditions to before but in freely-behaving mice. While head-fixation and the restraint tube are limitations with the previous experiments, medial forebrain bundle (MFB) stimulation does have advantages in terms of maintaining animal motivation over long timescales (Verdier et al. 2022). Nevertheless, with our new freely-behaving experiments, we use a water reward which is simpler to set up, not requiring a precise surgery to implant a stimulator probe into the MFB. Animals are water-deprived and voluntarily start a trial by entering an infrared beam, where they are delivered a water reward through a lickport upon successful trial completion (Fig. 6a, Fig. 6b, Supplementary Fig 14b). As described in response to comment 9, the 20 Hz, 50% duty cycle stimulation protocol has been used extensively to drive behavioural responses from optogenetic stimulation of the auditory cortex within the Bathellier lab (Ceballo et al. 2019; Bagur et al. 2025) and was a natural starting point in this study. The choice of this stimulation protocol does not reflect the technical limitations of the device but rather the choice of optimising behavioural outcomes in proof of concept experiments by producing strong and reliable responses.

Reviewer #3 (Remarks to the Author):

General comment

Greer et al. present a μ LED-based optogenetic stimulation system designed for chronic cortical surface implantation in mice. This study is notable for its comprehensive in vivo validation, integrating acute electrophysiology to assess neuronal activation and chronic implantation to evaluate long-term functionality and behavioral effects. The dual experimental approach strengthens the study's relevance in neural interface research. While the μ LED array is designed for high spatial resolution, the reported optical spread raises questions about the precision of neuronal targeting. Additionally, electrophysiological recordings were limited to a single probe insertion, making it unclear whether the findings generalize across the full μ LED array. Lastly, despite being intended for chronic use, the study does not include direct biocompatibility testing such as histological analysis of immune response or tissue damage. Addressing these aspects would further strengthen the validation of the system. Therefore, the following points should be considered before publication.

Major comment

Comment 1

As highlighted in the abstract and title, one of the main claimed advantages of this paper is the device's suitability for chronic implantation due to its flexibility. However, the device presented in the manuscript utilizes a sapphire wafer with a stated thickness of 150 μ m for the implant device. Given the rigidity of sapphire, there is concern whether sufficient flexibility has truly been achieved to justify this claim.

Thank you for highlighting this point. The device stimulation region is rigid with the flexibility claim arising from the flexible packaging, which makes the device lightweight and easy to manipulate and position during surgery for chronic implantation. We have clarified this point in response to Reviewer 2, comment 6 where we updated the text to better explain that the μ LED array itself is still rigid with the packaging being flexible.

Comment 2

Additionally, given the expected high Young's modulus of the described sapphire-based device, it likely exhibits limited flexibility, resulting in poor conformal contact with the brain's surface. Given this rigidity, the presence of wrinkles or folding—necessary for flexibility—seems unlikely. Therefore, it raises concerns regarding the precision and accuracy when claiming targeted stimulation in specific brain regions. Clarification and supporting data on the device's ability to accurately target stimulation despite potential mechanical mismatch would be beneficial.

As mentioned previously, the stimulation region of the device which is in direct contact with the brain tissue is rigid, with the brain being very malleable in contrast. The contact area is approximately 2.2 x 2.2 mm² which is small compared to the surface of the mouse cortex. At this scale, the mouse cortical surface is almost flat on the dorsal part of the brain and only slightly curved. This curvature is conformable, a property that is used for cranial window surgeries over up to 5 mm with flat glass coverslips. Moreover, the absence of circumvolution in the mouse brain in comparison to non-human primates makes our device suitable for experimentation in the mouse. In primates, implantation could be more challenging but at the millimetre scale, cortical surface could be considered flatter than in the mouse. The mouse cortical surface is approximately flat and is plastic enough at this scale to make sure that all μ LEDs are at the same distance to the brain by applying slight pressure during the implantation. This plastic deformation of the tissue is evident when looking at histological sections of the brain after implantation (Supplementary Fig. 2a, Supplementary Fig. 2b) (see following response). We also explored in response to Reviewer 1, comment 6, how we could scale up the stimulation area given the constraints with μ LED array rigidity. Several μ LED arrays were packaged onto a single flexible PCB which could target regions across the entire cortical surface, with the assumption that each region is approximately flat at the scale of an individual μ LED array chip (Supplementary Fig. 5).

Comment 3

Similarly, due to the rigidity of the device, issues related to micro-motion after implantation are likely to arise. It is expected that prolonged implantation of this rigid device may lead to damage in the brain tissue. Although the authors have confirmed device functionality post-explantation, the manuscript lacks information regarding potential damage to the brain. Therefore, histological data demonstrating the degree of tissue damage following device implantation should be provided. Additionally, despite describing an implantable device, the authors have not included data regarding immune responses, which should also be addressed.

We have added a supplementary figure showing that the mouse brain conforms to the flat surface of the μ LED array with a small, but statistically significant immune response limited to layer 1 - as observed for cranial windows (Holtmaat et al, 2009).

Supplementary Figure 2: Histological analysis. (a) Coronal view of brain slice stained with DAPI and Iba-1, with auditory cortex on implanted side indicated. (b) Zoomed in view of implanted auditory cortex; μ LED array geometry shown as dashed outline; layer 1-2/3 boundary and depth of analysed region indicated by solid yellow lines. (c) Zoomed example of implanted region analysed to examine immune response (staining: DAPI + Iba-1). (d) Same region as (c) (DAPI + mScarlet). (e) Same region as (c) and (d) (DAPI + Iba-1 + mScarlet). (f) Similar example region for contralateral side (DAPI + Iba-1). (g) Immune response as number of Iba-1+ cells (microglia) per square millimetre of imaged tissue, comparing both implanted and contralateral sides across $n = 7$ coronal slices for 3 mice; non-parametric Wilcoxon matched-pairs signed rank test. Layer 1 was removed from both sides to exclude surface immune response from the analysis (bottom plot).

We have added methods text as follows:

“Tissue preparation: At the end of the experiments, mice were deeply anaesthetised with an overdose of ketamine/xylazine administered intraperitoneally (100 mg/kg and 10 mg/kg i.p.

respectively) and perfused transcardially with 1X phosphate-buffered saline (PBS1X) followed by 4% paraformaldehyde solution. Brains were extracted from the skull and kept immersed overnight in 4% paraformaldehyde solution at 4°C. 40 µm-thick coronal sections were obtained using a vibratome (Leica VT1200S).”

“Immunofluorescence: Slices underwent blocking using a 10% normal goat serum (NGS) (Abcam AB7481)-0.5% Triton X (Merck, 648462) in PBS1X solution for 1 hour at room temperature. Slices were then incubated with primary antibody rabbit anti-Iba-1 (1/500) (Abcam, ab178846) in a 3% NGS-PBS1X solution overnight at 4°C. 3% NGS in PBS1X was used as a negative control. The following day, slices were washed with PBS1X for 5 minutes 3 times prior to an incubation with secondary antibody goat anti-rabbit AlexaFluor 488 (1/500) (Abcam AB150089) for 2h at room temperature. DAPI (Invitrogen Hoechst 33342 H3570) (1/5000) was applied as a nuclear stain to reveal the general anatomy of the preparation. Slices were mounted using FluorSave reagent (Merck, 345789). Fluorescence images were acquired using Nikon A1HD25 confocal microscope and analysed using ImageJ.”

We have referenced to and quantified the degree of immune response in Results:

“Histological analysis demonstrates plastic deformation around the implant site, with the size of the affected region comparable to the geometry of the µLED array (Supplementary Fig. 2). Immune response was then quantified by examining microglial build-up from the surface through the auditory cortex (Supplementary Fig. 2g). This was statistically significant when comparing the Iba-1+ cell response from the implanted side (228 ± 24 per mm^2) to the non-implanted (contralateral) side (181 ± 18 per mm^2). Excluding layer 1 from the analysis shows no significant difference between the implanted side (200 ± 34 per mm^2) and contralateral side (133 ± 11 per mm^2), highlighting a small localised immune response, approximately limited to layer 1 of the mouse cortex.”

Comment 4

Further data related to temperature management appear necessary. Duty cycle and frequency can significantly vary depending on the experimental conditions employed. Additional data specifically demonstrating temperature variations under different experimental conditions, such as varying duty cycles and frequencies, would enhance the robustness of the results and the claims made in this study.

Thank you for highlighting this. We have since carried out simulations on COMSOL using the already established thermal model, with a range of pulse repetition rates (10, 20 and 50 Hz) and duty cycles (1%, 10%, 50%) common for optogenetic applications (Supplementary Fig. 7i). These are presented over a 500 ms window and we show the peak temperature reached at the brain-sapphire interface. For simplicity, we used a thermal power in the model equivalent to 3 mA (per µLED) as this is representative of the maximum current used when the device is directly implanted on the cortical surface; for behavioural experiments 10 mW mm^{-2} is achieved at <3 mA. These results show that duty cycle is the dominant factor influencing thermal effects, which is intuitive given this dictates the time the µLED is on over the duration of the stimulation protocol. The updated figure and caption are shown below.

Additional analysis demonstrating the effect of drive current on peak temperature increase has also been included (Supplementary Fig. 7j). The simulated currents correspond to typical cortical irradiance levels utilised in behavioural experiments, with 5 mW mm^{-2} per µLED giving a peak temperature increase of less than 1 °C on the cortical surface under the same experimental protocol.

Supplementary Figure 7 (i) Evaluation of modelled peak temperature increase on brain surface, over 500 ms stimulation time, across different pulse repetition rates and duty cycles; interpolation is used on colour map between 9 data points per pattern. Modelled thermal power corresponds 3 mA equivalent per I.LLED. (j) Modelled peak temperature increase as a function of cortical irradiance; equivalent currents per I.LLED are (0.19, 0.38, 1.02, 2.26) mA.

Comment 5

In the current experimental setup, the authors investigated temperature changes with only 10 pulses of LED activation. However, if prolonged, continuous operation occurs beyond these conditions, it is expected that temperature would continually increase. Therefore, it remains unclear whether the device can safely operate for extended periods, such as 100 seconds, without encountering significant thermal issues. Additional data evaluating the device under prolonged LED activation conditions should be provided.

Prolonged activation of the μ LED over for example 100 seconds would result in elevated temperature on the surface of the brain compared with the intermittent stimulation protocols we used during in-vivo behavioural experiments, where there were approx. 5 second gaps between 500 ms bursts of pulses. We modelled the particular case of 100 seconds of stimulation with the same experimental protocol used throughout the report (20 Hz, 50% duty cycle) with no gaps. Thermal equilibrium is reached after several seconds, with reasonable temperature increases being observed, for example for a single μ LED, we remain well below an estimated safe limit of 1 °C. Over this extended duration we would not want to operate a 3x3 pattern at 3 mA, therefore consideration of experimental protocol, drive current and number of active pixels is important and has been added as Supplementary Fig. 7k. Thank you for highlighting this.

Supplementary Figure 7 (k) Modelled temperature increase on brain surface over prolonged operation (100 seconds); modelled heat protocol (20 Hz, 50% duty cycle), with simulated thermal power equivalent to a drive current of 3 mA per μ LED.

Comment 6

In the presented lifetime data, the sequential activation of ten LEDs, each with a 1-minute interval between

activations, does not adequately reflect typical *in vivo* scenarios where multiple LEDs might operate simultaneously or with shorter intervals. Consequently, the reported lifetime results may not accurately represent practical usage conditions. Additionally, simultaneous use of multiple LEDs could significantly raise local temperatures, potentially decreasing device lifetime. A detailed discussion addressing this scenario and its implications on device performance and longevity is warranted.

Thank you for highlighting this point. Indeed, we only tested single pixel illumination as part of our accelerated testing experiments and due to cycling through a sequence of ten μ LEDs, each one was activated approximately once per minute. We expect that fields breaking down the encapsulation layer primarily contributes to device failure with local temperature increases as a secondary effect. To address both points, we conducted an additional accelerated testing experiment on a device submerged in PBS at approximately 60 °C (Supplementary Fig. 8d). A 3x3 pattern was illuminated at 2 mA per μ LED (total of 18 mA) in bursts of 10 pulses (20 Hz, 50% duty cycle) repeated every ~5 seconds, closely mimicking our behavioural experiments. For this device, 10 mW mm⁻² required ~2 mA per μ LED. As before, voltage was measured over the duration of the experiment demonstrating stability of the 3x3 pattern for ~10 days of continuous *in-vivo* operation. Since experiments are only conducted for 1-2 hours per day, we expect devices should remain stable over the course of chronic implantation and experimentation (approx. 1 month). We also explanted a device which was used for behavioural experiments (operating 3x3 patterns) and demonstrated minimal degradation compared with before implantation (Supplementary Fig. 8c).

Supplementary Figure 8 (d) **Normalised measured voltage during accelerated ageing for single 3x3 I.LLED pattern; experimental protocol same as in Fig. 2d with 2 mA per I.LLED. Mean and standard deviation of voltage measurements across 3 columns of 3 I.LLEDs in parallel.**

Comment 7

Following the above points, the manuscript does not adequately address the implications of the inability to simultaneously operate all LEDs. If simultaneous activation is restricted due to thermal or other limitations, this undermines the claim of effectively covering a large area. Thus, further discussion and evidence are required to substantiate how the device achieves broad-area coverage when limitations on simultaneous LED operation are considered.

Thank you for highlighting this. All μ LEDs on the device can be operated over the course of an experiment, but activation of all 100 μ LEDs simultaneously, at the drive currents and duty cycle reported here, would result in unacceptable temperature increases on the brain surface. However, this does not preclude spatiotemporal patterns to be produced over the full extent of the implanted array. We show activation of 9 μ LEDs, stimulating distinct sub-regions of the auditory cortex, can drive behavioural responses, and supplement this with a Monte Carlo optical model (Fig. 4g), showing broad activation at tested cortical surface irradiance levels assuming a 1 mW mm⁻² threshold for ChR2. Therefore, it is unlikely that simultaneous activation of all μ LEDs would be necessary for the intended application of this device to deliver spatially-confined optical stimulation. However, if this were not

the case, the electronic driver system does support activation of all μ LEDs but duty cycle and drive current should be proportionally reduced. Ideal operation would involve applying time-division multiplexing across rows of the array, thus allowing arbitrary patterns and mitigation of thermal effects (Supplementary Fig. 9d). The use of more sensitive opsins is an obvious approach to mitigate both thermal effects, and reduction device lifetime due higher field strengths, from driving a higher number of μ LEDs, and drive currents could be significantly reduced.

The discussion has been updated to reflect these points.

“Our μ LED arrays are able to drive above-threshold irradiances several hundreds of microns deep and with broad spatial coverage over a cortical region, although thermal implications must be considered when activating a large number of pixels simultaneously. Reliable neuronal responses and behavioural outcomes were achieved while maintaining low temperature increases at the implanted site.”

Comment 8

The study presents compelling evidence that μ LED-based optogenetic stimulation can modulate behavior through artificial perception. However, the exact duration of the behavioral training period is not clearly stated in the manuscript. Given that Figure 5(d) shows a learning curve spanning multiple sessions, it would be helpful to specify how many days or weeks the behavioral training lasted. Please clarify the total duration of the behavioral training in the Methods section, including the number of sessions per day and whether any criteria were used to determine when training was complete. This information will help readers better understand the timescale required for optogenetic learning and its potential for long-term applications.

Thank you for highlighting these points. When we refer to an experimental session, this is a set of trials carried out for a single mouse in one day. The time for each session has now been added to Supplementary Tables 2 and 3. We have also provided this information in the results section.

“Experiments were carried out over a number of sessions (1 session per day), each consisting of several hundred trials usually over a 1-2 hour learning period dependent on mouse performance (Supplementary Tables 2 and 3).”

For association experiments (*go* trials only), once the mice achieved sufficiently high accuracy (usually >80%) for a single session, they moved onto discrimination experiments (*go* and *no-go* trials). During discrimination experiments, generally the accuracy started and was maintained at a high level (~80% or greater). Thus, we terminated these experiments after a set number of trials and report the overall accuracy based on the 500 consecutive trials with the highest accuracy (Fig. 5f). We have updated the Results to clarify this.

“After learning the association task, defined as maintaining consistent accuracy above 80% (usually after 4-6 sessions), the mouse moves onto the discrimination experiment”

“On average, mice were able to learn both tasks, achieving an accuracy of 80% at cortical surface irradiance levels similar to the thresholds determined from electrophysiology experimentation”

The Method has also been updated accordingly.

“Successful learning of the association task comprised the mouse reaching an accuracy of 80% over an experimental session. Mouse 3 struggled to reach 80% but was moved onto the discrimination task anyway.”

“Due to initially high performance, discrimination tasks were terminated after a set number of trials, dependent on time available for experimentation.”

We have also maintained consistency with these points when describing the new freely-behaving experiments which have been added to the manuscript as part of this review.

Comment 9

The manuscript claims that the μ LED array provides high spatial resolution, with a 200 μ m pitch enabling precise neuronal targeting. However, in Figure 4, the reported light spread (600–800 μ m) suggests that individual μ LEDs may activate overlapping neuronal populations. This raises the question of whether the effective resolution is truly determined by the μ LED pitch, or if light scattering imposes limitations on selectivity. Additionally, while the study provides an estimate of spatial resolution based on electrophysiological data and optical modeling, a direct comparison with previous optogenetic stimulation technologies (e.g., fiber-optic stimulation, lower-density μ LED arrays) is not clearly presented. Given that one of the key claims of this work is improved spatial precision, it would be beneficial to include a quantitative comparison of spatial resolution with prior studies using different optogenetic stimulation methods.

Thank you for highlighting this point. Our μ LED array has a high density of stimulation sites greater than or on par with similar cortical surface stimulation approaches based on chronically implantable LEDs or μ LEDs (Pollmann et al. 2024; Rajalingham et al. 2021; Shin et al. 2024; Komatsu et al. 2017; Ohta et al. 2021; Lee et al. 2018); we quantify this comparison based on stimulation site spacing (Supplementary Table 1). Despite dense stimulation site spacing in our device (200 μ m) and minimal overlapping of the illuminated spots on the cortical surface by adjacent μ LEDs (270 μ m diameter) (Supplementary Fig. 3b), due to light scattering, overlap of the activation fields in for example layer 2/3 is likely, confirmed by our electrophysiology recordings and optical models (Fig. 4). To improve this, we discuss using a collimating interposer to improve spatial resolution (this has since been assembled and experimentally validated in response to Reviewer 2, comment 4). The spatial resolution within layer 2/3 could be improved from \sim 600 μ m to \sim 500 μ m, however further reduction of interposer hole size could not improve this resolution. We therefore attribute light scattering in the tissue as the dominant factor affecting spatial resolution. However, this omits the fact that photons exiting the interposer into the tissue may still be doing so at an angle. Therefore, we carried out a Monte Carlo simulation of a perfectly collimated source, similar to that which is possible with a DLP, projecting light over the same area and with the same delivered power (and cortical irradiance, because the surface areas are the same) (Supplementary Fig. 16d). We found that the illuminated optical profile was broadly the same as the interposer profile, confirming scattering as the dominant effect which will limit any non-penetrating, optical cortical surface stimulation technology.

Supplementary Figure 16 (d) Modelled optical profile for perfect collimator, projecting light over 200 μ m diameter area. Contours show 1 mW mm^{-2} threshold corresponding to equivalent optical power delivered to brain as (c).

We have provided additional discussion around this point.

“To confirm this, we modelled a perfectly collimated light source (as can be implemented in optical systems, such as digital light projectors - DLPs) incident on a 200 μm diameter area of brain tissue (Supplementary Fig. 16d). The illuminated volume and resolution is similar to the interposer models, demonstrating light scattering is indeed the dominant factor limiting any non-penetrating, optical cortical surface stimulation technology.”

Furthermore, in response to Reviewer 2, comment 1, we have significantly improved our literature review to where we have better contextualised our work in comparison to other implantable LED/ μLED approaches. In response to Reviewer 2, comment 2, we also have also better acknowledged other approaches to delivering spatially-confined light patterns to the cortex, which include laser (single-photon or two-photon), fibre-based or digital light projector (DLP) approaches.

Comment 10

The authors highlight the high spatial resolution enabled by the 10 \times 10 μLED array. However, if neuronal responses were recorded from only a single insertion site using a single 64-channel electrode probe, this significantly limits the validation of the claimed spatial resolution. The potential advantage of the μLED array—precisely targeting different neuronal populations—cannot be fully demonstrated if only a small region is sampled. Clarifying whether multiple insertion sites were used across experiments or discussing how the recorded neuronal responses generalize across the full μLED array would provide stronger evidence supporting the advantages of the high-density μLED system.

Thank you for bringing up this concern regarding the limited spatial recording capability provided by our single shank electrophysiology recording regime. A single insertion site, positioning the recording probe relative to the μLED array was used across experiments. This allowed us to make a direct comparison between different experiments/mice to establish a stimulation threshold based on single-unit spiking activity (Fig. 3e). We could then infer the spatial extent of activation by a single μLED by analysing single-unit spiking activity based on the activation of individual μLED s located near the recording probe (Fig. 4a, Fig. 4b). By doing this, we could establish spatial resolution by moving the activation field across the neuron.

Reviewer 1 had similar comments regarding our single insertion site electrophysiology recording (comments 4 and 5), where we provide a more detailed response, along with additional justifications in the manuscript which we hope address this point.

Minor comment

Comment 1

The study demonstrates that μLED stimulation can modulate neuronal activity and behavior. However, it remains unclear whether the frequency or temporal structure of the μLED pulses influences neuronal responses differently. Further discussion on how different stimulation frequencies affect neuronal activation and behavioral performance would provide deeper insight into the functional implications of the stimulation paradigm.

Exploration of the temporal code of the cortex is possible using our devices and control electronics, which can provide a wide range of pulse repetition rates and duty cycles. In this study, we focussed on obtaining stimulation thresholds and spatial resolution, and achieving robust behavioural responses. A stimulation protocol consisting of a 20 Hz pulse rate with 50% duty cycle was selected based on parameters already established in the Bathellier lab, targeted at maintaining strong continuous spiking responses from auditory neurons. While outwith the scope of this study, exploration of temporal coding would be an interesting future direction of work with these devices. We provided a more detailed response to Reviewer 2 (comment 9) and provided additional discussion in the final paragraph of the discussion section.

Comment 2

The manuscript outlines the positioning of the μ LED array on the cortical surface and the insertion of a 64-channel electrode probe at a 28° angle. However, it does not provide sufficient detail on how the electrode was aligned relative to the μ LED array to ensure accurate mapping of neuronal responses to specific stimulation sites. Given the study's emphasis on spatial resolution, further clarification on the alignment strategy would strengthen the interpretation of the recorded neuronal activity.

We attempted to keep the recording probe position consistent, relative to the position of the μ LED array, across electrophysiology recording sessions allowing us to generalise across experiments to obtain stimulation thresholds and spatial resolution. This was done using a microscope above the craniotomy (Supplementary Fig. 10a) with the angle (28°) and lateral proximity of the insertion site to the μ LED array chosen such that the recording sites would be as close as possible to the optical source. However, due to the compact nature of the craniotomy, some variation in this relative positioning was observed across experimental sessions. We have since updated the Results section to discuss this as a potential cause of inter-experiment variation in the strength of the neuronal response.

“Variation in the neuronal response between experiments could be due to variation in the relative positioning of the recording electrode array to the μ LED stimulation site (Supplementary Fig. 10a), the flatness of the μ LED array placement on the cortical window, or the overall electrode array insertion quality and surrounding tissue health.”

Comment 3

The authors describe the use of Parylene-C, Polyimide, and Silicone for chronic implantation, as these materials are commonly used in neural interfaces. However, the manuscript does not provide any direct biocompatibility testing or histological analysis to confirm the absence of immune responses or tissue damage over time. Given that the μ LED array is designed for long-term implantation, including supporting data on tissue response or referencing previous biocompatibility studies using similar packaging materials would strengthen the validation of its chronic use.

Histological analyses have since been carried out demonstrating the degree of immune response (Supplementary Fig. 22). We compared the immune response from Iba-1 staining, demonstrating elevated but small microglial response on the nearby the implantation site when comparing with the contralateral auditory cortex which was not implanted (Supplementary Fig. 2g). We found this immune response was localised to the cortical surface, and did not extend beyond layer 2/3.

Comment 4

The multi-unit analysis in Supplementary Figure 10 effectively demonstrates how μ LED stimulation modulates neuronal activity across different cortical depths. This analysis provides valuable insight into the spatial extent of optogenetic activation, which many readers may find crucial for understanding depth-dependent electrophysiological responses. Given its importance, incorporating this data into a main figure would improve accessibility and further strengthen the discussion on the spatial resolution of μ LED-driven neuronal activation.

Due to journal restrictions on article length and number of figures, it was difficult to incorporate the multi-unit analysis as a main figure in the manuscript. We appreciate the reviewer acknowledging these results however, and have provided additional reference to these within the Discussion, which we hope is sufficient to emphasise the additional insights provided by the multi-unit analysis.

“To gain further insight, we conducted a multi-unit analysis (MUA) using a thresholding approach, providing a neuronal response as a function of recording depth (Supplementary Fig. 17). These results show robust responses at the tip of the electrode array in layer 5. Optical modelling suggests the light did not penetrate this deep, even at the highest tested power (Fig. 4c), indicating we may be recording responses from layer 5 pyramidal neurons being driven through activation of their dendrites in layer 2/3, or by other second-order effects. **Spatial resolution appears to widen at**

deeper recording sites (Supplementary Fig. 17e), further suggesting integration of dendritic responses contributing to deeper neuronal responses.”

Bibliography

1. Klein, E., Gossler, C., Paul, O., Schwarz, U. T. & Ruther, P. High-yield indium-based wafer bonding for large-area multi-pixel optoelectronic probes for neuroscientific research. *Journal of Micromechanics and Microengineering* 29. Publisher: IOP Publishing, 095006. issn: 0960-1317. doi:10.1088/1361-6439/ab2a53 (July 2019).
2. Sekiguchi, H. et al. Adhesionable flexible GaN-based microLED array film to brain surface for in vivo optogenetic stimulation. *Appl. Phys. Express* 15, 046501. doi:10.35848/1882-0786/ac5ba3 (2022).
3. Gu, W., Wang, L., Wang, X., Zhao, C. & Guan, S. Large-Scale, High-Density MicroLED Array-Based Optogenetic Device for Neural Stimulation and Recording. *Nano Lett.* doi:10.1021/acs.nanolett.4c03645 (2024).
4. Clark, A. M. et al. An optrode array for spatiotemporally-precise large-scale optogenetic stimulation of deep cortical layers in non-human primates. *Commun Biol* 7, 1–18. doi:10.1038/s42003-024-05984-2 (2024).
5. McAlinden, N. et al. In vivo optogenetics using a Utah Optrode Array with enhanced light output and spatial selectivity. *en. Journal of Neural Engineering* 21. Publisher: IOP Publishing, 046051. issn: 1741-2552. doi:10.1088/1741-2552/ad69c3 (Aug. 2024).
6. Pollmann, E. H. et al. A subdural CMOS optical device for bidirectional neural interfacing. *Nat Electron* 7, 829–841. doi:10.1038/s41928-024-01209-w (2024).
7. Shin, H. et al. Transcranial optogenetic brain modulator for precise bimodal neuromodulation in multiple brain regions. *Nat Commun* 15, 10423. doi:10.1038/s41467-024-54759-0 (2024).
8. Taal, A. J. et al. Optogenetic stimulation probes with single-neuron resolution based on organic LEDs monolithically integrated on CMOS. *Nat Electron* 6, 669–679. doi:10.1038/s41928-023-01013-y (2023).
9. Rajalingham, R. et al. Chronically implantable LED arrays for behavioral optogenetics in primates. *Nat Methods* 18, 1112–1116. doi:10.1038/s41592-021-01238-9 (2021). Komatsu, M., Sugano, E., Tomita, H. & Fujii, N. A Chronically Implantable Bidirectional Neural Interface for Non-human Primates. English. *Frontiers in Neuroscience* 11. Publisher: Frontiers. issn: 1662-453X. doi:10.3389/fnins.2017.00514 (Sept. 2017).
10. Ohta, Y. et al. Micro-LED Array-Based Photo-Stimulation Devices for Optogenetics in Rat and Macaque Monkey Brains. *IEEE Access* 9, 127937–127949. issn: 2169-3536. doi:10.1109/ACCESS.2021.3111666 (2021).
11. Ji, B. et al. Flexible polyimide-based hybrid opto-electric neural interface with 16 channels of micro-LEDs and electrodes. *Microsyst Nanoeng* 4, 1–11. doi:10.1038/s41378-018-0027-0 (2018).
12. Kwon, K. Y., Sirowatka, B., Weber, A. & Li, W. Opto- ECoG array: a hybrid neural interface with transparent ECoG electrode array and integrated LEDs for optogenetics. *IEEE Trans Biomed Circuits Syst* 7, 593–600. doi:10.1109/TBCAS.2013.2282318 (2013).
13. Lee, S. H. et al. Optogenetic control of body movements via flexible vertical light-emitting diodes on brain surface. *Nano Energy* 44, 447–455. doi:10.1016/j.nanoen.2017.12.011 (2018).
14. Steude, A., Witts, E. C., Miles, G. B. & Gather, M. C. Arrays of microscopic organic LEDs for high-resolution optogenetics. *Science Advances* 2, e1600061. doi:10.1126/sciadv.1600061 (2016).

15. Ceballo, S. et al. Cortical recruitment determines learning dynamics and strategy. *Nat Commun* 10, 1479. doi:10.1038/s41467-019-09450-0 (2019).
16. Pinto, L. et al. Task-Dependent Changes in the Large-Scale Dynamics and Necessity of Cortical Regions. *English. Neuron* 104, 810–824.e9. issn: 0896-6273. doi:10.1016/j.neuron.2019.08.025 (Nov. 2019).
17. Adesnik, H. & Abdeladim, L. Probing neural codes with two-photon holographic optogenetics. *eng. Nature Neuroscience* 24, 1356–1366. issn: 1546-1726. doi:10.1038/s41593-021-00902-9 (Oct. 2021).
18. Tian, F., Zhang, Y., Schriver, K. E., Hu, J. M. & Roe, A. W. A novel interface for cortical columnar neuromodulation with multipoint infrared neural stimulation. *en. Nature Communications* 15, 6528. issn: 2041-1723. doi:10.1038/s41467-024-50375-0 (Aug. 2024).
19. Sych, Y., Chernysheva, M., Sumanovski, L. T. & Helmchen, F. High-density multi-fiber photometry for studying large-scale brain circuit dynamics. *en. Nature Methods* 16, 553–560. issn: 1548-7105. doi:10.1038/s41592-019-0400-4 (June 2019).
20. Vöröslakos, M. et al. HectoSTAR LED Optoelectrodes for Large-Scale, High-Precision In Vivo Opto-Electrophysiology. *Advanced Science* 9, 2105414. doi:10.1002/adv.202105414 (2022). Wu, F. et al. Monolithically Integrated LEDs on Silicon Neural Probes for High-Resolution Optogenetic Studies in Behaving Animals. *Neuron* 88, 1136–1148. doi:10.1016/j.neuron.2015.10.032 (2015).
21. Caldwell, R. et al. Characterization of Parylene-C degradation mechanisms: In vitro reactive accelerated aging model compared to multiyear in vivo implantation. *Biomaterials* 232, 119731. doi:10.1016/j.biomaterials.2019.119731 (2020).
22. Kim, T. et al. Thermal effects on neurons during stimulation of the brain. *J Neural Eng* 19, 056029. doi:10.1088/1741-2552/ac9339 (2022).
23. McAlinden, N. et al. Thermal and optical characterization of micro-LED probes for in vivo optogenetic neural stimulation. *Opt. Lett., OL* 38, 992–994. doi:10.1364/OL.38.000992 (2013).
24. McAlinden, N., Gu, E., Dawson, M. D., Sakata, S. & Mathieson, K. Optogenetic activation of neocortical neurons in vivo with a sapphire-based micro-scale LED probe. *Front. Neural Circuits* 9. doi:10.3389/fncir.2015.00025 (2015).
25. Owen, S. F., Liu, M. H. & Kreitzer, A. C. Thermal constraints on in vivo optogenetic manipulations. *Nat Neurosci* 22, 1061–1065. doi:10.1038/s41593-019-0422-3 (2019).
26. Holtmaat, A. et al. Long-term, high-resolution imaging in the mouse neocortex through a chronic cranial window. *en. Nature Protocols* 4. Publisher: Nature Publishing Group, 1128–1144. issn: 17502799. doi:10.1038/nprot.2009.89 (Aug. 2009).
27. Abbasi, A. et al. Brain-machine interface learning is facilitated by specific patterning of distributed cortical feedback. *Science Advances* 9, eadh1328. doi:10.1126/sciadv.adh1328 (2023).
28. Yaroslavsky, A. N. et al. Optical properties of selected native and coagulated human brain tissues in vitro in the visible and near infrared spectral range. *Phys Med Biol* 47, 2059–2073. doi:10.1088/0031-9155/47/12/305 (2002).
29. Aravanis, A. M. et al. An optical neural interface: in vivo control of rodent motor cortex with integrated fiberoptic and optogenetic technology. *J. Neural Eng.* 4, S143–S156 (2007).
30. Yona, G., Meitav, N., Kahn, I. & Shoham, S. Realistic Numerical and Analytical Modeling of Light Scattering in Brain Tissue for Optogenetic Applications. *en. eNeuro* 3. issn:2373-2822. doi:10.1523/ENEURO.0059-15.2015 (Jan. 2016).

31. Bagur, S. et al. A spatial code for temporal information is necessary for efficient sensory learning. *Science Advances* 11, eadr6214. doi:10.1126/sciadv.adr6214 (2025).
32. Glickman, B. & LaLumiere, R. T. Theoretical considerations for optimizing the use of optogenetics with complex behavior. *Curr Protoc.* 3, e836 (2023).
33. Verdier, A. et al. Enhanced perceptual task performance without deprivation in mice using medial forebrain bundle stimulation. *Cell Rep Methods* 2, 100355. doi:10.1016/j.crmeth.2022.100355 (2022).

Author responses to reviewer comments of the paper titled:
**Chronically implantable μ LED arrays for optogenetic cortical surface
stimulation in mice**
for submission to Nature Communications.

DOCUMENT VERSION 2: addressing remaining concerns following initial author rebuttal.

Ryan Greer^{1†*}, Antonin Verdier^{2†}, Emma Butt¹, Yunzhou Cheng¹, Ella Callas², Niall McAlinden¹,
Alicia Aniorte², Eya Mabrouk Kakaouia², Magdalena Pereyra², Martin D. Dawson¹, Brice
Bathellier^{2‡} & Keith Mathieson^{1‡*}

Affiliations:

¹University of Strathclyde, Institute of Photonics, Glasgow, G1 1RD, UK

²Institut Pasteur, Institut de l'Audition, Paris, 75012, France

[†]These authors contributed equally

[‡]Joint senior authors

*Corresponding authors; contact at ryan.greer@strath.ac.uk or keith.mathieson@strath.ac.uk

Dear editor and review panel,

The authors thank you for reviewing our rebuttal letter and for your feedback on our revised manuscript titled “Chronically implantable μ LED arrays for optogenetic cortical surface stimulation in mice” for publication in Nature Communications. We were encouraged by the feedback and have addressed the remaining concerns in this document and in a revised version of the manuscript and supplementary material document. Please do not hesitate to reach out to the corresponding authors (Ryan Greer or Keith Mathieson) if you have any further questions or comments.

Yours sincerely,

The authors

26th October 2025

Reviewer #1

The authors have provided a carefully revised manuscript which discusses and as best as could be expected addresses the issues I brought up in my previous report. I would only ask that, for clarity, the following part is amended as follows to avoid any ambiguity about whether these additional steps have already been implemented (line 377):

“The trade-off is a reduction in spatial resolution due to the 150 μm standoff between source and brain surface. IN THE FUTURE, this COULD be reduced by using ...” (or of course similar wording that clarifies this point)

Apart from this minor change (that will not require me to look at the manuscript again), I recommend publication of the study as is.

Thank you for highlighting this, the discussion text has been updated as follows:

“The trade-off is a reduction in spatial resolution due to the 150 μm standoff between source and brain surface. **In the future, this could** be reduced by using top-emitting devices or removing the LED structures from their substrates, either through laser lift-off processes or through selective chemical etching”

Reviewer #2

In the revised manuscript, the authors carried out new experiments demonstrating the use of their device in freely behaving behavioral paradigms, and they expanded the discussion to better position their work relative to other existing technologies. From these perspectives, the revised document is appropriate in terms of significance. The authors have now addressed what they originally claimed as a major technical advantage (freely-behaving experiments) and have more clearly contextualized their contributions within the broader literature.

Furthermore, the authors included experimental validations of their prior optical and thermal simulations. While these validations remain indirect — for example, relying on IR imaging in air or water rather than in vivo, or optical measurements without considering tissue scattering and absorption — they are reasonable given the technical challenges of in vivo validation and the supporting role these data play for the core technological contributions.

Overall, I am positive about this work. Many of the previous major points have been addressed, and the technical advancement has strong potential for broad impact in the field. However, I have several remaining concerns regarding specific experiments and internal consistency in the revised manuscript. These issues should be clarified to improve scientific rigor and highlight important caveats.

Remaining concerns

Comment 1

The addition of freely behaving behavioral experiments is exciting. However, it is unclear whether opsin-negative control mice were included. If these experiments were conducted, the results should be shown; if not, please justify their omission (noting that some other experiments already included controls).

We did not include control mice for the new freely-behaving experiments as we had confirmed there were no confounding effects (such as thermally-induced neuronal firing) influencing behavior in the head-fixed experiment. Since the implantation and stimulation protocol were very similar (3x3 patterns up to 10 mW/mm²), and to minimise the number of animals used, we decided that further control experiments were not essential. However, we did improve the controls within the experiments by using a masking light (external light providing bright conditions that desensitizes the retina) in these experiments, improving upon the head-fixed experiment and mitigating the possibility of the mouse visually perceiving the light from the device. We also note that the experiment is a discrimination task with identical illumination patterns activating spatially separated locations on the cortex, which also provides an element of inherent control within the experiment.

Thank you for highlighting this. We have now included the necessary detail in the results.

“Control (opsin-negative) mice were not included as the stimulation protocol used here (Fig. 6c) was identical to the head-fixed behavioural experiments - these already confirmed successful optogenetic activation and no confounding effects influencing behaviour. Per- μ LED cortical irradiance was set at 10 mW/mm² and progressively reduced to 1 mW/mm² in the cases where the behavioural outcome accuracy was maintained at a strong level (typically >80%) across sessions.”

Comment 2

Most behavioral tasks used 3x3 LED activation patterns rather than single-pixel stimulation. Is this because individual LEDs within a 3x3 block cannot be independently addressed simultaneously due to PN junction wiring? If so, this limitation undercuts the repeated claim of spatial precision. Please clarify and justify this when presenting the behavioral experiments.

For all behavioural experiments reported, we utilised 3x3 μ LED activation patterns. This was not due to any limitation in the device wiring, as individual μ LEDs can be illuminated; although diagonal patterns require more complex control due to matrix addressing (Supplementary fig. 9d). We chose 3x3 patterns to activate a broad region, replicating previous work in the Bathellier lab where a Digital Light Projector (DLP), illuminating spots of 400 μ m diameter, was used for optogenetic activation of the mouse auditory cortex to drive behavioural outcomes (Ceballo, Bourg et al. 2019; Ceballo, Piwkowska et al. 2019). Here, we adopted a similar approach by activating a broad cortical surface area using a 3x3 pattern instead of individual pixels. We felt that this was correct approach to start with, as these were our first studies using the μ LED arrays for behaviour. We have provided additional clarification on this when introducing the behavioural experiments.

“Although the device facilitates individual pixel operation, a 3x3 square of μ LEDs was used to illuminate one corner of the diagonal depending on the trial type (go or no-go). This achieved a broad area of illumination and was based on previous experiments using a digital light projector (DLP) system to illuminate 400 μ m diameter regions of the AC, to drive behavioural outcomes in Chr2-expressing mice (Ceballo, Bourg et al. 2019; Ceballo, Piwkowska et al. 2019).”

Since this study, we have continued the freely-behaving experiments utilising individual pixel stimulation and adjustment of the separation between go and no-go stimuli. The data are preliminary and only for a single animal, thus we have not included them in this manuscript. Despite being out with the scope of this manuscript, we include the data below for interest and to evidence the single pixel performance to the referee.

(Figure caption) Same experimental paradigm as Fig. 6e for single-pixel discrimination. Sessions 1-4: illuminated spots are separated by 2 mm; session 5: 1 mm separation; sessions 6-8: 0.566 mm separation.

Comment 3

Many in vivo experiments were conducted at ~ 10 mW/mm² at the cortical surface, which would yield >1 °C heating based on the authors' own simulations. The rebuttal argues that because cortical windows are absent in chronic implants, the same irradiance can be achieved at <3 mA per LED (Supplementary Fig. 1d), implying this is safe. This reasoning is flawed: removal of the window decreases the source–tissue distance and can increase heating at the interface, even if the driving current is lower. This caveat should be explicitly acknowledged.

We absolutely agree with the referee's comment that reducing the source-tissue distance increases the thermal effects from the implant, and at no point did we mean to suggest otherwise. We perhaps did not make this point clear in the text and rebuttal and apologise for any confusion. We have clarified our reasoning as follows.

The cortical window experiments used single pixel illumination, which requires significantly less power than patterned stimulation. As can be seen in Fig 2b, single pixel illumination for all currents used never heats the tissue interface by a ΔT of 0.5 °C. This thermal modelling (and all of the thermal modelling we conduct) is for our worst-case scenario, which is direct implantation on the cortical surface without any cortical window. We only modelled direct implantation because we were also worried about the greater thermal risk in this case. Conclusions drawn from thermal modelling in the context of cortical window experiments simply ignore effect of the window as a thermal insulator. Therefore to improve the clarity of the text, we remove the mention of the cortical window in the sections relating to thermal effects. Specifically:

Discussion

Original: "Activation thresholds were determined through electrophysiology recordings from the auditory cortex of mice, with less than 5 mW/mm² on the cortical surface driving robust neuronal responses. The μ LEDs can drive these irradiances at less than 3 mA when the device is in direct contact with the tissue (Supplementary Fig. 1d), resulting in safe temperature increases (Fig. 2b). With the cortical window implanted, the drive current increases to ~ 5 mA, which is still safe for single μ LED stimulation as utilised in the electrophysiology experiments. The cortical window used here further acts as a thermal insulator."

Revised: "Activation thresholds were determined through electrophysiology recordings from the auditory cortex of **awake** mice, with less than 5 mW/mm² on the cortical surface driving robust neuronal responses. **This corresponds to a drive current of ~ 5 mA (Supplementary fig. 1d), resulting in safe temperature increases according to thermal models for single-pixel illumination (Fig. 2b).**"

Results

Original: "On average, mice were able to learn both tasks, achieving an accuracy of 80% at cortical surface irradiance levels similar to the thresholds determined from electrophysiology experimentation (5 - 10 mW/mm²) but over an increased illuminated area and volume as was shown by the optical modelling in Fig. 4g. Since the cortical window is not present for chronic implantation, these irradiances can be driven at a per- μ LED current of less than 3 mA (Supplementary Fig. 1d). Thermal analyses suggest this current level is safe, even for up to nine μ LEDs illuminated simultaneously (Fig. 2b)."

Revised: “On average, mice were able to learn both tasks, achieving an accuracy of 80% at cortical surface irradiance levels similar to the thresholds determined from electrophysiology experimentation (5 - 10 mW/mm²) but over an increased illuminated area and volume as was shown by the optical modelling in Fig. 4g. **Consistent behavioural responses were observed as low as 5 mW/mm², which** can be driven at a per- μ LED current of less than 3 mA (Supplementary Fig. 1d). Thermal analyses suggest this current level is safe, even for up to nine μ LEDs illuminated simultaneously (Fig. 2b, **Supplementary fig. 7j**).”

Introduction:

Original: “Electrophysiology recordings from the mouse auditory cortex demonstrate robust activation thresholds of 4 - 5 mW/mm² on the cortical surface. This corresponds to drive currents of <3 mA for direct cortical implantation, resulting in safe temperature changes even with several μ LEDs active simultaneously.”

Revised: “Electrophysiology recordings from the mouse auditory cortex demonstrate robust activation thresholds of 4 - 5 mW/mm² on the cortical surface. This corresponds to a drive current of **~5 mA**, resulting in safe temperature changes **predicted by our modelling for single-pixel illumination.**”

We have also made it clearer when introducing thermal models that they always assume direct implantation on the cortical surface.

“Infrared (IR) imaging was used in conjunction with an equivalent 2D rotationally-symmetric mathematical model replicating the **device with flexible packaging**, chronically implanted **with the sapphire surface in direct contact with** cortical tissue”

Comment 4

Throughout the manuscript, the authors emphasize the accuracy of their optical and thermal simulations. Given the simplifying assumptions (water model, omission of absorption, dendritic contributions, etc.), it would be more accurate to present these as approximations that may over- or under-estimate in vivo outcomes. Please moderate claims of “accuracy” and frame results as approximate guides with caveats.

Thank you for highlighting this. For our thermal analysis, we apply simplifications in our model of the brain, using the thermal coefficients for water, and have updated the text to emphasise this.

“Appropriate material thermal properties were applied, with the brain approximated as water. To verify the accuracy of the modelled device, the sapphire surface of the μ LED array was imaged using an IR camera in air (Supplementary Fig. 6), which agreed closely with the modelled output (Fig. 2a). These experiments, along with previous work demonstrating the accuracy of this model architecture with in-vivo IR imaging (McAlinden et al. 2024), gave **us confidence in** the thermal models to **provide a good approximation of** experimental conditions.”

In optical models at the wavelengths relevant here, the important physical parameters are the absorption and scattering coefficients and the refractive index. In our optical models, we began with the default

parameters for water and then applied appropriate scattering and absorption coefficients for brain tissue. Therefore, only the refractive index of our modelled brain tissue is shared with water and this is approximately in the same range as the experimental values quoted in the literature (Binding et al. 2011) ($n = 1.33$ for water and between 1.34-1.37 for mouse cortex). We do however omit the effects of light absorption in vasculature as this is hard to estimate as a bulk effect and will vary over cortical layers and by animal. We have updated the results and method text to better explain this.

Results:

“Brain tissue was modelled using the scattering and absorption parameters for grey/white matter (Yaroslavsky et al. 2002, Bernstein et al. 2008) and setting the refractive index to that of water ($n=1.33$). This omits contributions from structures such as the vasculature. The device model was experimentally validated with in-air optical measurements (Supplementary fig. 3), giving us confidence in the optical models to provide a good estimate of experimental conditions.”

Method:

“The refractive index of water was used as it approximately matches the reported values for brain tissue (Binding et al. 2011), while incorporating suitable scattering coefficients to approximate the optical characteristics of the brain”

We experimentally validated both optical and thermal models as far as possible with in-air measurements, to provide confident estimates of in-vivo effects. However, the discussed changes have now moderated the relevant parts of the text to emphasise that modelled results should be considered approximations.

Comment 5

The histological analysis raises questions. When layer 1 is included, the implanted side shows only a modest increase in Iba-1+ cells (228 ± 24 vs. 181 ± 18 per mm^2), albeit statistically significant. Excluding layer 1 increases the numerical difference but eliminates statistical significance. The use of a non-parametric Wilcoxon test may not capture this nuance appropriately.

The overall Iba-1 signal appears surprisingly low given that the dura was removed and the device directly contacted cortex. Was tissue collected long after implantation, potentially missing acute responses? Please indicate the interval between surgery and tissue harvest.

Adding GFAP staining (astrocyte marker) would provide an important complementary measure, particularly given the low microglial response reported.

We agree with the referee that it is difficult to capture the statistical nuances in the histological analysis. However, we used a non-parametric test because the data did not pass commonly used normality tests (Lilliefors, and Cramer-Von Mises tests), possibly due to the low sample size. In this case, the standard and conservative procedure is to use a non-parametric test such as the paired Wilcoxon signed rank test, which was not significant when excluding layer 1 from the analysis. We agree with the referee that there are two measurements in the “excludes layer 1” panel, which indicate a significantly larger Iba-1+ cell count on the implanted side. However, the data does not all trend in the same direction when the implanted

side is compared with the contralateral side, which reduces statistical significance across the sample population. There is also variability in staining across the slice (visible in the example shown), which is likely to complicate interpretation further.

The Iba-1 signal is low because the tissue samples were collected between 83 and 101 days after implantation since we completed an extensive series of behavioural tasks. In summary, after implantation, mice were allowed to recover for 1 month and then behavioural training/experiments (5 days per week) lasted between ~50-70 days. Our measurements reflect the long-term impact of the μ LED array after full operation, which we feel is most relevant for future users. Note also that the behavioural performance was maintained at a high level until the tissue was harvested, indicating the excellent stability of the cortical network over more than 2 months of training with the μ LED arrays. We edited the manuscript to provide this important piece of information:

Results:

“Histological analysis after three months chronic implantation, demonstrates plastic deformation around the implant site”

Discussion:

“Histological analyses from mice implanted for three months showed an increase in immune response on the brain surface, and plastic deformation around the implant site (Supplementary Fig. 2).”

Method:

“Tissue samples were obtained and analysed from mice chronically implanted with the μ LED array between 83-101 days; post-surgery recovery: 30 days, behavioural training: 50-70 days.”

We agree that the F+GFAP staining would be a useful addition, but unfortunately this staining failed when we performed the histological analysis and we do not have comparable samples at hand to complement Iba1 staining.

Comment 6

Supplementary Fig. 10b shows that filtering introduces potential artifacts, and the apparent spiking activity may in part reflect residual LFP responses after filtering. The single-unit spike sorting (Fig. 3c, right) is therefore not fully convincing. It would strengthen confidence if the authors showed examples of clearly identified spikes in raw, unfiltered traces at higher temporal resolution (e.g., 10 ms windows), even if superimposed on LFPs. While the LFP responses themselves suggest successful opsin activation, the claims of single-unit resolution need clearer support.

Prior to spike sorting, we use a standard approach, applying a filter with a wide passband [300 Hz, 5.5 kHz] which removes the LFP component which we do not analyse, but the small spiking signals (~1-3 ms in length) should remain. This can be seen in the raw, unfiltered trace which we added (Supplementary fig. 10b) which corresponds to the same recording channel and time window as Fig. 3c where we highlight single unit spike times identified from spike sorting, and their average waveforms. However, it should be noted that spike sorting algorithms identify spikes by considering the signal across several channels nearby

to the neuron and typically using a principal component analysis to extract the salient features. Supplementary fig. 10b has been updated to show a zoomed inset, and a further zoom to a 10 ms time window to clearly show spikes embedded in the unfiltered trace.

Supplementary Figure 10 (b) Electrophysiology waveforms **from a single electrode** corresponding to same μ LED and trial as Fig. 3c. Top: raw trace; middle: raw trace after filtering with bandpass FIR filter (cutoffs 300 Hz and 5.5 kHz); bottom: raw trace after common median reference subtracted (no filtering). **Insets show zoom of raw, unfiltered trace showing spiking waveforms induced by optogenetic stimulation.**

Comment 7

The choice of 20 Hz at 50% duty cycle is not ideal: it imposes a higher thermal load and diminishes the temporal precision that is a key advantage of optogenetics. Please explicitly note that this protocol was chosen as a proof-of-concept for behavioral salience, but is not optimal for optogenetic control.

Thank you for highlighting this. We appreciate that the chosen optogenetic stimulation protocol is not ideal and provided a detailed justification for this in our response to comment #9 from the initial review, but we did not add this to the manuscript. Stimulation protocol timing was based on parameters previously established in the Bathellier lab to achieve robust neuronal signalling and behavioural responses (Ceballos, Bourg et al. 2019; Bagur et al. 2025). We have now provided this justification in the manuscript when introducing the experimental protocol in both electrophysiology and behaviour sections, and re-emphasised this in the discussion.

Discussion:

“This study was limited to spatial patterning with fixed timing (20 Hz, 50% duty cycle), **selected to drive strong neuronal activation and behavioural salience. This protocol was chosen as a proof-of-concept for behavioral experiments, but is not necessarily optimal for optogenetic control. Future studies could include** optimisation of the optogenetic stimulation protocol and more sophisticated temporal patterning of the μ LED array could also allow exploration of the involvement of temporal features in the neural code of the cortex.”

We also emphasized this further as follows:

Results:

“Each experiment comprised a set of trials, each trial illuminating an individual μ LED at a set drive current for 10 pulses (20 Hz, 50% duty cycle) with 500 ms wait between trials. **This optogenetic protocol was chosen based on previous use of these parameters to drive strong neuronal activation (Ceballo, Bourg et al. 2019; Bagur et al. 2025).**”

“**Again, this optogenetic protocol was selected to drive consistent neuronal activation, aiming to achieve strong salience for the animal.**”

Comment 8

It remains difficult to track which encapsulation scheme (material, thickness, coatings) was used in each experiment. Please provide a supplementary table summarizing encapsulation conditions across all experimental contexts for clarity.

We have now included Supplementary table 5 which summarises the packaging details for devices used in all in-vivo experiments, and refer to this in the text. Thank you for highlighting this. Discussion:

“Chronic implantation of the μ LED arrays allowed repeated optogenetic stimulation of the same neuronal ensembles over **many days (50-70 days following post-surgery recovery)**, as well as experiments in freely-behaving animals, with a relatively straightforward surgical procedure **enabling implantations across many animals (Supplementary table 5).**”

Experiment type	Experiment #	Device type	Encapsulation	Cortical window used
Electrophysiology	1, 2, 1-2, 3, 4, 5	Acute	PaC (7 μm)	Yes
Behaviour (head-fixed)	1	Chronic	PaC (15 μm)	No
	2	Chronic	PaC (15 μm)	No
	3	Chronic	PaC (15 μm)	No
	4	Chronic	PaC (15 μm)	No
	5	Chronic	PaC (15 μm)	No
	6	Chronic	PaC (15 μm)	No
Behaviour (freely-moving)	1	Chronic	PaC (15 μm)	No
	2	Chronic	PaC (30 μm) / silicone	No
	3	Chronic	PaC (15 μm)	No
	4	Chronic	PaC (15 μm)	No

Supplementary Table 5: Summary of devices and encapsulation schemes used in experiments; each row represents a different device. Note, the use of silicone in one of the encapsulation strategies covers only the device packaging and not the stimulation area; full details are given in the methods section.

Reviewer #3

The concerns regarding the behavioral experiments, LED operational parameters, and temperature increase appear to be well addressed. However, my primary and most significant concern, which was initially raised regarding the device's suitability for chronic applications, remains. The rigidity of a device is largely determined by its thickness and the Young's modulus of its constituent materials. In our view, the device's thickness and materials are not suitable for it to be considered a chronic implant. Furthermore, it is difficult to obtain precise information about the chronic in-vivo experiments conducted to resolve this issue, and we have reservations about whether the immune response data has been interpreted correctly. Further discussion on these points is required.

Remaining concerns

Comment 1

Our concern regarding the device's lack of 'flexibility' from its rigid sapphire substrate was misunderstood. The response incorrectly addresses surgical 'positioning' instead of this fundamental material limitation. The use of a rigid sapphire substrate makes any discussion of flexibility meaningless. Because the justification provided does not resolve this issue, the response is insufficient. Please provide a revised answer focused on the material's rigidity.

Thank you for highlighting this point again. We agree that due to the material and thickness of the μ LED substrate (150 μ m sapphire), the stimulation region of the device cannot achieve flexible, conformable contact with the brain surface. Previously, we updated the text to clarify that the stimulation area of the μ LED array remains rigid, constrained by the GaN-on-sapphire fabrication process, but the supporting polyimide PCB packaging is flexible. The flexibility of the packaging is a valuable attribute of our devices for the surgical positioning and skull fixation, allowing minimally-invasive positioning of the μ LED array within a compact craniotomy (equivalent to the size of the device, $\sim 2.2 \times 2.2 \text{ mm}^2$), and helping to facilitate chronic implantation and studies in behaving mice. We understand the referees point around possible confusion with devices that utilise fully flexible components as the neural interface, which is a distinct approach that is rightly gathering much attention in the field. As such, we agree that some readers may miss the fact that our μ LED array is rigid, especially since the detailed device description of device material is after the introduction. Therefore, we have updated several parts of the text where the term “flexible” is mentioned, and either removed its use, or fully clarified that we refer to the package and not the μ LED array itself.

Abstract:

Original: “We fabricated 100-element μ LED arrays (200 μ m pixel pitch, $2 \times 2 \text{ mm}^2$ footprint) coupled into flexible, chronically implantable systems for optogenetic stimulation of the mouse cortex.”

Revised: “We fabricated 100-element μ LED arrays (200 μ m pixel pitch, $2 \times 2 \text{ mm}^2$ footprint) coupled into chronically implantable systems for optogenetic stimulation of the mouse cortex.”

Introduction:

Original: “The μ LED array is integrated into miniaturised opto-electronic systems for in-vivo experimentation in mice, including a flexible, chronically implantable system which remains stable for up to 300 hours experimental time.”

Revised: “The μ LED array is integrated into miniaturised, opto-electronic systems for in-vivo experimentation in mice. **Chronic implantation of the μ LED array is enabled through integration to a flexible package (though the array itself is rigid), and can** remain stable for up to 300 hours experimental time.”

Figure 3 caption:

Original: “(h) Schematic of chronically implantable, flexible device and photograph of side-profile;”

Revised: “(h) Schematic of chronically implantable **device packaging** and photograph of side-profile;”

Results:

Original: “Infrared (IR) imaging was used in conjunction with an equivalent 2D rotationally-symmetric mathematical model replicating the flexible device”

Revised: “Infrared (IR) imaging was used in conjunction with an equivalent 2D rotationally-symmetric mathematical model replicating the **device with flexible packaging**”

Discussion:

Original: “Furthermore, flexible, biocompatible packaging enabled chronic implantation in the mouse model and scalability through potential integration of several μ LED arrays tiled on a single device (Supplementary Fig. 5).”

Revised: “Furthermore, flexible, biocompatible packaging enabled chronic implantation **of the μ LED arrays** in the mouse model. **Although the μ LED arrays are rigid, scalability over a larger surface could be achieved through** integration of several μ LED arrays tiled on a single device (Supplementary Fig. 5).”

Added: “Histological analyses from mice implanted for three months showed an increase in immune response on the brain surface, and plastic deformation around the implant site (Supplementary Fig. 2). This is expected given the μ LED array is rigid, demonstrating similar effects to standard cranial window implantations (Holtmaat et al. 2009). Nevertheless, behavioural performance was maintained at a high-level, highlighting the stability of the cortical network over the duration of the chronic implantation.”

We hope our new updates to the manuscript fully resolve this concern.

Comment 2

We question the authors' assertion that the 2x2 mm² cortical area is 'flat.' This claim is directly refuted by the authors' own data in Supplementary Figure 2(a), which clearly shows tissue deformation at the implant site caused by the rigid device. A visible difference in compression exists between the implanted and non-implanted hemispheres, which would not occur if the surface were truly flat. Conformal contact with soft tissue is governed by bending stiffness, not the topography of the target area. Given the high stiffness imparted by the sapphire and the

device's thickness, the claim that it can achieve conformal contact on the soft cortical surface is not well-supported. This comment has not been satisfactorily addressed.

Thank you for highlighting this. It was not our intention in the rebuttal to suggest that the cortical surface is flat; rather the small size of the μ LED array ($2 \times 2 \text{ mm}^2$) and its planar geometry mean that the curvature mismatch with the brain is reduced. When introducing the histology data, we do mention that there is plastic deformation around the implant site comparable to the geometry of the device. We rely on the device's stiffness to maintain its shape against the brain's soft tissue. From a surgical standpoint, we believe that the discrepancies between the curvature of the brain and the rigid μ LED array at this scale are small enough that it can be resolved by applying reasonable amounts of pressure on the implant during surgery, as is the case with cranial window implantations (Holtmatt et al. 2009). This rigidity thus allows for robust contact between the cortical surface and the μ LED arrays as well as ensuring that the implant geometry is not modified by the brain tissue. Within the manuscript, we do not describe this as conformal contact, and indeed the word "conformal" never appears in our manuscript in this context. We have expanded the text to better explain this constraint and further distinguish our device from flexible neural interfaces.

Results:

Original: "However, the stimulation area of the μ LED array remains rigid, constrained by the GaN-on-sapphire fabrication process; the brain locally adapts to this constraint in the same way as it does as for cranial window implantations (Holtmatt et al. 2009). Histological analysis demonstrates plastic deformation around the implant site, with the size of the affected region comparable to the geometry of the μ LED array (Supplementary Fig. 2)"

Revised: "However, the stimulation area of the μ LED array remains rigid, constrained by the GaN-on-sapphire fabrication process. **Therefore, the device cannot achieve conformal contact with the brain surface as with flexible neural interfaces, however** the brain locally adapts to this constraint in the same way as it does as for cranial window implantations (Holtmatt et al. 2009). Histological analysis **after three months chronic implantation**, demonstrates plastic deformation around the implant site, with the size of the affected region comparable to the geometry of the μ LED array (Supplementary Fig. 2)."

Comment 3

The response lacks information on the total implantation period for the histology study, a critical parameter for the authors' claims. To validate the device as a 'chronic implant,' the immune response must be assessed following a genuinely long-term implantation. This essential information has not been provided, which weakens the claims of chronic biocompatibility. The duration needs to be explicitly stated.

The tissue samples were collected between 83 and 101 days after implantation in three animals that underwent behavioural training with the device. After implantation, mice were allowed to recover for 1 month (necessary for AAV virus expression) and then behavioural training (5 days per week) lasted between ~50-70 days. Our measurements reflect the long-term impact of the μ LED array after full operation, which is most relevant for future users. Note also that the behavioral performance was

maintained at a high level until the tissue was harvested, indicating the excellent stability of the cortical network over more than 2 months of training with the μ LED arrays. We edited the manuscript accordingly as discussed in response to reviewer 2, comment 5.

Bibliography

1. Ceballo, S. et al. Cortical recruitment determines learning dynamics and strategy. *Nat Commun* 10, 1479. doi:10.1038/s41467-019-09450-0 (2019).
2. Ceballo, S., Piwkowska, Z., Bourg, J., Daret, A. & Bathellier, B. Targeted Cortical Manipulation of Auditory Perception. *Neuron* 104, 1168–1179.e5. doi:10.1016/j.neuron.2019.09.043 (2019).
3. McAlinden, N. et al. In vivo optogenetics using a Utah Optrode Array with enhanced light output and spatial selectivity. *en. Journal of Neural Engineering* 21. Publisher: IOP Publishing, 046051. issn: 1741-2552. doi:10.1088/1741-2552/ad69c3 (Aug. 2024).
4. Jonas Binding, Juliette Ben Arous, Jean-François Léger, Sylvain Gigan, Claude Boccara, and Laurent Bourdieu, "Brain refractive index measured in vivo with high-NA defocus-corrected full-field OCT and consequences for two-photon microscopy," *Opt. Express* **19**, 4833-4847 (2011).
5. Yaroslavsky, A. N. et al. Optical properties of selected native and coagulated human brain tissues in vitro in the visible and near infrared spectral range. *Phys Med Biol* 47, 2059–2073. doi:10.1088/0031-9155/47/12/305 (2002).
6. Bernstein, J. G. et al. Prosthetic systems for therapeutic optical activation and silencing of genetically-targeted neurons. *Proc SPIE Int Soc Opt Eng* 6854, 68540H. doi:10.1117/12.768798 (2008).
7. Bagur, S. et al. A spatial code for temporal information is necessary for efficient sensory learning. *Science Advances* 11, eadr6214. doi:10.1126/sciadv.adr6214 (2025).
8. Holtmaat, A. et al. Long-term, high-resolution imaging in the mouse neocortex through a chronic cranial window. *en. Nature Protocols* 4. Publisher: Nature Publishing Group, 1128–1144. issn: 1750-2799. doi:10.1038/nprot.2009.89 (Aug. 2009).